# Inheritance of H3K9 methylation regulates genome architecture in *Drosophila* early embryos

Nazerke Atinbayeva [1,2], Iris Valent [3], Fides Zenk[4], Eva Loeser[1], Michael Rauer [1], Shwetha Herur[1], Piergiuseppe Quarato[5], Giorgos Pyrowolakis[6], Alejandro Gomez-Auli [1], Gerhard Mittler [1], Germano Cecere [7], Sylvia Erhardt [3], Guido Tiana[8], Yinxiu Zhan [9✉] & Nicola Iovino [1✉]

## Abstract

**Constitutive heterochromatin is essential for transcriptional silencing and genome integrity. The establishment of constitutive heterochromatin in early embryos and its role in early fruitfly development are unknown. Lysine 9 trimethylation of histone H3 (H3K9me3) and recruitment of its epigenetic reader, heterochromatin protein 1a (HP1a), are hallmarks of constitutive heterochromatin. Here, we show that H3K9me3 is transmitted from the maternal germline to the next generation. Maternally inherited H3K9me3, and the histone methyltransferases (HMT) depositing it, are required for the organization of constitutive heterochromatin: early embryos lacking H3K9 methylation display decondensation of pericentromeric regions, centromere-centromere de-clustering, mitotic defects, and nuclear shape irregularities, resulting in embryo lethality. Unexpectedly, quantitative CUT&Tag and 4D microscopy measurements of HP1a coupled with biophysical modeling revealed that H3K9me2/3 is largely dispensable for HP1a recruitment. Instead, the main function of H3K9me2/3 at this developmental stage is to drive HP1a clustering and subsequent heterochromatin compaction. Our results show that HP1a binding to constitutive heterochromatin in the absence of H3K9me2/3 is not sufficient to promote proper embryo development and heterochromatin formation. The loss of H3K9 HMTs and H3K9 methylation alters genome organization and hinders embryonic development.**

Keywords Constitutive Heterochromatin; Early Embryogenesis; H3K9 Methylation; Intergenerational Inheritance; HP1a Clusters
Subject Categories Chromatin, Transcription & Genomics; Development

## Introduction

Constitutive heterochromatin is a condensed part of the genome that is important for genome integrity and stability. It forms mainly in repeat-rich regions near the centromeres (Pericentromeric Heterochromatin or PCH) and telomeres and is enriched for di- and trimethylation of histone H3 Lysine 9 (H3K9me2/3). In *Drosophila melanogaster*, the histone methyltransferases (HMTs) dSetDB1, Su(var)3-9, and G9a deposit H3K9me2/3 (Bannister et al, 2001; Clough et al, 2007; Eissenberg and Elgin, 2014; Lachner et al, 2001; Marsano et al, 2019; Nakayama et al, 2001; Padeken et al, 2022; Rea et al, 2000; Schotta et al, 2002). In turn, HP1 proteins, which are evolutionarily conserved from fission yeast to humans, recognize and bind H3K9me2/3 via their chromodomains (Padeken et al, 2022). HP1 proteins bound to chromatin recruit H3K9 HMT Su(var)3-9, and this feedback loop mechanism leads to further compaction of the heterochromatin (Allshire and Madhani, 2018; Padeken et al, 2022).

The embryo undergoes substantial epigenetic reprogramming immediately after fertilization. In *D. melanogaster*, however, maternally transmitted H3K27me3 and H4K16ac escape reprogramming and are essential for proper embryo development (Samata et al, 2020; Zenk et al, 2017). Moreover, loss of H3K27me3 in *D. melanogaster* leads to altered long-distance chromatin contacts and it correlates with a loss of transgenerational heritable repression (Cavalli and Paro, 1998; Ciabrelli et al, 2017; Fitz-James et al, 2023). Similar to *D. melanogaster*, in mice maternal H3K27me3 is also intergenerationally inherited (Inoue et al, 2017). Instead, constitutive heterochromatin mark H3K9me2/3 has been shown to appear around cycles 12–13 during fly early embryogenesis and strongly colocalize with the pericentromeric regions at zygotic genome activation (ZGA)(cycle 14) (Fig. 1A) (Seller et al, 2019; Yuan and O'Farrell, 2016). However, in *Drosophila miranda*, H3K9me3 is already detectable at nineth mitotic divisions after fertilization (cycle 9 embryos) (Wei et al, 2021). In the early mouse zygote, H3K9me3 is already detectable at the maternal pronucleus, and H3K9me3-enriched regions in the early zygotes correlate with those in the oocyte (Arney et al, 2002; Santos et al, 2005; Wang et al, 2018). However, whether H3K9me3 is maternally

[1]Max Planck Institute of Immunobiology and Epigenetics, 79108 Freiburg im Breisgau, Germany. [2]Albert-Ludwigs-Universität Freiburg, Fahnenbergplatz, 79085 Freiburg im Breisgau, Germany. [3]Karlsruhe Institute of Technology (KIT), Zoological Institute, 76131 Karlsruhe, Germany. [4]Brain Mind Institute, School of Life Sciences EPFL, SV3809, 1015 Lausanne, Switzerland. [5]San Raffaele Telethon Institute for Gene Therapy, IRCCS San Raffaele Scientific Institute, 20132 Milan, Italy. [6]Centre for Biological signaling studies, University of Freiburg, 79104 Freiburg im Breisgau, Germany. [7]Institute Pasteur, Mechanisms of Epigenetic Inheritance, Department of Developmental and Stem Cell Biology, UMR3738, CNRS, 75724, Cedex 15 Paris, France. [8]Università degli Studi di Milano and INFN, Milan, Italy. [9]Department of Experimental Oncology, European Institute of Oncology-IRCCS, Milan, Italy. ✉E-mail: yinxiu.zhan@ieo.it; iovino@ie-freiburg.mpg.de

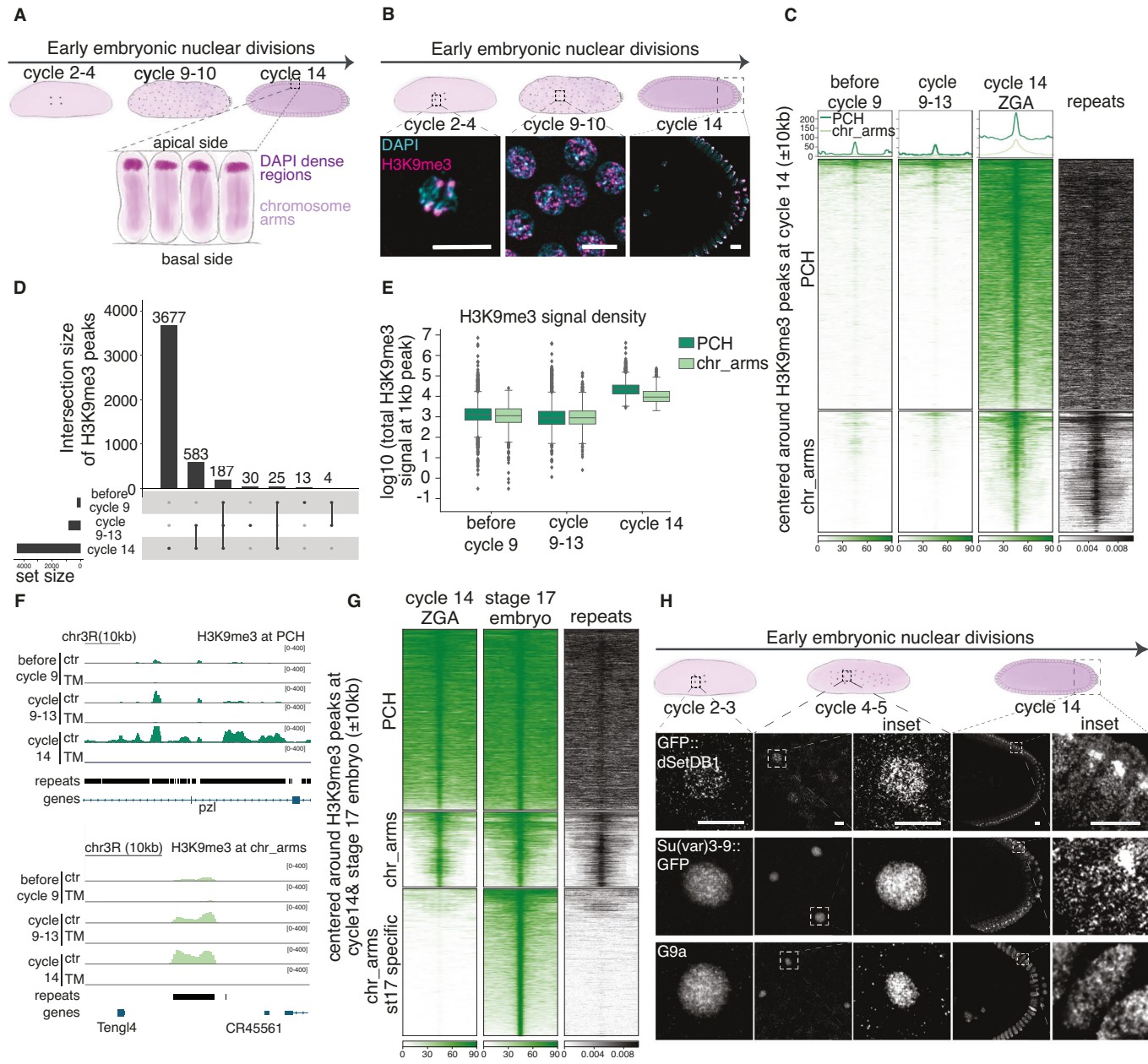

inherited or deposited de novo after fertilization is not functionally shown.

The early embryonic development immediately after fertilization until ZGA relies mainly on maternally loaded RNAs and proteins. Hence, to study the function of specific histone modifications before and at ZGA, the maternal stock of enzymes that is loaded into the embryo has to be depleted. In a nematode, *Caenorhabditis elegans*, eliminating the two H3K9 HMTs leads to a complete loss of H3K9 methylation. However, embryos develop into adults with some mutant worms showing a stochastic developmental delay and reduced fertility (Towbin et al, 2012; Zeller et al, 2016). In *D. melanogaster*, both Su(var)3-9 and G9a knockout flies are viable and fertile, whereas dSetDB1 mutants are sterile (Clough et al, 2007; Schotta et al, 2002; Seum et al, 2007; Stabell et al, 2006).

Replacing H3K9 with unmodifiable H3K9R in *Drosophila* leads to decreased HP1 binding to chromocenters of the polytene chromosomes in somatic salivary gland cells (Penke et al, 2016; Penke et al, 2018). Higher organisms like mice have at least six H3K9 HMTs SUV39H1/H2, SETDB1/SETDB2 and G9A/GLP. In mice, the early zygotic and after ZGA function of H3K9 methylation was studied by depleting single or few H3K9 methyltransferases simultaneously as they are expressed at varying levels, and they are essential for the proper development of embryos (Dodge et al, 2004; Eymery et al, 2016; Kim et al, 2016; Padeken et al, 2022; Peters et al, 2001; Tachibana et al, 2002). However, the comprehensive depletion of all six H3K9 HMTs during late oogenesis and in early mouse embryos is a complex challenge, which limits the understanding the full scope of the role of H3K9me3 from fertilization until ZGA. The

**Figure 1. H3K9me3 is present from fertilization onward in early *Drosophila* embryos.**

(A) A schematic illustration of early embryonic development of *Drosophila* embryos. After fertilization, the embryo undergoes 13 nuclei divisions after which at cycle 14, nuclei get cellularized and transcriptionally fully activate their genome. The chromatin is in a Rabl configuration, where DAPI-dense regions located apically are H3K9 methylated. (B) Top panel: a schematic illustration of early embryonic development of *Drosophila* embryos. Bottom panel: antibody for H3K9me3 and 4′,6-diaminido-2 phenylindole (DAPI) staining of early embryos at different developmental cycles. For clarity, mitotic chromosomes for some cycles are shown. Scale bars, 10 µm. (C) Heatmaps of H3K9me3 at different developmental stages of early embryogenesis: before cycle 9, cycle 9–13 and ZGA (cycle 14) clustered into pericentromeric regions (PCH) and chromosome arms (chr_arms). The last column represents the density of repeats at H3K9me3 cycle 14 peaks. The profile plots are shown for the corresponding stages above the heatmaps. The y-axis is the same for all three profile plots from 0 to 200. The signal is ±10 kb centered on H3K9me3 peaks at cycle 14 embryos and ranked by signal intensity at cycle 14. The mean signal of two replicates is shown. (D) Upset plot showing the number of H3K9me3 peaks specific for each stage and common for different developmental stages. (E) Boxplot of H3K9me3 signal density distribution at 1 kb H3K9me3 peak at PCH and chr_arms at before cycle 9, cycle 9–13, and cycle 14 embryos. The boxplot indicates interquartile range from 1st (Q1) and 3rd (Q3) quartile, whiskers denote 1.5 times interquartile region (IQR) below Q1 and above Q3. Dots represent outliers. (F) H3K9me3 CUT&Tag signal tracks at PCH and chr_arms for control (ctr) and triple mutant (TM) that is devoid of H3K9me3. (G) Heatmaps of H3K9me3 at ZGA (cycle 14) and stage 17 embryos clustered into PCH, chr_arms, and H3K9me3 signal specific for stage 17 embryos. The last column represents the density of repeats at H3K9me3 cycle 14 and stage 17 peaks. The signal is ±10 kb centered on H3K9me3 peaks at cycle 14 and stage 17 embryos and ranked by H3K9me3 signal intensity at cycle 14 and stage 17 embryos. The mean signal of two replicates is shown. (H) Antibody for GFP for endogenously GFP tagged dSetDB1 and Su(var)3-9, and G9a staining with G9a antibody of early embryos at different developmental cycles. Insets on the right of cycle 4–5 and cycle 14 embryos show a magnification of the signals at cycle 4–5 and cycle 14 embryos, respectively. *n* is three embryos per genotype and per stage. Scale bars, 10 µm. Source data are available online for this figure.

significance of H3K9 methylation and its HMTs during the early stages of embryogenesis in flies remains unclear, particularly whether their role is as crucial for proper embryonic development as it is in mice, or if they are not necessary as seen in *C. elegans* (Dodge et al, 2004; Eymery et al, 2016; Kim et al, 2016; Padeken et al, 2022; Padeken et al, 2019; Peters et al, 2001; Tachibana et al, 2002; Zeller et al, 2016).

Constitutive heterochromatin can be visualized at ZGA when HP1a is enriched at the pericentromeric regions (Larson et al, 2017; Seller et al, 2019; Yuan and O'Farrell, 2016; Zenk et al, 2021). HP1 can form liquid-like condensates in vitro and in vivo, which might drive the heterochromatin formation (Larson et al, 2017; Sanulli and Narlikar (2020); Strom et al, 2017). In this study, we depleted all three H3K9 HMTs in *D. melanogaster* early embryos to study H3K9me3 function in the formation of constitutive heterochromatin from fertilization until ZGA. In addition, we explored whether the underlying histone methylation plays a role in HP1a foci formation in vivo.

Here by depleting all H3K9 HMTs, we show that H3K9me3 in *D. melanogaster* is inherited from the oocyte to the zygote, and it is actively maintained throughout early embryonic divisions. H3K9me3 reaches its full genomic occupancy during ZGA. Unlike in mouse early embryos (Burton et al, 2020), where Suvar39h is important for H3K9me3 establishment at pericentromeric regions, dSetDB1 is the main H3K9me3 HMT in early fly embryos concordant with (Seller et al, 2019). We also disentangled the function of dSetDB1 in early embryogenesis from the one in early oogenesis by depleting dSetDB1 only in later stages of oogenesis. Our results show that the depletion of individual H3K9 HMTs does not lead to substantial embryonic death; only upon the removal of all the H3K9 HMTs, we observed strong phenotypes. In the absence of H3K9 methylation and its HMTs, embryos have defects in chromosomal segregation, pericentromeric compaction, repression of repetitive elements and nuclear shape. We also show that embryonic developmental defects can be rescued by dSetDB1 alone but not with a catalytically inactive dSetDB1. Finally, we show that HP1a binding to chromatin at ZGA is not completely dependent on H3K9me2/3. Instead, HP1a cluster formation is dependent on H3K9me2/3.

# Results

## H3K9me2/3 is present on chromatin from fertilization and throughout early embryogenesis

We monitored the distribution and amount of H3K9 methylation during early embryogenesis in *Drosophila* by immunofluorescence (IF). We found that H3K9me3 and H3K9me2 were present from fertilization throughout early embryo development (Fig. 1B and Fig. EV1A). To characterize the distribution of H3K9me3 in early embryos, we performed quantitative CUT&Tag in hand-sorted before cycle 9, cycle 9–13, and cycle 14 (ZGA) embryos (Tadros and Lipshitz, 2009). H3K9me3 was already detected in embryos before cycle 9, consistent with our IF data (Fig. 1C,D and "Methods"). The H3K9me3 mainly localized at repeat-rich regions (Fig. 1C) and was specific, as the signal was absent in the mutant embryos devoid of H3K9me3 (Fig. EV1B). H3K9me2 had a similar distribution to H3K9me3 (Fig. EV1C).

The number of H3K9me2/3 peaks increased gradually from before cycle 9 to cycle-9–13 embryos and then dramatically increased at ZGA (Figs. 1D and EV1D). H3K9me3 was enriched in around 230 transposable elements (TEs) in embryos before cycle 9 and around 790 TEs gained H3K9me3 de novo, during cycles 9–13 (Fig. EV1E; Dataset EV1). As the embryo developed from before cycle 9 to ZGA, the H3K9me2/3 signal density and peak size increased, especially in the PCH but also on the chromosomal arms (Figs. 1E,F and EV1F,G). In addition, our quantitative CUT&Tag data revealed that PCH regions of late-stage embryos (stage 17) had similar H3K9me3 distribution and levels compared to ZGA embryos, suggesting that H3K9me3 is fully established at PCH regions in the first two hours of development, by the time of ZGA (Fig. 1G). Interestingly, some repeat-free regions on the chromosomal arms acquired de novo H3K9me3, specifically in late-stage embryos, compared to ZGA (Fig. 1G, st17-specific cluster, bottom), suggesting that H3K9me3 could play a role in tissue-specific gene regulation.

The *D. melanogaster* genome encodes three H3K9 HMTs, Su(var)3-9, dSetDB1/Eggless, and G9a, among which only dSetDB1 is essential for fertility (Clough et al, 2007; Schotta et al, 2002; Seum et al, 2007; Stabell et al, 2006). To investigate the

distribution of H3K9 HMTs during early embryogenesis, we endogenously tagged both dSetDB1 and Su(var)3-9 with GFP (both homozygous viable) and examined the distribution of three H3K9 HMTs by IF. All the three H3K9 HMTs were present in nuclei of control embryos from fertilization until cycle 14 (ZGA) (Figs. 1H and EV1H). By ZGA, both Su(var)3-9 and dSetDB1 were mainly enriched in DAPI-dense regions, whereas G9a was distributed along the chromosome arms (Figs. 1H and EV1H). Overall, the three enzymes are present throughout early development, similar to H3K9me2/3.

## H3K9me3 is inherited from the maternal germline and actively propagated in early embryos

To investigate if the three H3K9 HMTs are required to propagate H3K9me3 from the maternal germline to the next generation early embryos (F1), we generated triple mutant (TM) embryos by combining a knockout (KO) of Su(var)3-9 with short hairpin RNA (shRNA)-mediated knockdown (KD) of dSetDB1 and G9a at late stages of oogenesis (Fig. 2A).

We confirmed the lack of Su(var)3-9, dSetDB1, and G9a proteins in TM ovaries compared to the control in late oocytes and early embryos by mass spectrometry, RT-qPCR and Western blot (WB) (Dataset EV2; Fig. EV2A'–D'). To study the importance of H3K9 methylation in embryo development, we investigated the hatching rate of TM embryos. Notably, the hatching rate of TM embryos was strongly compromised compared to controls, with only around 30% of embryos completing embryogenesis, on average (Fig. 2B). In contrast, single Su(var)3-9-KO, dSetDB1-KD, and G9a-KD mutant embryos did not display severe defects in embryogenesis, suggesting redundancy among the H3K9 HMTs (Fig. EV2E).

Using IF, we found that H3K9me3 was still enriched in the oocyte of both control and TM ovaries (Fig. 2C P0 oocyte, arrowhead), in line with the fact that the oocyte, which is arrested in meiosis I and therefore noncycling (Von Stetina and Orr-Weaver, 2011), does not require H3K9 HMT activity to retain H3K9 methylation on chromatin. When we stained the pronuclei at apposition, H3K9me3 was enriched in the maternal pronucleus (defined by the presence of H3K27me3) of the control zygote

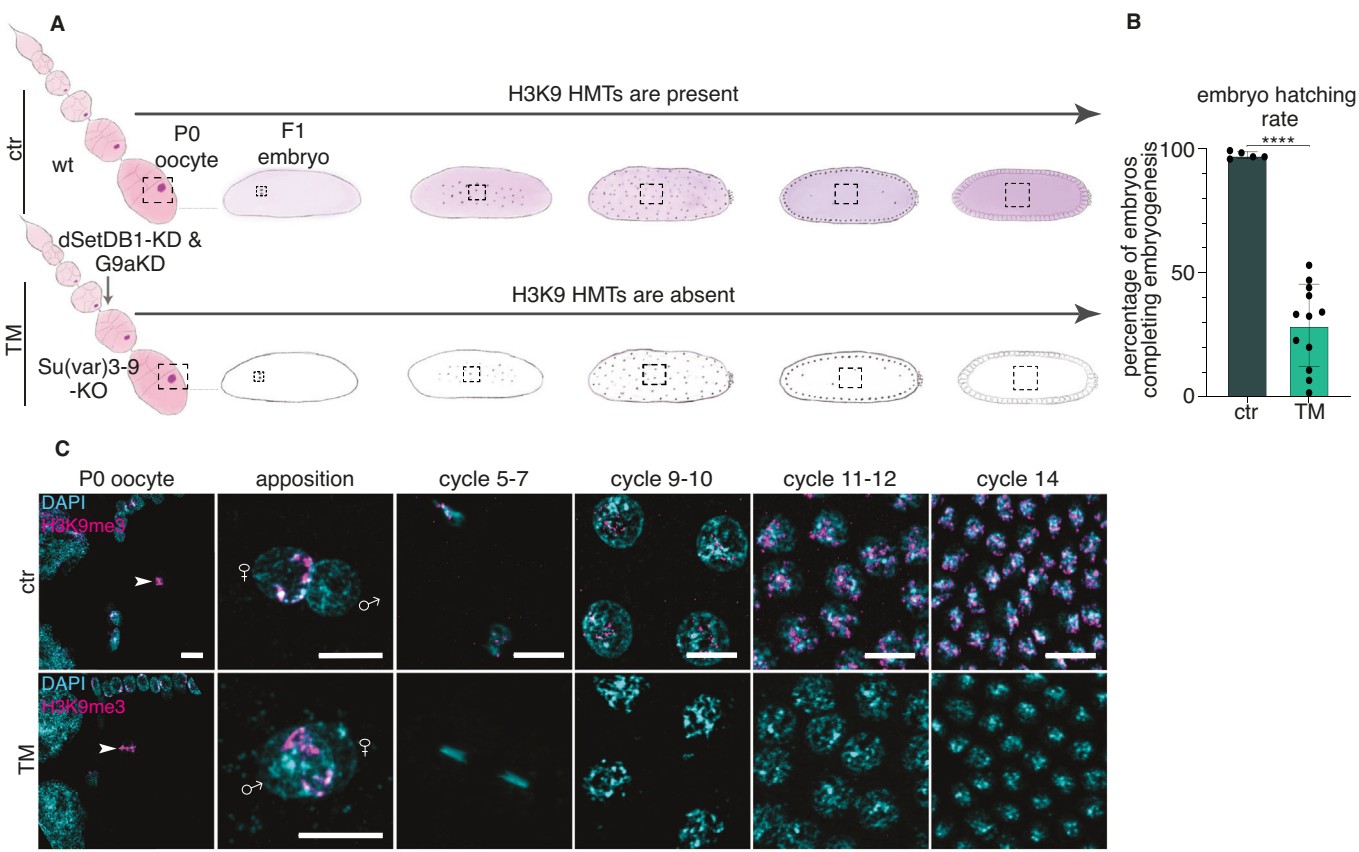

**Figure 2. H3K9me3 is intergenerationally inherited from the oocyte to the embryo.**

(A) A schematic illustration of an ovariole in the germline of *Drosophila* and of the embryo at apposition of maternal and paternal pronuclei and embryos with several early embryonic nuclei divisions. The arrow indicates the germline knockdown (KD) of dSetDB1 and G9a in Su(var)3-9 knockout (KO) background flies that we call triple mutant (TM). (B) Hatching rate of ctr and TM embryos (P value < 0.0001, two-tailed Welch two-sample t test). n = 5 biological replicates with a total of 600 embryos were used for ctr and n = 12 biological replicates with total 1440 embryos were used for TM embryos. Each dot represents one biological replicate. Shown mean +/− s.d. (C) H3K9me3 and DAPI staining of ctr and TM stage 10 egg chamber oocyte and staining of early embryos at different developmental cycles. The arrowhead indicates the oocyte. To define the maternal pronucleus, H3K27me3 was used as it stains the maternal pronucleus (Zenk et al, 2017). For clarity, mitotic chromosomes for most cycles are shown. n is at least three embryos per genotype and per stage. Scale bars, 10 µm. Source data are available online for this figure.

(Figs. 2C and EV2F) (Zenk et al, 2017). Surprisingly and importantly, H3K9me3 was also detected in the maternal pronucleus of the TM zygote (Figs. 2C and EV2F), despite the absence of all three H3K9 HMTs, formally demonstrating the inheritance of this modification. However, after a few nuclear divisions, H3K9me3 was lost from TM embryos but not in control embryos (Fig. 2C), suggesting that the HMTs activity in the early embryo is required for the active propagation of the mark from fertilization onwards. Instead, H3K9ac that is found in the oocyte is reprogrammed in the fly early embryos as we did not detect H3K9ac in the early stage (cycle 6–7) embryos (Fig. EV2G) but it appears around cycle 8 (Ciabrelli et al, 2023). Hence, our results show that H3K9me3 is transmitted from the oocyte to the zygote and that the maternally supplied H3K9 HMTs are required for its propagation throughout early embryogenesis.

## Depletion of H3K9me2/3 results in mitotic defects, disorganized heterochromatin, and deformed nuclear shape

In *Schizosaccharomyces pombe* depletion of Clr4 H3K9 HMT leads to an increase in the fraction of cells with lagging chromosomes (Ekwall et al, 1996). Primary mouse embryonic fibroblasts (PMEF) derived from *Suvar39h* double knockout (dn) mice and B-cell lymphomas formed in *Suvar39h* dn mice show a polyploid DNA content (Peters et al, 2001), and mouse oocytes devoid of SETDB1 show misaligned chromosomes and multiple spindles (Eymery et al, 2016; Kim et al, 2016). However, *C. elegans* embryos that lack both H3K9 HMTs and, consequently, H3K9 methylation do not show any mitotic defects (Zeller et al, 2016). To investigate whether fly embryos that lack H3K9 methylation and its HMTs show mitotic defects, we performed live imaging of cycle 13 nuclear division in control and TM embryos that carry H2Av-RFP. Unlike *S. pombe* lacking Clr4 (Ekwall et al, 1996), nuclei in cycle 13 TM embryos did not have a significant increase in lagging chromosomes, as well as almost no change in fiber formation during anaphase (Fig. 3A). Moreover, there was no significant increase in the fraction of nuclei with micronuclei in TM compared to the control (Fig. 3A). However, TM embryos exhibited a significantly increased fraction of nuclei that formed bridges and multipolar spindles, failed to condense during metaphase and arrested during the nuclear division, as well as a significant increase in the fraction of nuclei that fused together (Fig. 3A). These observations are in line with the presence of polyploid cells in *Suvar39h* double-null PMEFs (Peters et al, 2001). Finally, the number of apoptotic nuclei was much higher at cycle 14 in TM embryos than in control embryos (Fig. 3A). These results show the importance of H3K9me2/3 in promoting the proper nuclear division by preventing the metaphase arrest, favoring the condensation of chromatin during metaphase and preventing nuclei fusions and polyploidy in early embryogenesis. These results show that there are species-specific differences in terms of the importance of H3K9 methylation and its HMTs in proper mitosis: *D. melanogaster* is more like a mouse than *C. elegans*. Furthermore, embryos lacking H3K9me2/3 exhibited a prolonged period from fertilization to gastrulation compared to control embryos similar to *C. elegans* (Fig. 3B) (Padeken et al, 2019; Zeller et al, 2016).

To investigate how site-specific loss of H3K9me2/3 affects transcription, we performed Global Run-On sequencing (Gro-seq)

on hand-staged ZGA control and TM embryos. Only a few genes were misregulated in TM embryos compared to control embryos, similar to results seen in *C. elegans* and mouse embryos (Burton et al, 2020; Zeller et al, 2016), and these genes were not involved in the formation or maintenance of constitutive heterochromatin (Fig. EV3A,B; Dataset EV3). Importantly, control and TM embryos showed similar expression of genes that become activated at ZGA (Fig. EV3C). In contrast, repetitive elements, including telomeric repeats like HETA, TART, and TAHRE and simple repeats, were upregulated in TM embryos (Figs. EV3B and EV3D). It is important to note that not all members of the TE family were transcriptionally misregulated upon loss of H3K9me2/3 (Dataset EV3), implying the presence of other mechanisms that regulate repression of TEs in ZGA embryos. This is similar to *C. elegans* where only around one-third of repeat subfamilies showed more than 1.5-fold upregulation in the absence of H3K9 methylation (Padeken et al, 2019; Zeller et al, 2016) as well as mouse embryos (Burton et al, 2020). We acknowledge that our study does not cover all satellite repeats because of an incomplete assembly of the fly genome at pericentromeric regions (Hoskins et al, 2015). Therefore, it will be interesting to study the effect of H3K9me2/3 absence on those regions in the future as the repetitive regions of *D. melanogaster genome* get assembled fully. Nevertheless, using RNA-FISH and RT-qPCR, we confirmed that satellite III (SATIII), a repeat region found primarily on chromosome X, was significantly upregulated in TM embryos compared to control embryos (Figs. 3C and EV3E). Notably, RT-qPCR in dSetDB1-KD and Su(var)3-9-KO single mutants did not show increased levels of SATIII RNA (Fig. EV3E), suggesting that the misregulation of SATIII is specific to TM embryos, and due to a complete loss of H3K9me2/3. We examined a DNA damage by detecting phosphorylated H2A.Z (phospho-H2A.Z), but did not see a significant increase in the phospho-H2A.Z signal in TM versus control embryos (Fig. EV3F), implying that DNA damage does not accumulate at ZGA in embryos lacking H3K9me2/3. However, we note that our failure to observe a higher percentage of nuclei with the elevated phosphorylated H2A.Z signal in TM embryos may be attributed to the increased nuclei death at cycle 14, leading to their disposal into the central yolk of the embryo.

We observed that the complete loss of H3K9 methylation in TM embryos has a significant impact on heterochromatin organization. We noticed that the volume of DAPI-dense regions decreased significantly in TM cycle 14 embryos compared to control cycle 14 embryos (Fig. 3D). Moreover, we used IF to detect the centromere-specific H3 variant CENP-A (CID in *D. melanogaster*) and observed centromere disorganization and de-clustering in TM embryos at ZGA (6–7 CID foci on average) compared to control ZGA embryos (3-4 CID foci) (Fig. 3E and Fig. EV3G). DNA-FISH revealed a significant decompaction of SATIII DNA in TM compared to control ZGA embryos (Fig. 3F). Finally, to investigate the H3K9me2/3 effect on a nuclear shape, we stained control and TM ZGA embryos with Lamin dm0. We found that nuclear sphericity and roundedness were significantly lower in some TM embryos compared to control embryos (Figs. 3G and EV3H), suggesting that H3K9me2/3 is important for establishing or/and maintaining the proper nuclear shape in early embryos. Overall, our findings strongly indicate that the inheritance and propagation of H3K9me2/3 modifications play a pivotal role in ensuring the integrity of the nuclear division by preventing nuclear fusion and

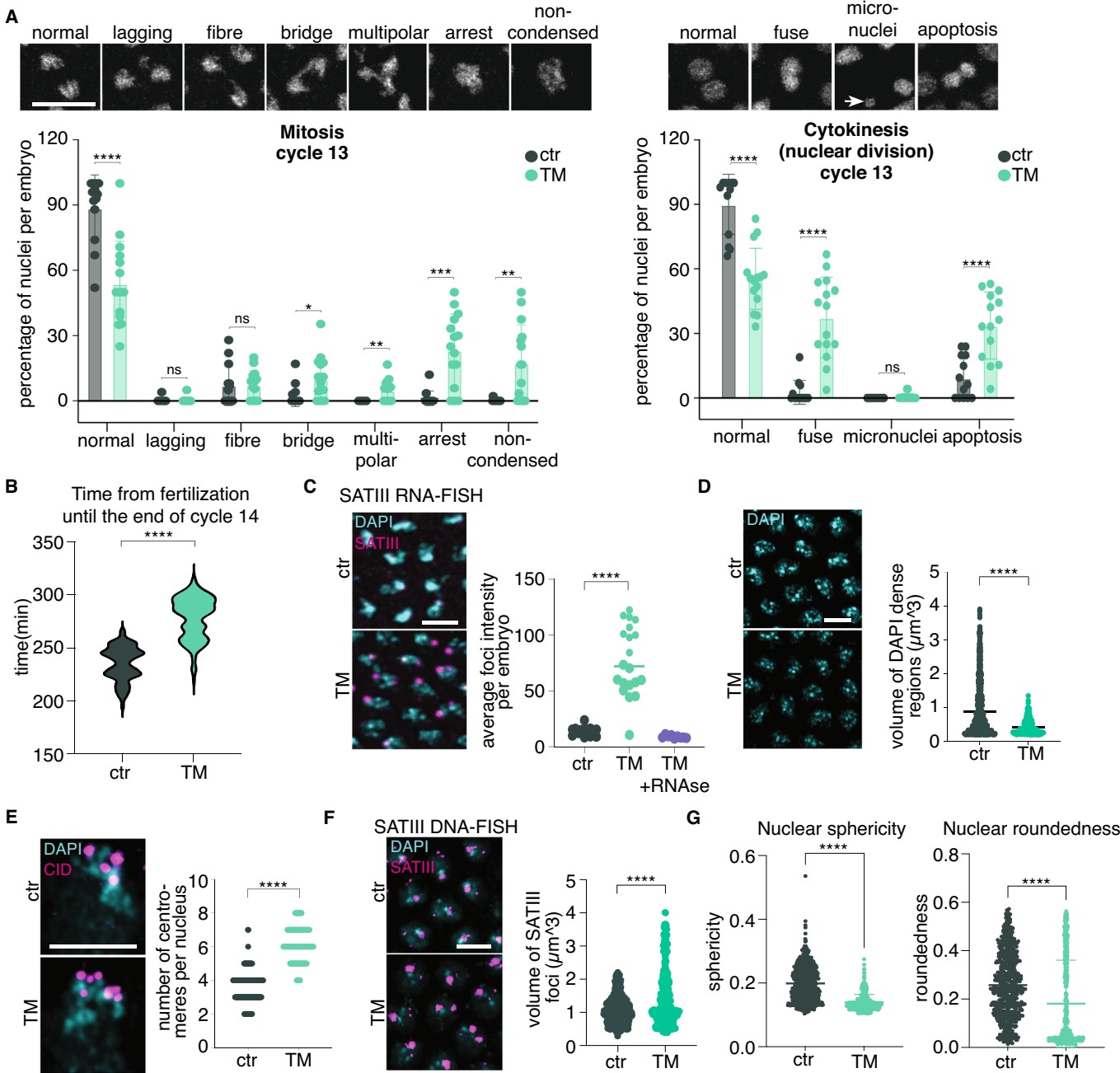

**Figure 3. Loss of H3K9 methylation leads to defective constitutive heterochromatin establishment.**

(A) Top panel: snapshots of images as examples of the mitotic and cytokinesis (nuclear division) defects at division 13 (from cycle 13 to cycle 14). An arrow points to the micronucleus. Scale bars, 10 μm. Bottom panel: quantification of the percentage of nuclei with mitotic defects during mitosis and cytokinesis in ctr and TM embryos. $n = 2$ biological replicates with 13 embryos for ctr and 15 embryos for TM were used for quantification. (B) Violin plot showing the time it takes for ctr and TM embryos from fertilization until the end of cycle 14 (P value < 0.0001, two-tailed Welch two sample t-test). $n = 3$ biological replicates with total 177 embryos for ctr and 139 embryos TM were used for quantification. (C) Left panel: RNA-FISH of SATIII repeat at cycle 14 ctr and TM embryos. Scale bars, 5 μm. Right panel: quantification of average SATIII foci intensity per embryo in ctrl, TM and TM treated with RNAse cycle 14 embryos (P value < 0.0001, two-tailed Welch two-sample t test). $n = 19$–20 embryos for ctr and TM and $n = 9$ embryos for RNase treated TM were used for quantification. (D) Left panel: DAPI staining of cycle 14 ctr and TM embryos. Scale bars, 5 μm. Right panel: quantification of volume of DAPI-dense regions in ctr and TM cycle 14 embryos (P value < 0.0001, two-tailed Welch two-sample t test). $n = 7$–8 embryos were used for quantification for each genotype. (E) Left panel: antibody for CID and DAPI staining of cycle 14 ctr and TM embryos. Scale bars, 5 μm. Right panel: quantification of number of CID foci in ctr and TM cycle 14 embryos (P value < 0.0001, two-tailed Welch two-sample t test). $n = 3$–4 embryos were used for quantification for each genotype. (F) Left panel: DNA-FISH of SATIII repeat at cycle 14 ctr and TM embryos. Scale bars, 5 μm. Right panel: quantification of volume of SATIII foci in ctr and TM embryos (P value < 0.0001, two-tailed Welch two-sample t test). $n = 3$ biological replicates with 12 embryos were used for quantification for each genotype. (G) Left panel: quantification of nuclear sphericity of cycle 14 ctr and TM embryos. Right panel: quantification of nuclear roundedness of cycle 14 ctr and TM embryos (P value < 0.0001, two-tailed Welch two-sample t test). $n = 2$ biological replicates with 9 embryos were used for quantification for each genotype. Source data are available online for this figure.

cell death. Moreover, these modifications are crucial for compacting and repressing some repetitive genomic regions, facilitating the assembly of apical chromocenter clusters, ultimately, establishing constitutive heterochromatin effectively following the fertilization.

## Single-histone methyltransferase mutants are not sufficient to deplete H3K9me2/3 signals

Our quantitative CUT&Tag confirmed the complete reduction of H3K9me2/3 (Figs. 4A,B and EV4A,B) in cycle 14 TM embryos. dSetDB1 is essential for early oogenesis and keeping the female sex identity; therefore, we had to conditionally knock it down by a short hairpin RNA (shRNA) at the late stage of oogenesis to overcome the sterility (Clough et al, 2007; Clough et al, 2014; Smolko et al, 2018). When dSetDB1 is knocked down in the early oogenesis, it leads to the sterility of the fly (Clough et al, 2007). Therefore, we used an early oogenesis driver (see "Methods") to check the specificity of dSetDB1 shRNA, which led to the sterility of flies (Fig. EV4C,D). However, the sterility of the flies was rescued upon restoring the dSetDB1 mRNA expression by introducing a short hairpin-resistant transgene (Fig. EV4C,D). After confirming

the shRNA specificity, we sought to elucidate the functions of each H3K9 HMTs. We performed IF for H3K9me2/3 in Su(var)3-9-KO, G9a-KO, and dSetDB1-KD (a late oogenesis driver) single mutant embryos at ZGA. Our results showed that H3K9me3 levels remained unchanged in G9a-KO embryos, exhibited a minor reduction in Su(var)3-9-KO and a substantial reduction in dSetDB1-KD ZGA embryos (Fig. EV4E). All three enzymes were important for depositing H3K9me2 as H3K9me2 levels decreased in all three H3K9 HMT single mutant embryos (Fig. EV4F). Similar changes were observed by quantitative CUT&Tag for H3K9me2/3 levels at ZGA (Fig. EV4G,H). These data confirm that dSetDB1 is the main H3K9me3 HMT in the early *Drosophila* embryo in line with (Seller et al, 2019).

## HP1a binding in early embryos is independent of H3K9me2/3

The primary readers of the H3K9me2/3 modification are HP1 proteins. The *D. melanogaster* genome encodes five HP1 paralogs, HP1a being the main paralog that binds to the heterochromatin (Vermaak and Malik, 2009; Zenk et al, 2021). HP1a contains an

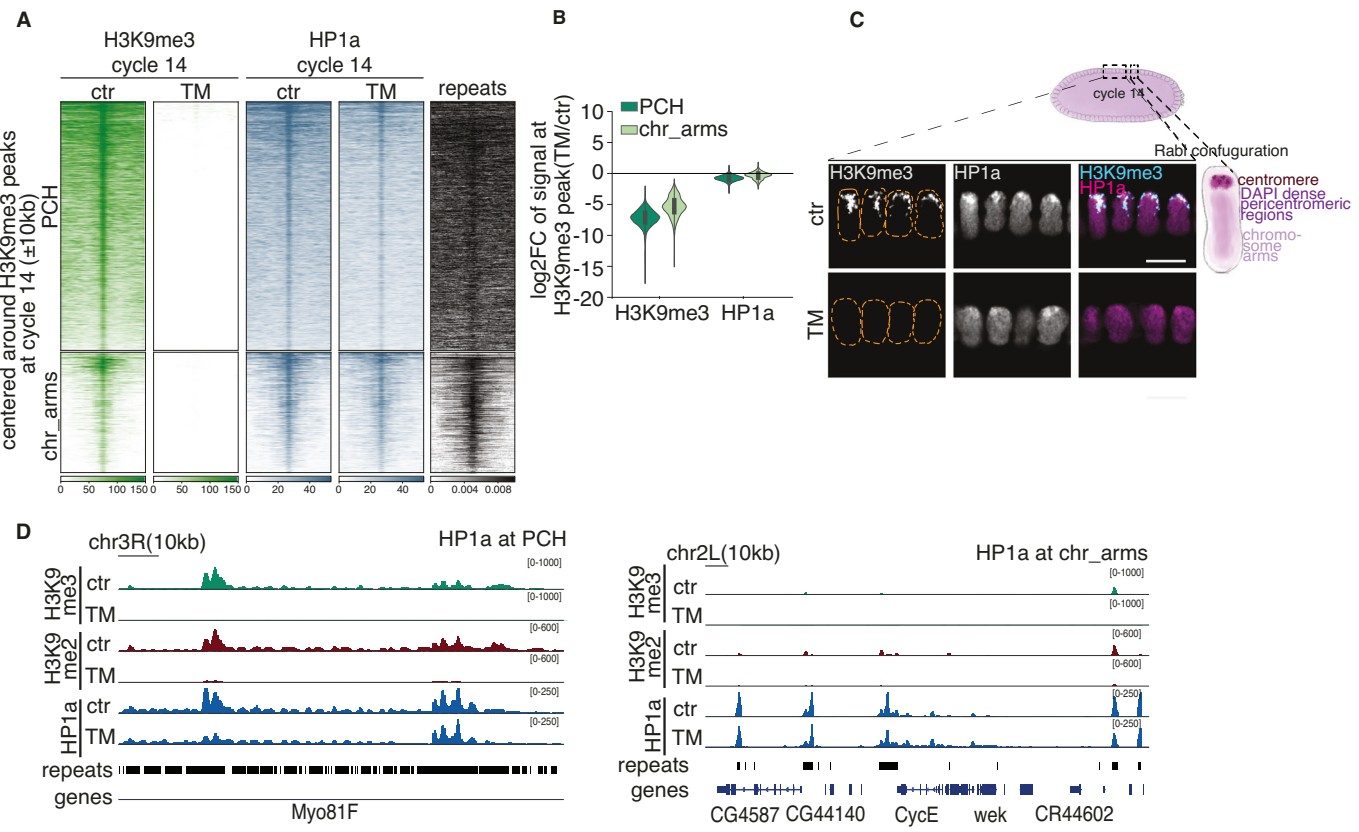

**Figure 4. HP1a binding to chromatin is not completely dependent on H3K9 methylation.**

(A) Heatmaps of H3K9me3 and HP1a at cycle 14 ctr and TM embryos clustered into PCH and chr_arms. The last column represents the density of repeats at H3K9me3 cycle 14 peaks. The signal is ±10 kb centered on H3K9me3 peaks at cycle 14 ctr embryos and ranked by signal intensity at cycle 14 ctr embryos. Mean signal of two biological replicates is shown. (B) Violin plot of log2 fold change of H3K9me3 and HP1a signal on H3K9me3 peaks found in PCH and chr_arms regions in TM cycle 14 embryos compared to ctr embryos. The boxplot inside indicates interquartile range from 1st (Q1) and 3rd (Q3) quartile, whiskers denote 1.5 times interquartile region (IQR) below Q1 and above Q3. (C) Right panel: a schematic illustration of cycle 14 embryo with a nucleus with a Rabl configuration. Left panel: H3K9me3 and HP1a staining of ctr and TM cycle 14 embryos. The yellow dotted lines are drawn to represent the nucleus. *n* is at least five embryos per genotype. Scale bars, 10 µm. (D) H3K9me2/3 and HP1a CUT&Tag signal tracks at PCH (left) and chr_arms (right) for ctr and TM cycle 14 embryos. Source data are available online for this figure.

N-terminal chromodomain that binds specifically to H3K9me2/3 histone modification (Bannister et al, 2001; Jacobs et al, 2001; Lachner et al, 2001). It also contains a C-terminal chromo shadow domain (CSD) that can bind to proteins, including H3K9 methyltransferase Su(var)3-9, as well as forming a homodimer (Cowieson et al, 2000; Schotta et al, 2002). HP1a dependence on H3K9me2/3 for binding to chromatin has been primarily shown in differentiated somatic cells (Eissenberg and Elgin, 2014; Padeken et al, 2022). However, it remains unknown whether HP1a binding to heterochromatin also depends on H3K9me2/3 in undifferentiated early embryos nuclei. To address this question, we examined HP1a distribution on chromatin during early embryogenesis by performing quantitative CUT&Tag on hand-staged ZGA embryos. In control ZGA embryos, HP1a was bound to H3K9me3-enriched regions in PCH and chromosome arms (Fig. 4A) as well as to chromosome arm regions that lack repeats and H3K9me3 (Fig. EV4I), as previously observed (James et al, 1989; Zenk et al, 2021). Unexpectedly, HP1a binding was reduced in TM embryos albeit not completely (on average HP1a binding was reduced by 35% on PCH regions, Figs. 4A'–D' and EV4J). Moreover, HP1a binding to regions on chromosome arms slightly increased in TM embryos, consistent with previous studies (Fig. EV4I,J) (Penke et al, 2016; Zenk et al, 2021). Importantly, HP1a protein levels were the same in control and TM ZGA embryos (Fig. EV4K). HP1a enrichment at PCH and chromosome arm regions was slightly changed in single mutant dSetDB1-KD and Su(var)3-9-KO ZGA embryos compared to the control (Fig. EV4E,J). Overall, these results show that, in early embryonic development, HP1a can bind PCH in a H3K9me2/3-independent manner and in the absence of the H3K9 HMTs.

## H3K9me3 before and at ZGA is crucial for embryonic development

To investigate when H3K9 HMT activity is required during development, we restored the expression of dSetDB1 in the TM embryos at distinct developmental stages. We restored the expression of dSetDB1 in the maternal germline (maternal rescue, MR) and in the embryo from ZGA onwards (zygotic rescue, ZR). We found that dSetDB1 expression in both ZR and MR restored H3K9me3 levels in late (stage 17) TM embryos (Fig. EV5A). Moreover, H3K9me3 was enriched in the same regions in TM + MR, TM + ZR, and control stage 17 embryos by CUT&Tag (Fig. EV5B). However, to our surprise, the hatching rate was restored in TM + MR embryos but not in TM + ZR embryos, despite the similar rescued H3K9me3 levels in late embryos (Fig. 5A). This finding strongly suggests that H3K9me3 accumulation is required during early embryogenesis, before and at ZGA.

To confirm the requirement of H3K9me3 modification during early embryogenesis, we generated endogenously flag-HA tagged catalytic dead dSetDB1 (Y1236N) flies. We restored the expression of catalytic dead dSetDB1 in the maternal germline (maternal rescue catalytic dead, TM + MR_Cat.dead) in TM embryos (Fig. EV5C). As a control, we restored the expression of endogenously flag-HA tagged wild-type dSetDB1 in TM embryos (TM + MR). WB analysis showed that H3K9me3 levels are partially restored in control ZGA embryos (Fig. EV5C). In contrast, H3K9me3 is not restored in TM rescued by catalytically inactive dSetDB1(TM + MR_Cat.dead), confirming that our dSetDB1 is

catalytically inactive (Fig. EV5C, the amount of wild-type and catalytic dead dSetDB1 protein is similar). Importantly, the hatching rate was restored in TM + MR embryos but not in TM + MR_Cat.dead embryos (Fig. 5A). Thus, catalytically inactive dSetDB1 does not rescue embryogenesis in TM embryos (Fig. 5A). Moreover, to rule out the possibility that catalytic dead dSetDB1 does not rescue the phenotype because of the compromised binding of dSetDB1 to its binding partners, we have performed affinity purification followed by mass spectrometry (IP-MS) analysis on wild-type endogenously tagged flag-HA-dSetDB1 and catalytic dead endogenously tagged flag-HA-dSetDB1 early embryos (Fig. EV5D). In flies, dSetDB1 binds Windei (Wde), a homolog of human activating transcription factor 7-interacting protein (ATF7IP), mouse mCAF1, and C.elegans LIN-65 (Fujita et al, 2003; Koch et al, 2009; Mutlu et al, 2018; Tsusaka et al, 2019). Our results showed that both wild-type and catalytic dead dSetDB1 bind to Wde (Fig. EV5D). Moreover, our results revealed that both wild-type and catalytic dead dSetDB1 bind a protein encoded by an uncharacterized gene CG14464, which is homologous to C. elegans protein ARLE-14 that has been shown to bind to MET-2, C. elegans homolog of dSetDB1 (Fig. EV5D) (Mutlu et al, 2018). Overall, these results show that H3K9me3 dependent on the catalytic activity of dSetDB1 is crucial for early embryogenesis.

## H3K9me2/3 is required to regulate heterochromatin organization and compaction

HP1a has been shown to form a liquid-like condensate in vitro and in vivo (Larson et al, 2017; Strom et al, 2017). During D. melanogaster early embryogenesis, small HP1a clusters are first visible in cycles 10–11 interphase and grow slightly in volume during cycles 12–13, ultimately becoming mature PCH condensates during the extended ZGA stage interphase (cycle 14) (Larson et al, 2017). To analyze the role of H3K9me2/3 on HP1a cluster formation, we endogenously tagged HP1a with GFP (homozygous viable) and performed 4D high-resolution microscopy on control and TM ZGA embryos (Fig. 5B). The levels of HP1a mRNA and protein were similar between control and mutant embryos (Fig. EV5E,F). Quantification of live-embryo imaging showed that in control embryos, the number of HP1a cluster increases at earlier cycle 14 and then decreases at later cycle 14, suggesting that these clusters fuse at late cycle 14 (Figs. 5C and EV5G). Moreover, the cluster size and maximum intensity increase over time, further supporting the fusion of HP1a clusters at late cycle 14 (Fig. 5D,E). The distribution of HP1a-GFP cluster sizes revealed a bimodal distribution in control embryos that is clearly separated at later time points in ZGA (Fig. EV5H), suggesting the presence of HP1a clusters with different behaviors. In contrast, TM embryos displayed more uniformly distributed HP1a throughout the nucleus, and the cluster size and maximum intensity remained almost constant over cycle 14 interphase (Fig. 5B,E). By comparing the size distribution of clusters at a late time point (after 30 min) (Fig. EV5I) as well as at different time points (Fig. EV5H), we found the population of bigger clusters disappears in the TM, suggesting a critical role of H3K9me2/3 in mediating the clustering of HP1a into bigger clusters, hence for the maturation of HP1a clusters.

Taken together, our quantitative CUT&Tag data suggested that, during early embryogenesis, H3K9me3 accumulation on chromatin

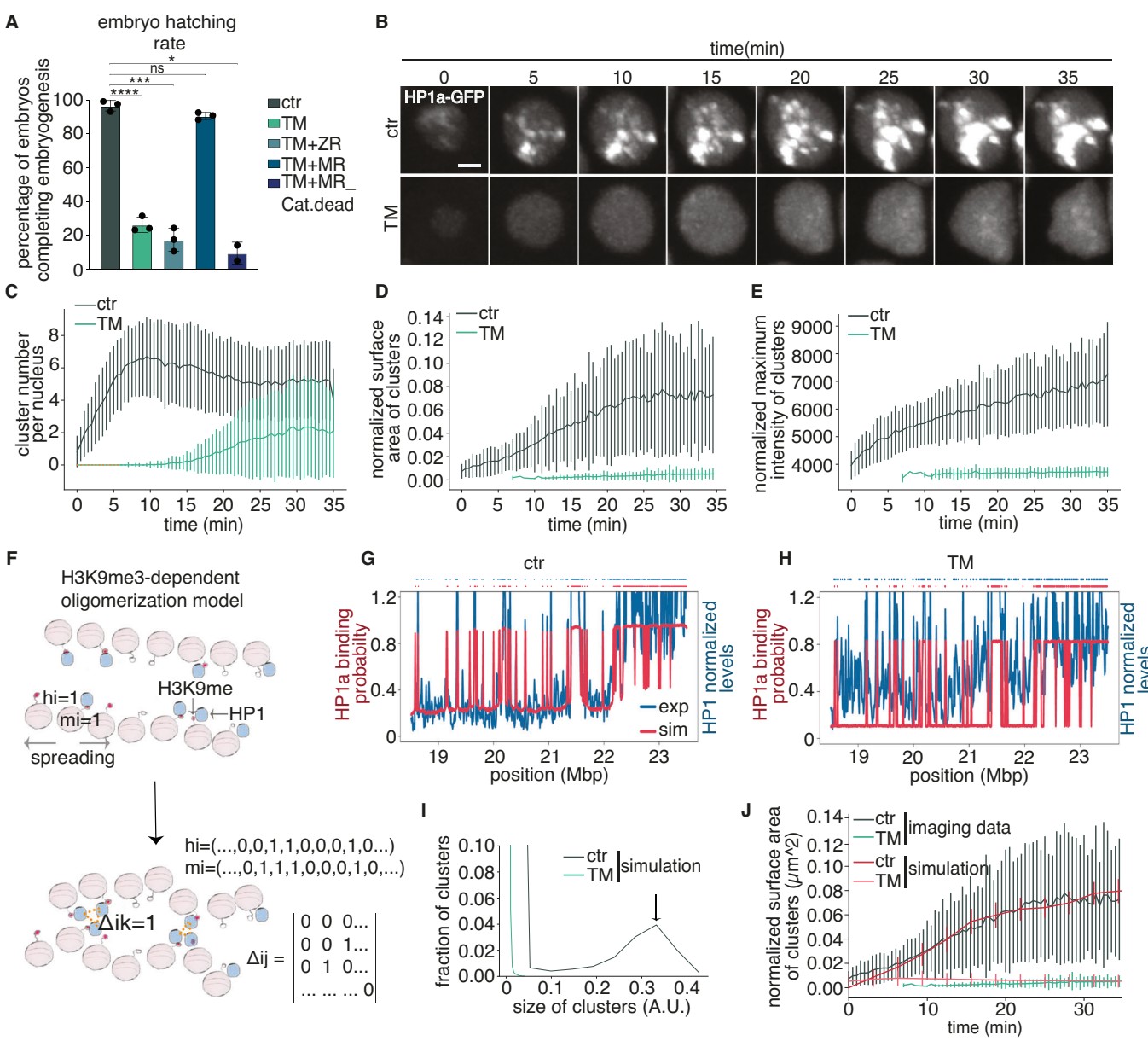

**Figure 5. H3K9 methylation is important for the formation of HP1a clusters at ZGA.**

(A) Hatching rate of ctr, TM, dSetDB1 zygotic rescue (TM + ZR), maternal wild-type rescue (TM + MR) and maternal catalytic dead dSetDB1 rescue (TM + MR_Cat.dead) embryos (ctr vs TM, P value < 0.0001, two-tailed Welch two-sample t test; ctr vs TM + ZR, P value = 0.0006, two-tailed Welch two-sample t test; ctr vs TM + MR, P value = 0.0578, two-tailed Welch two-sample t test, ctr vs TM + MR_Cat.dead, P value = 0.0160, two-tailed Welch two-sample t test). n = 3 biological replicates with 360 embryos were used for each genotype, except for TM + MR_Cat.dead rescue with n = 2 biological replicates with 240 embryos were used. Each dot represents one biological replicate. Shown mean +/− s.d. (B) A single nucleus maximum projection of 30 z-stacks images of endogenously GFP tagged HP1a for cycle 14 embryos for a time period t = 0 until 35 min. n = 3 embryos for ctr and n = 4 embryos for TM embryos were used for quantification. Scale bars, 2 μm. (C) Number of clusters per nucleus as a function of time for ctr (dark green curve, n = 3 embryos) and TM (coral green curve, n = 4 embryos). Shown mean +/− s.d. (D) Cluster size normalized over nucleus size as a function of time for ctr (dark green curve) and TM (coral green curve). Shown mean +/− s.d. (E) Maximum intensity of cluster normalized over average nucleus intensity as a function of time for ctr (dark green curve) and TM (coral green curve). Shown mean +/− s.d. (F) A schematic representation of the model where H3K9 methylation is important for HP1 oligomerization. Top panel: each locus (pink bead) can undergo H3K9 methylation (red circle) and bind HP1 (blue shape) and HP1 can also bind to the locus without H3K9 methylation (open circle). The methylation state can be spread to sites that are close along the fiber or in space contact if bound to HP1. Bottom panel: each pair of beads can form a contact (orange dashed line) which is stabilized if methylated at H3K9 and bound to HP1. (G) Normalized HP1 binding profile detected in CUT&Tag (blue curve, exp-experimental) and predicted by the model where H3K9me3 mediates only HP1-HP1 interactions (red curve, sim-simulation) in ctr. Bound sites are shown in the upper part of the panel for the simulation (red track) and in CUT&Tag (blue track, see "Methods"). (H) as in (G) for TM. (I) The distribution of network size in the ctr (dark green curve) and in the TM (coral green curve) obtained by simulation. An arrow indicates the population with the bigger size. (J) Comparison of foci size as a function of time for ctr (dark green curve) and TM (coral green curve) obtained by live-embryo imaging and size in the ctr (red curve) and in the TM (pink curve) obtained in the simulations (see "Methods"). Shown mean +/− s.d. Source data are available online for this figure.

is required for proper embryogenesis, while it is dispensable for HP1a binding to the heterochromatin. In parallel, live-embryo imaging data showed that H3K9me2/3 is required for the formation and maturation of HP1a clusters.

## Biophysical modeling predicts HP1a binding to chromatin in the absence of H3K9me2/3

To further reinforce our findings supporting the crucial role of H3K9me2/3 in facilitating HP1-HP1 interactions but not for its recruitment to chromatin, we developed a biophysical model. Consistent with in vitro findings (Canzio et al, 2011; Canzio et al, 2013), our hypothesis centered around the notion that H3K9me3 promotes the assembly of HP1a clusters by facilitating the oligomerization of HP1 molecules (Canzio et al, 2013). In order to validate this concept, we devised a simplified biophysical model that describes the dynamics of H3K9 methylation and HP1 binding on chromatin (depicted in Fig. 5F). The model is an extension of those used to study the evolution of epigenetic states in replicating cells (Becker et al, 2017). In the model, each locus can bear H3K9me3 and can also be bound by HP1. To account for the role of HP1 in propagating H3K9me3 (Allshire and Ekwall (2015)), we allowed the loci with both H3K9me3 and HP1 to propagate methylation to its flanking regions (either along the chromatin fiber or contacting region in 3D) (Fig. 5F). To model the role of H3K9me3 in mediating HP1-HP1 interactions, we stabilized the interactions between loci that are bound by HP1 and also have H3K9me3 (Fig. 5F, lower panel). We simulated the dynamics of the last 5 Mbp on chromosome 2L (chr2L:18520000-23513712). First, we fitted the model parameters to reproduce the experimental HP1 chromatin-binding profiles and H3K9me3 enrichments (Figs. 5G and EV5J, accuracy and P values Dataset EV4). We then simulated the TM by turning off the rate of H3K9 methylation, while keeping constant the binding rate of HP1 to chromatin, which could be due to redundant mechanisms of HP1 recruitment via RNA/DNA or chromatin-binding protein partners ((Canzio et al, 2014) and see "Discussion") in the absence of H3K9me3. Strikingly, this recapitulated the HP1 chromatin-binding profiles as obtained by CUT&Tag in the TM (Fig. 5H, see also H3K9me3 enrichments in Fig. EV5J and accuracy and P values in Dataset EV4). Interestingly, the model predicted also that the HP1 level bound to the PCH should be slightly decreased (~15%) upon removal of H3K9me3 (Fig. EV5K), consistent with CUT&Tag data. Given that the model could reproduce the experimental results, we next used the model to investigate the role of H3K9me3 in mediating the formation of HP1 clusters. Since our model does not account for a fraction of freely diffusing HP1 molecules, we used the network of HP1 interaction molecules bound to chromatin as a proxy for HP1 foci formation (see "Methods"). In line with the live-embryo imaging data, the model predicted that the size of the HP1 interactions networks decreased in the TM compared to the control with the disappearance of the bigger foci population (Fig. 5I,J). Thus, in accordance with our experimental data, also in our model, H3K9me3 resulted as a driving force for the formation of HP1 clusters, in particular by mediating HP1-HP1 interactions. We next set out to verify whether H3K9me3 is required for HP1-HP1 interactions. To this aim, we studied a model where H3K9me3 affected only HP1 binding to chromatin, which represents the main working hypothesis in the field (Eissenberg and Elgin, 2014),

without affecting HP1-HP1 interactions (Fig. EV5L). While the model recapitulated the HP1 and H3K9me3 profiles in the control (Fig. EV5M), it predicted that in the absence of H3K9me3 the levels of HP1 on chromatin should be heavily reduced, which is in sharp contrast with our CUT&Tag data (Fig. EV5N). Thus, a simple model of the methylation process showed that H3K9me3 is essential for mediating the interaction between HP1 (bridging) and resulting in its oligomerization.

Overall, our data indicate that: (1) H3K9me3 is inherited from the maternal germline and present from fertilization onward on chromatin in early embryos: (2) H3K9me2/3 mediates clustering and compaction of constitutive heterochromatin; (3) H3K9me2/3 and its HMTs are crucial for proper mitosis and nuclear shape (4) HP1a can bind heterochromatin in an H3K9me2/3-independent manner; (5) HP1a cluster formation depends on the presence of H3K9me2/3 (Fig. 6A,B).

## Discussion

We show that H3K9me3 resists early epigenetic reprogramming during meiosis in *D. melanogaster*, and it is intergenerationally inherited from the maternal germline to the embryo. Previous studies have detected H3K9me2/3 only around cycle 12 in *D. melanogaster* embryos (Seller et al, 2019; Yuan and O'Farrell, 2016). However, we demonstrate by both IF and CUT&Tag that both H3K9me2/3 are present on chromatin from fertilization throughout early embryonic development in the fly. The absence of H3K9me2/3 detection in earlier studies could be attributed to antibody sensitivity issues for the low-abundant H3K9me2/3 levels at the earliest stages. In *Saccharomyces cerevisiae*, the minimal requirement for H3K9 methylation inheritance over multiple generations is methylating H3K9 and deacetylating H4K16 together with recognizing unmodified H4K16 (Yuan and Moazed, 2024). During *D. melanogaster* embryogenesis, H3K9me3 establishment is not dependent on PIWI-interacting RNA (piRNA) pathway as depletion of the member of the piRNA pathway does not affect H3K9me3 levels at most TEs (Fabry et al, 2021). It will be interesting to study how H3K9me3 is intergenerationally inherited in fly embryos, whether it is dependent on RNA interference (RNAi) and small RNAs (Rechavi and Lev, 2017), and how H3K9me3 is established de novo at regions that newly gain H3K9me3 around ZGA. Furthermore, we show that the H3K9me3 mark is fully established at the pericentromeric regions within about 2 h after fertilization by ZGA, when its levels are similar to those found in late-stage embryos. We suggest that H3K9me2/3 does not regulate the gene expression at ZGA, as our GRO-seq data shows that only a few genes are transcriptionally misregulated upon loss of H3K9me3. Alternatively, the removal of H3K9me2/3 might not be sufficient to derepress the genes alone; the presence of specific transcription factors might be necessary, as shown in *C.elegans* muscle and human fibroblast cells (McCarthy et al, 2021; Methot et al, 2021). H3K9me2/3 instead is important for tissue-specific gene regulation at later stages of embryogenesis (Methot et al, 2021; Nicetto et al, 2019). Indeed, we identified new H3K9me3 peaks that appear at euchromatic repeat-free regions in late-stage embryos. This newly established H3K9me3 could play a role in a tissue-specific gene regulation in fly embryos and needs further investigation (Methot et al, 2021; Nicetto et al, 2019; Padeken et al, 2022).

Primary mouse embryonic fibroblasts (PMEF) derived from *Suvar39h* dn mice and B-cell lymphomas formed in *Suvar39h* dn

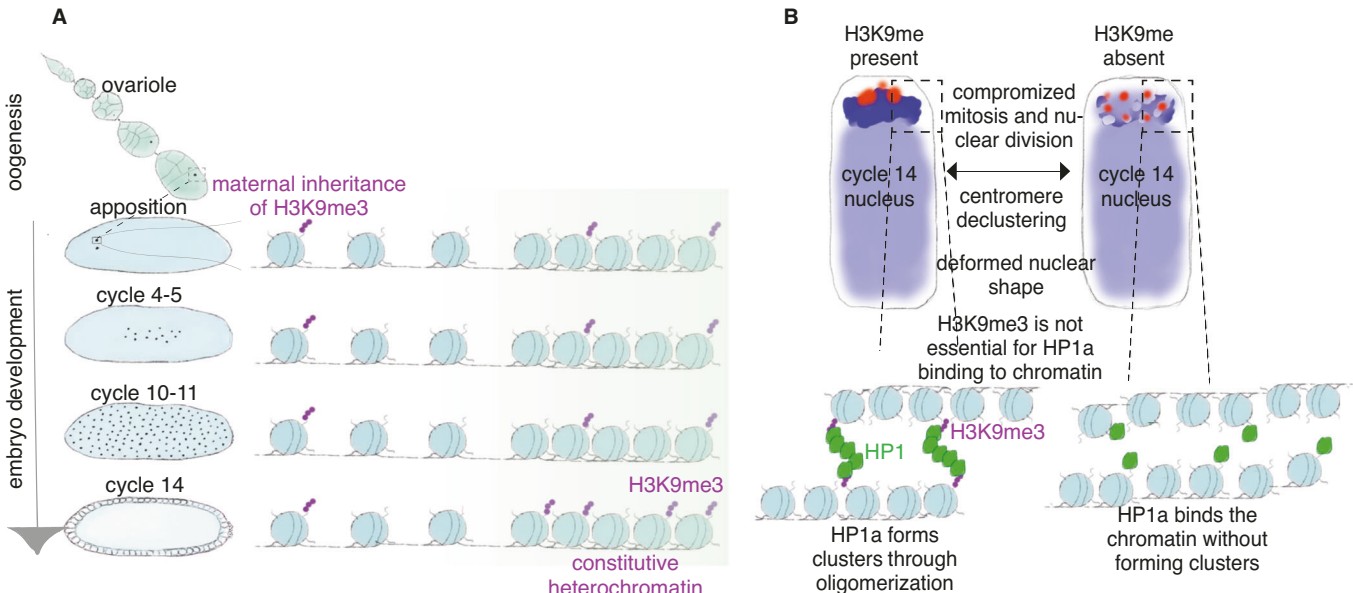

**Figure 6. H3K9me2/3 is important for the constitutive heterochromatin structure.**

(A) H3K9me3 is intergenerationally inherited from the oocyte to the embryo. Upon fertilization, H3K9me3 is found at the maternal pronucleus and actively maintained on the chromatin as the embryo progresses in development. The levels of H3K9me3 increase dramatically around cycle 14 embryos. Blue circles represent nucleosomes. Purple dots represent H3K9me3. The darker green regions represent constitutive heterochromatin. (B) A schematic representation of nuclei of cycle 14 *Drosophila* embryos with and without H3K9 methylation. Red dots represent centromeres. Dark blue represents the pericentromeric regions and brighter blue represents the rest of the genome. H3K9 methylation is required for centromere and DAPI-dense region clustering, proper chromatin segregation and nuclear shape. Although HP1a can bind to chromatin without H3K9me2/3, it cannot form clusters without H3K9me2/3. Hence, H3K9me2/3 is essential for HP1a oligomerization. Blue circles represent nucleosomes. Purple dots represent H3K9me3. Green shapes represent HP1a.

mice show a polyploid DNA content (Peters et al, 2001), whereas mice oocytes devoid of SETDB1 show misaligned chromosomes and multiple spindles (Eymery et al, 2016). In *S. pombe* depletion of Clr4 H3K9 HMT leads to an increase in the fraction of cells with lagging chromosomes (Ekwall et al, 1996). *Drosophila* TM mutant, instead, shows defects primarily in condensation and arrest at metaphase and an increased fusion of nuclei. In addition, there is almost no increase in the micronuclei formation in the absence of H3K9me2/3, consistent with no increased lagging chromosomes in TM embryos (Crasta et al, 2012). Interestingly, the misexpression of satellite repeats can lead to mitotic defects, as ectopic expression of satellite repeats either in murine erythroleukemic (MEL) cells or primary human mammary epithelial cells (HMECs) results in mitotic defects (Bouzinba-Segard et al, 2006; Zhu et al, 2011). Also, *C. elegans* embryos do not show mitotic defects in the absence of H3K9 methylation and its HMTs unlike fly embryos that we detect in this study and mice cells and oocytes (Eymery et al, 2016; Peters et al, 2001; Zeller et al, 2016). Therefore, it will be interesting to find out the causes for the mitotic defects and inter-species differences in mitotic defects in the absence of H3K9 methylation and its HMTs.

In mammalian cells, H3K9me2/3-enriched regions are found at lamin-associated domains (LADs), and a heterochromatic transgene array is released from the nuclear periphery upon removal of both H3K9 HMTs in worm embryos (Padeken et al, 2022; Towbin et al, 2012). Moreover, H3K9me3 mediates nuclear stiffness and membrane tension in human cells, and upon stretching the cells, H3K9me3 levels decrease, allowing nuclear softening (Nava et al,

2020). Our findings reveal that the absence of H3K9me2/3 leads to noticeable deformations in the majority of ZGA nuclei, suggesting a crucial role for H3K9me2/3 in preserving nuclear morphology, likely through its influence on LADs. LADs have been shown to be de novo established in early mouse embryos (Borsos et al, 2019). Although the reduction of H3K9 methylation by expression of H3K9 demethylase, KDM4D did not lead to changes in LAD structure in early mice zygotes (Borsos et al, 2019), we still speculate that H3K9 methylation could be one of the important players in establishing proper nuclear shape in the early fly embryos.

HP1a recognizes and binds H3K9me2/3 via its chromodomain. It has been suggested that the chromodomain is required to recruit HP1a to the chromatin (Bannister et al, 2001; Eissenberg and Elgin, 2014; Jacobs et al, 2001; Lachner et al, 2001). Unexpectedly, the absence of H3K9me2/3 in TM early embryos does not completely compromise the de novo recruitment of HP1a to PCH regions in the early embryo. This suggests that the initial recruitment of HP1a to chromatin occurs independently of H3K9me2/3, in line with the observation in mice where HP1β could associate with the paternal pronucleus major satellites in the absence of H3K9me3 (Fadloun et al, 2013; Probst et al, 2007; Puschendorf et al, 2008). Moreover, H3K9 methylation-independent binding of HPL-2, a homolog of HP1, is also conserved in *C. elegans* (Garrigues et al, 2015). Despite the correct recruitment of HP1a, in the absence of H3K9me2/3, we measured chromatin decompaction, upregulation of some of the repetitive elements, and centromere de-clustering at ZGA, indicating that the establishment of constitutive heterochromatin was

impaired. A fission yeast homolog of HP1a, Swi6, binds to chromatin as a tetrameric complex and can bridge the nucleosomes (Canzio et al, 2011). Moreover, it was suggested that H3K9me3 acts as an activator to remove the autoinhibition of Swi6 dimers so that Swi6 dimers can oligomerize (Canzio et al, 2013; Grewal, 2023). In turn, Swi6 oligomerization increases its ability to discriminate methylated H3K9 from non-methylated H3K9 nucleosomes and prevents Swi6 promiscuous nucleic acid binding (Biswas et al, 2022). In line with these studies, we suggest that in ZGA, H3K9me3 is required for HP1a oligomerization and cluster formation and prevents HP1a promiscuous nucleic acid binding. Having said that, in vitro studies suggested that the presence of H3K9me3 modification on chromatin plays a crucial role in determining the binding dynamics of HP1 so that in the absence of H3K9me3, HP1 exhibits different binding and dissociation behaviors (Kilic et al, 2015). Therefore, it will be interesting in the future to study the in vivo dynamics of HP1 using our model system coupled with super-resolution single-molecule microscopy. Interestingly, a complete loss of H3K9me2/3 by removing all H3K9 HMTs and loss of HP1a clustering did not lead to a complete loss of DAPI-dense regions in early embryos, although the DAPI-dense regions are much smaller in volume. Instead, in immortalized mouse embryonic fibroblasts (iMEFs), the complete removal of all six H3K9 HMTs leads to almost complete loss of DAPI-dense regions (Montavon et al, 2021). Furthermore, in the same cells removal of five H3K9 HMTs (SUV39H1/H2, SETDB1 and G9A/GLP) together with inhibition of H3K27me3 HMTs (EZH1/2) led to the drastic disorganization of chromocenters in many cells compared to five H3K9 HMT mutant cells, implying compensation of H3K27me3 for the loss of H3K9me3 (Fukuda et al, 2023). The presence of the visible DAPI-dense regions in TM embryos means that there might be additional mechanisms independent of H3K9 methylation and its HMTs to form the DAPI-dense regions in early fly embryogenesis. Indeed, AT-hook containing protein, D1 was shown to be enriched at some satellite repeats in early Drosophila embryos (Aulner et al, 2002). Double mutant of D1 and another satellite binding protein Prod leads to chromocenter disruption implicating their involvement in chromocenter formation (Jagannathan et al, 2019).

In Drosophila early embryogenesis, HP1a undergoes liquid–liquid phase separation (Strom et al, 2017), whereas, in mouse early 2-cell zygotes, the pericentromeric regions undergo liquid–liquid phase separated condensation (Guthmann et al, 2023) without expressing HPα (Wongtawan et al, 2011). Moreover, in mouse cell lines, the clustering of PCH regions, chromocenters, do not show the properties of HP-driven liquid–liquid phase separated condensates, suggesting that alternative mechanisms might drive heterochromatin compaction (Erdel et al, 2020; Erdel, 2023). It would be interesting to study the function of HP1-driven clustering in early fly embryogenesis in the future.

The binding of HP1a in the absence of H3K9me2/3 suggests that there are other mechanisms for HP1 binding to chromatin. One mechanism could be via RNA, given that mammalian HP1α requires both H3K9me2/3-binding and RNA-binding activities to associate with chromatin and fission yeast Swi6 can bind to RNA via its hinge region (Keller et al, 2012; Muchardt et al, 2002). Moreover, treatment of mouse cells with RNase leads to dispersion of HP1α, which is restored by the addition of nuclear RNA, meaning RNA is important for HP1α localization (Maison et al,

2002). HP1a redistributed along the entire polytene chromosomes of the salivary gland of D. melanogaster in RNA helicase, spindle E mutant, implying that RNA might play an important role in HP1a binding to pericentric heterochromatin (Pal-Bhadra et al, 2004). Alternatively, HP1 might be recruited via interacting protein partners that can directly bind the AT-rich heterochromatic DNA or RNAs transcribed from pericentromeric regions. Indeed, an in vitro reconstitution study showed that HP1 binding to chromatin requires additional auxiliary factors in addition to H3K9 methylation (Eskeland et al, 2007). In D. melanogaster, one such protein could be HP2, which binds to HP1 and has AT-hook domains that can bind to DNA (Shaffer et al, 2006; Shaffer et al, 2002).

In this study, we presented evidence of an epigenetic system designed to transmit maternal H3K9me3 and regulate the establishment of constitutive heterochromatin in early embryos of the next generation. We speculate that H3K9me3 and H3K27me3 (Zenk et al, 2017), beyond their physiological roles in transmitting epigenetic information across generations, may also serve as carriers of environmentally induced germline chromatin changes. Further research is required to explore these possibilities.

# Methods

## Fly stocks

All the flies used in this study were grown in standard fly food at 25 °C. The fly stocks used in this study: as a control TRIP line (BDSC 36303), G9a-shRNA line (BDSC 36798), mat-tub-Gal4 (BDSC # 80361, only mat-tub-Gal4 on 2nd chromosome was used), mat-tub-Gal4 (BDSC 7063), G9a knockout from P.Spierer lab, Su(var)3-9 knockout line from J. Brennecke lab (IMBA, Austria), H2Av-mRFP1 line (BDSC 23651), dSetDB1 shRNA (this study), UASp-Flag-HA-dSetDB1 (dSetDB1 Zygotic rescue, this study), endogenous Flag-HA-dSetDB1 (dSetDB1 maternal rescue, this study), endogenous Flag-HA-catalytic dead dSetDB1 (dSetDB1 catalytic dead maternal rescue, this study), endogenous GFP-dSetDB1 (this study), endogenous Su(var)3-9-GFP (this study), endogenous HP1a-GFP (this study) as a control for live imaging.

## RNAi-mediated maternal knockdown of genes

The maternal knockdown of genes was performed by using the maternal tubulin Gal4 driver of Bloomington stock (BDSC # 80361), where only Gal4 driver on 2nd chromosome is used and RNAi was generally done as described in (Zenk et al, 2017). In brief, shRNA containing virgin females were crossed to mat-Gal4 males and F1 generation was obtained where the gene of interest is depleted in the germline therefore the protein of interest is not maternally loaded into the early embryo. F1 siblings were aged for 4 days and transferred into the cage, and the embryo was collected on an apple juice agar plate for subsequent studies.

## Rescue for shRNA specificity

To verify the specificity of shRNA against dSetDB1, we used early oogenesis driver mat-tub-Gal4 line (BDSC 7063) to knockdown dSetDB1 and overexpress the transgene that is short hairpin-resistant.

## Triple mutant cross

Triple mutant fly crosses were done by crossing the virgins that are homozygous knockout for Su(var)3-9 and carry shRNAs against dSetDB1 and G9a and males that are also homozygous knockout for Su(var)3-9 and carry maternal tubulin Gal4 driver on 2nd chromosome only from of Bloomington stock (BDSC # 80361). F1 flies were collected within 24 h and aged for 4 days and embryos were collected for further experiments.

## Zygotic and maternal rescues

Zygotic rescue embryos were generated by crossing triple mutant F1 virgins to males that carry a transgene for dSetDB1 that is short hairpin-resistant. Also, males were wild type for Su(var)3-9. Both wild-type and catalytic dead dSetDB1 maternal rescues were performed by creating the flies that contain an endogenous single copy short hairpin-resistant dSetDB1 of either wild-type or catalytic inactive dSetDB1 (Y1236N) in the background of Su(var)3-9-KO, G9a-shRNA and dSetDB1 shRNA and crossed with males that had maternal tubulin-Gal4 driver on 2nd chromosome only from of Bloomington stock (BDSC # 80361) and knocked out homozygous Su(var)3-9.

## Recombination of genes and creation of triple mutant flies

Flies that carried the modified version of gene of interest, shRNA or mat-tub-Gal4 driver were crossed and F1 female virgins (in germline of which the recombination takes place) were collected and crossed to the appropriate balancer line. Single F2 male was crossed to the appropriate balancer virgin females and after 2–3 days the male was sacrificed and genomic DNA was extracted and genotyping PCR was done with specific primers listed in Dataset EV6 to check the recombination bearing flies.

## Embryo phenotypic characterization

Embryogenesis completion (embryo hatching rate) was quantified as described in (Zenk et al, 2017). Also, the cellularisation rate was performed as described in (Ibarra-Morales et al, 2021). Briefly, embryos were collected on an apple juice agar plate for 0–1 h and 120 and 50 embryos were aligned on a plate in batches of 10 embryo for hatching and cellularisation rate, respectively. The embryos were incubated in 25 °C. After 1–1.5 h the number of cellularized embryos was quantified. The number of hatched embryos was counted after 48 h.

To quantify approximate time an embryo takes from fertilization till the end of cycle 14, the embryos were collected 0–1 h and 60 embryos per replicate were aligned and incubated in 25 °C. After 1–1.5 h the embryos were checked every 10 min to count the time embryo takes to gastrulate.

## Embryo collection

Embryo collections for IF was done as described in (Zenk et al, 2021). Both for IF and RNA-FISH the embryos were collected in the following way. 0–4 h old embryos were collected on an apple juice agar plate and dechorionated with 50% bleach, washed with water and dried with tissue paper. The dried embryos were transferred to the 5 ml Heptane. Immediately after transferring the embryos 5 ml of 4% paraformaldehyde in 1× PBS was added and the embryo were shake vigorously by hand and incubated on an orbital shaker for 20 min. To devitellinized the embryos, the lower aqueous phase was removed and 5 ml of methanol was added followed by 30 s of vortexing. The devitellinized embryos sink to the bottom of the tube and can be collected and washed with cold methanol and stored in −20 °C until the further usage. For DNA-FISH, similar procedure was done except the 4% paraformaldehyde was in FISH buffer A (60 mM KCl, 15 mM NaCl, 15 mM Pipes (pH 7.4), 2 mM EDTA, 0.5 mM EGTA (pH 8), 0.5 mM Spermidine, 0.15 M Spermine).

For Gro-seq, the embryo collection was done as described in (Ibarra-Morales et al, 2021) and 500 embryos were used per replicate. Briefly, 0–1 h embryos laid on apple juice agar plate were aged at 25 °C for 2 h. Cycle 14 embryos were hand-staged with submerging the embryos in halocarbon oil 27 (Sigma Aldrich H8773) under a stereoscope microscope with transmitted light. In total, 100 embryos were transferred to 1.5 ml Eppendorf tube and snap-frozen and kept at −80 °C until further use. For RNA extraction 25 embryos were used. In all, 0–1 h embryos laid on apple juice agar plate were aged at 25 °C for 2 h to get cycle 14 embryos or directly used to get before cycle 9 embryos. Embryos were hand-staged with submerging the embryos in halocarbon oil 27 (Sigma Aldrich H8773) under a stereoscope microscope with transmitted light. 25 hand-staged embryos were transferred into 50 µl of Trizol (ThermoScientific, 15596026), homogenized by a pestle, snap-frozen and kept in −80 °C until further usage.

For embryo fractionation 0–4 h old embryos were collected on an apple juice agar plate and dechorionated with 50% bleach, washed with water and dried with tissue paper. The dried embryos were transferred into 1.5 ml protein low binding Eppendorf tubes and snap-frozen in liquid nitrogen and kept in −80 °C freezer until further use.

For CUT&Tag 50–100 embryos were used per replicate. The embryo collection and PFA fixation of embryos for Cut and Tag was done as described for ChIP-sequencing according to (Zenk et al, 2021). Briefly, 0–40 min embryos laid on apple juice agar plate were aged at 25 °C for time needed to get specific stages: 40 min of aging for before cycle 9, 90 min of aging for cycle 9–13, 140 min of aging for cycle 14. For embryos at stage 17, 0–2 h embryos were collected on a plate and aged for 18 h at 25 °C. Embryos on a plate were dechlorinated with 50% bleach and washed thoroughly with water, carefully dried and transferred to 10 ml of heptane. 5 ml of fixative composed of 1% paraformaldehyde in buffer A (60 mM KCl, 15 mM NaCl, 15 mM HEPES (pH 7.6), 4 mM MgCl$_2$) was added and embryos are incubated for 15 min at orbital shaker at maximum speed. The 225 mM of glycine was added to stop the fixation and the embryos were incubated for 5 min on a rotator wheel. The fixative buffer is removed and embryos were washed with buffer A + 0.1% TritonX100 and embryos with correct developmental stage were hand-picked under the microscope and snap-frozen in Eppendorf tubes and kept at −80 °C until further use.

## Cloning of shRNA, transgenes, and production of transgenic and CRISPR-Cas9 flies

A short hairpin against 3'UTR of dSetDB1 was cloned into the vector Walium 22 according to (Ni et al, 2011; Perkins et al, 2015)

and integrated into locus 68A4 (fly line: $y^1$ $w^{67c23}$;P{CaryP } attP2). dSetDB1 overexpression rescue construct was amplified from 0 to 1-h embryo cDNA and restriction sites and Flag-HA was added to the N-terminus of the cDNA and cloned into the vector pUASp-attB (DGRC_1358) and integrated into locus 25C6 (fly line: $y^1$ $w^{67c23}$;P{CaryP } attP40). Two different strategies were used to endogenously tag the proteins. Fly stocks where dSetDB1 and Su(var)3-9 genes were replaced by attP were borrowed from G. Pyrowolakis lab (University of Freiburg). For endogenously tagging the GFP-dSetDB1, Flag-HA-dSetDB1 and Su(var)3-9-GFP proteins, both dSetDB1 and Su(var)3-9 genes were amplified from genomic DNA of adult flies. Restriction sites and either Flag-HA or GFP were added to the N-terminus of dSetDB1a and GFP was added to the C-terminus of Su(var)3-9 and cloned into the vector RIVwhite (DGRC_1330). To create a catalytic inactive dSetDB1, site directed mutagenic PCR was performed on Flag-HA-dSetDB1 with K10 3'UTR cloned on pJET. The PCR product was digested with Dpn1 enzyme. After digestion the product was transformed into competent E. coli cells. The construct was cloned into RIVwhite plasmid. For Flag-HA-dSetDB1 and catalytic dead Flag-HA-dSetDB1 (Y1236N), the 3'UTR was replaced by 3'UTR of a gene K10 to create a short hairpin-resistant allele. Constructs were integrated into flies that contain corresponding attP sites.

For HP1a-GFP, two guide RNAs (gRNAs) were cloned into pCFD4-U6:1_U6:3tandemgRNAs (addgene #49411) according to (Port et al, 2014). The donor plasmid was constructed by amplifying the HP1a gene with extra left arm of around 800 bp and right homology arm of around 900 bp from the adult fly genomic DNA. Fragments of HP1gene with left arm homology, GFP and right arms homology were fused by fusion PCR and at the same time restriction sites were added. Both PAM sequences were mutated by using two primers for each PAM that flank the PAM and amplifying the construct. The construct was cloned into the vector pJet1.2 (ThermoFisher #1232). Both pCFD4 and pJet1.2 plasmids were injected into yw;;nos-Cas9(III-attP2) stock and the positive flies were selected for future experiments. The primers used for cloning are listed in Dataset EV6.

## Immunofluorescence, imaging, and quantification

Immunofluorescence was performed as described in (Zenk et al, 2021). For imaging of embryos with apposing pronuclei, 0–1 h embryos were collected on an apple juice agar plate. All images were acquired by high-resolution microscopies Zeiss LSM 880 Airyscan or Zeiss LSM 900 Airyscan 2. All the images were processed with airyscan processing software. Stacks were assembled either by Fiji (Schindelin et al, 2012) or Imaris 9.7 (Bitplane). All antibodies used for this study can be found in Dataset EV6. For quantification of volume of DAPI-dense regions a surface function of Imaris 9.7 (Bitplane) was used. Briefly, surfaces were created on the raw DAPI image stacks and default parameters were chosen. Threshold of 4000–20107 was used for absolute intensity and volumes above $0.202\,\mu m^3$ was chosen. For each genotype 7–8 embryos were used for quantification. For quantification of H2Avgamma positive nuclei, the average intensity of z-stack images was assembled and the total number of nuclei per view per embryo was counted by DAPI staining. In order to count the fraction of H2Avgamma high nuclei, the number of nuclei with high H2Agamma signal was divided to total number of nuclei per embryo.

## Nuclear shape analysis

Airyscan LSM 880 imaged embryos were processed with airyscan software and for further analysis processed imaged were used. Arivis vision4D software was used for the analysis. Briefly, to account for the changes that happen to the nuclear shape during cellularization, the length of chromatin from apical to basal side was measured from DAPI staining and nucleus with similar sizes for both control and TM was used for the analysis. Nucleus was segmented by using the total intensity threshold of Lamin staining with more than 99% and filtering the nuclei with a size threshold bigger than $20\,\mu m^3$ to remove small non-nuclei objects. Also, all nuclei that is not full shape in the frame of imaging were removed from the analysis. For control 9 embryos with approximately on average 66 nuclei per view per embryo were used. For triple mutant, 9 embryos with approximately on average 51 nuclei per view per embryo were used. Mean with ±standard deviation was plotted.

## Probe labeling, DNA-FISH, and quantification

The SATIII DNA probe used in the DNA-FISH is amplified from a plasmid that contained one unit of SATIII repeat. The probe was generated by High Fidelity ATTO488 PCR labeling kit according to the kit (Jena Bioscience) protocol. The primer sequences that were used for probe production are: forward primer: TATTCTTA-CATCTATGTGACCattttta and reverse primer: GTTTTGAG-CAGCTAATTACCag. DNA-FISH was performed as described in (Zenk et al, 2021). All incubations were done on a nutator at RT, unless otherwise mentioned. Briefly, fixed embryos were re-hydrated by washing the embryos in 1 ml of following solutions with incubating for 3–5 min: (1) 90% MeOH with 10% PBTw0.1% (0.1% Tween-20 in PBS); (2) 70% MeOH with 30% PBTw0.1%; (3) 50% MeOH with 50% PBTw0.1%; (4) 30% MeOH with 70% PBTw0.1%; (5) 100% PBTw0.1%. Then embryos were incubated in 200 µg of RNase A in 1 ml PBTw0.1% at 4 °C for overnight and next day they were permeabilized with 1 ml of PBS with 0.3% of Triton X (PBS-T$_{X0.3\%}$) for 2.5 h. Embryos were treated with 1 ml of prehybridization buffer and incubated for 20 min in the following step: (1) 80% PBS-T$_{X0.3\%}$ with 20% prehybridization mixture (pHM) (50% deionized formamide, 4× SSC, 100 mM NaH2PO4 (pH 7.0), 0.1% Tween-20); (2) 50% PBS-T$_{X0.3\%}$ with 50% (pHM); (3) 20% PBS-T$_{X0.3\%}$ with 80% (pHM) 4) 100% pHM. The embryos were incubated in 100% pHM at 82 °C for 15 min. Around 400 ng of labeled probe was used per experiment. For quantification of SATIII DNA-FISH volume a surface function of Imaris 9.7 (Bitplane) was used. Briefly, surfaces were created on the raw DAPI and SATIII image stacks and default parameters were chosen. Threshold of 2000–15078 was used for absolute intensity and area above $3.50\,\mu m^2$ was chosen. For number of voxels per image=1, above 10 was selected. For each genotype, 4–5 embryos were used for quantification.

## RNA-FISH and quantification

For RNA-FISH HuluFISH from PixelBiotech was used. A custom designed and atto647-labeled probe set that covers the length of SATIII was used for high signal to noise ratio single-molecule FISH analysis. RNA-FISH was performed according to manufacturer's

instructions. Briefly, fixed embryos were rinsed twice and washed $3 \times 10$ min with 0.8 ml Hulu Wash buffer ($2\times$ SSC, 2 M Urea and 0.1% Tween-20), before adding 50 µl Hulu Hybridization buffer ($2\times$ SSC, 2 M Urea, 10% Dextran sulfate sodium salt, $5\times$ Denhardt's solution and 0.1% Tween-20) with 0.5-µl Hulu probe. The embryos were incubated at 30 °C with 400 rpm in the dark for overnight. The samples were rinsed twice and washed $4 \times 10$ min with 0.8 ml Hulu Wash buffer. DAPI (1 µg/ml) counterstaining in Hulu Wash buffer were performed for 5 min and rinsed once more with 0.8 ml Hulu Wash buffer. The embryos were pipetted onto a coverslip, excess buffer was removed and replaced by Aqua-Poly/Mount (Polysciences) mounting medium. The embryo was repositioned under the microscope if needed and the mounting medium was dried for 20 min before mounting the coverslips on slides. Slides were incubated for 2 h at RT and stored at 4 °C until imaging. As a control, the RNase A treatment of embryos was performed before the staining steps. The embryos were rinsed twice and washed twice for 10 min with 0.8 ml PBS-T (PBS with 0.1% Tween-20) and incubated with final concentration of 0.02 µg/µl RNase A (Sigma Aldrich) in PBS-T for 1.5 h at 37 °C.

After background subtraction and max intensity projection, SATIII foci intensity was measured using a custom python plugin made by Patrick van Nierop Y Sanchez from Ingrid Lohmann's lab at COS. To capture signals with a wide range of intensity during thresholding, the histograms of all analyzed images were stretched to one reference image (highest brightness without having any overexposed pixels) using the stack contrast adjustment plugin (Michalek et al, n.n.). After having the images that have similar brightness histograms, auto thresholding with the Moments method was used to select ROIs to measure the intensities of the original images. All washes and incubations were performed on a tilting mixer unless otherwise specified. All reagents were RNase-free. For control and triple mutant 19–20 embryos and for RNase treated triple mutant 9 embryos were used for quantification.

### Embryo fractionation

Around 200 µl of embryos was used per sample. In all, 0–4 h embryos were thawed on ice and resuspended in 1 ml hypotonic buffer (15 mM HEPES (pH 8.0), 350 mM Sucrose, 5 mM MgCl$_2$, 10 mM KCl, 0.1 mM EDTA, 0.5 mM EGTA, 10 mM Beta-Mercaptoethanol, 0.2 mM PMSF, 1xProtease inhibitors) and dounced with 15 strokes. The extract was transferred to the protein low binding (PLB) 1.5 ml Eppendorf tubes and incubated on ice for 15 min. The nuclei were pelleted at $9000 \times g$ for 15 min at 4 °C and the fatty yolk was carefully removed. The nuclei were washed with 500 µl of hypotonic buffer, pelleted and resuspended in high-low-Salt buffer mix (20 mM HEPES (pH 7.9), 25% glycerol, 1.5 mM MgCl$_2$, 400 mM KCl, 0.2 mM EDTA, 0.2 mM PMSF, 0.5 mM DTT, 1×Protease inhibitors) and incubated on ice for 15 min. The chromatin was pelleted at $20{,}000 \times g$ for 15 min at 4 °C. The supernatant that contains the nuclear soluble fraction was used for affinity purification by using flag tag. The quality of the fractionation was checked with Western blot. Before proceeding to the affinity purification, protein concentration was measured by Bradford assay.

### Affinity purification

Three biological replicates were performed for all samples. First 30 µl of EZview$^{\text{TM}}$ Red ANTI-FLAG$^{\text{TM}}$ M2 affinity gel (Sigma,

F2426) beads per sample were incubated with the same amount of 0.1 M glycine (pH 2.5) for 3 min at room temperature (RT). Ten times for (300 µl) of 1 M Tris-HCl (pH 7.9) was added and the beads were pelleted at $1000 \times g$ for 3 min at RT. The beads were washed twice with 300 µl of IP buffer(20 mM Tris-HCl (pH 7.5), 10% glycerol, 5 mM MgCl$_2$, 150 mM KCl, 0.05% NP-40, 0.1% Tween-20, 1 mM DTT, 1× Protease inhibitor cocktail). 30 µl was reconstituted with IP buffer. 30 µl of beads were added to the nuclear extract and incubated for 4 h on the rotating wheel at 4 °C. The beads were centrifuged at $800 \times g$ for 3 min at 4 °C and washed three times with 500 µl of IP buffer. After the final wash, three bead volume of 200 ng/µl flag peptide (Sigma F3290) diluted in IP buffer was added to the beads and incubated for 1 h on the rotating wheel at 4 °C. The supernatant that contains eluted proteins were kept for the Mass spectrometry. Then the beads were eluted one more time as previous elution. The supernatant of each elution were combined and cleared with Pierce columns (ThermoFisher, 69702). The quality for the affinity purification was checked by western blot before subjecting the samples for mass spectrometry analysis.

### Mass spectrometry sample preparation

Bait and control Flag affinity purifications were prepared pursuing the paramagnetic bead-based single-pot, solid-phase-enhanced sample-preparation (SP3) method essentially as described (Ibarra-Morales et al, 2021). Briefly, Flag peptide eluates were buffered and adjusted to 50 µl volume in 100 mM Tris-HCl (pH 7.5), reduced (DTT; 12 mM final concentration) at 45 °C for 30 min, followed by cysteine alkylation (2-iodoacetamide; 40 mM final concentration) for 30 min at room temperature (RT) in the dark. Overall, 100 µg SP3 beads (1:1 mixture of GE Life Sciences carboxylate-modified Sera-Mag Speed Beads A (hydrophilic) and B (hydrophobic), respectively) were added together with acetonitrile (70% final concentration) and incubated on a shaker for 15 min at RT. In all further steps, magnetic beads were collected on an in-house fabricated Neodymium (Supermagnete, Germany) magnetic rack. Beads suspension was washed twice with ethanol (70%) and once with 100% acetonitrile. Proteolytic digestion was carried out in 50 µl 50 mM ammonium bicarbonate. First, we added 100 ng LysC (FUJIFILM Wako Pure Chemical Corporation) incubating 2 h at 37 °C followed by addition of 500 ng trypsin (Promega) with overnight incubation at 37 °C. After digestion, peptides were sonicated for 1 min (water bath sonicator) and spun down. For peptide clean-up bead suspension was vacuum concentrated to 5 µl, sonicated for 1 min (water bath sonicator), adjusted to >90% acetonitrile concentration and washed twice with neat acetonitrile, incubating 15 min at RT each time. Lastly, samples were eluted sequentially by addition of 30 µl ultra HPLC-grade water (Pierce) followed by 30 µl 0.1% trifluoroacetic acid, pooling the eluates. Eluted samples were concentrated in vacuo and resuspended in 0.1% formic acid prior to nanoLC-MS.

### Liquid chromatography-mass spectrometry

Samples were measured on a Q Exactive Plus coupled to an EASY-nLC 1200 liquid chromatography system (both ThermoFisher Scientific), as previously described (Musa et al, 2018) but with some modifications. A one-hour nonlinear gradient was applied 5 min: 5%, 40 min: 45%, 4 min: 80% (250 nl/min flow rate), followed by a

column washout step: 5 min: 80% B buffer (350 nl/min flow rate), for which an in-house packed fused-silica emitter column (75 μm × 25 cm) (SilicaTip PicoTip; New Objective) packed with 1.9-μm reverse-phase ReproSil-Pur 120 C18-AQ beads (Dr. Maisch) was employed.

Samples were injected twice and measured in data-dependent mode following the "sensitive" or "fast" method described previously (Kelstrup et al, 2012). Briefly, for the "sensitive" method a scan range of 300–1650 $m/z$, 70,000 resolution, AGC 3e6, and maximum injection time of 50 ms was used at the MS1 level, followed by fragmentation of the top 12 most intense precursors, 1.2 $m/z$ isolation window, resolution 35,000, 1e5 AGC, NCE 28, and 120 ms maximum injection time at MS2. The "fast" method had equal settings at MS1, but at MS2 a top 12, 2 $m/z$ isolation window, resolution 17500, 1e5 AGC, NCE 28, and 60 ms maximum injection were employed.

## Mass spectrometry data analysis

RAW data were processed with MaxQuant (v1.6.14.0) (Cox and Mann, 2008). Files were searched against a UniProt *Drosophila melanogaster* database containing Swiss-Prot and TrEMBL sequences (23523 entries, downloaded on 24.01.2022) plus an extended version of the MaxQuant contaminant database. Trypsin/P was used as enzyme and up to 2 missed cleavages were allowed. Carbamidomethylation of cysteines was used as fixed modification, and variable modifications included oxidation (M), acetylation of protein N-termini, and deamidation (N, Q). Relative quantification was calculated using MaxLFQ (Cox et al, 2014), enabling the match between runs option (matching window 0.5 min). Both peptide and protein FDR were kept at 1%. All other settings were kept at default values.

The obtained proteinGroups.txt file was further analyzed in R version 4.3.1 (Team RC (2023)) using in-house developed R scripts (Gomez-Auli et al, 2021) assembled as an R package (to be published elsewhere), which had been built around the DEP suite (Zhang et al, 2018). First, reverse, contaminants, and entries only identified by site were removed. At least two valid values in any condition were required for further analysis. Missing values were imputed following a mixed strategy. Values completely missing in a condition or missing in up to 50% of the replicates of a condition were imputed using a left-censor strategy as described (width = 0.3, shift = 1.6) (Keilhauer et al, 2015). The remaining missing values were imputed using the missing forest algorithm (Stekhoven and Buhlmann, 2012).

Differentially enriched proteins were evaluated by fitting a linear model using *limma* (with trend=TRUE) (Ritchie et al, 2015). Obtained $P$ values were adjusted by multiple hypothesis testing using the Benjamini–Hochberg procedure (Benjamini and Hochberg, 1995). Differentially enriched proteins were classified by having a $P$ value ≤ 0.05.

## CUT&Tag

In total, 50–100 embryos were used per replicate. The snap-frozen embryos were defrosted and resuspended in digitonin wash buffer (20 mM HEPES (pH 7.5), 150 mM NaCl, 0.5 mM Spermidine, 5 mM sodium butyrate, 0.01% Digitonin and 1× Protease inhibitor cocktail). The nuclei were extracted by breaking the embryo using a

pestle for 30 s. H3 was used as a loading control and for that, after extracting nuclei, the suspension was equally divided into Eppendorf tubes. The rest of the protocol was performed according to (Kaya-Okur et al, 2019). For quantification of global change, 1 pg of lambda spike-in DNA previously digested and adapter loaded pA-Tn5 was used. The list of the antibodies used in this study including CUT&Tag is given in the Dataset EV6.

## RNA extraction and RT-qPCR

RNA extraction and RT-qPCR were performed as described in (Zenk et al, 2021). For RNA extraction of SATIII, all steps were done according to (Zenk et al, 2021), except the DNase treatment was performed twice to remove the DNA completely as SATIII is a highly repetitive element. Briefly, after first DNase (2U Turbo DNase Ambion) digestion, another 2U of the DNase was added and the sample was digested for 30 min at 37 °C. The reaction was stopped by addition of 15 mM EDTA and incubating at 75 °C for 10 min. RNA was cleaned up by RNA Clean and Concentrator™-100 (Zymo Research) according to the kit instructions. For RT-qPCR of SATIII, the following was performed: for reverse transcription the QuantiTect Reverse Transcription Kit (Qiagen) was used according to manufacturer's instructions. The cDNA was diluted 1:2 with water. qPCRs were performed using LightCycler® 480 SYBR Green I Master Mix (2X) (Roche) with one reaction containing: 7.5 μl 2X SYBR Green, 1 μl cDNA, 0.75 μl each of forward and reverse primers (10 μM) and 5 μl H2O. Each sample was measured in triplicate, one mastermix was made which was divided over three wells of the 384-well plate (Roche). The list of primers used for RT-qPCR is given in Dataset EV6.

## Western blot

Hand-staged embryos were collected in Laemmli buffer and homogenized by a pestle and boiled for 5 min at 95 °C. In total, 2–10 embryos were loaded into SDS-PAGE gel and transferred to PVDF membrane. Primary and secondary antibodies and ECL were used and the signal was developed by ChemiDoc (Bio-Rad). Ponceau staining was used as a loading control together with loading control proteins. All antibodies used for this study can be found in Dataset EV6.

## Gro-Seq

Three for control and four for the triple mutant biological replicates were performed. The steps of Gro-seq were conducted as described in (Ibarra-Morales et al, 2021). Briefly, snap-frozen 500 embryos in a 1.5 ml tube were resuspended in the nuclei extraction buffer (3 mM CaCl$_2$, 2 mM MgCl$_2$, 10 mM Tris-HCl pH 7.5, 0.25% Np-40, 10% glycerol, protease inhibitors, and RNase inhibitor 4 U/mL) and stroked 30-35 times. The lysate was cleared from debris and the nuclei were pelleted and washed four times with nuclei extraction buffer. Then, another wash with freezing buffer was done and nuclei were resuspended with 100 μl of freezing buffer. The Nuclear Run-On (NRO) reaction was performed by adding 100 μl of 2× NRO buffer (10 mM Tris-HCl, 5 mM MgCl$_2$, 1 mM DTT, 300 mM KCl, 1% Sarkosyl, 0.5 mM ATP, CTP and GTP, and 0.8 U/μL RNase inhibitor) and Bio-11-UTP with a final concentration of 1 mM. The reaction was incubated at 30 °C for 5 min and RNA was

extracted by TRIzol extraction. The reverse transcriptase buffer was added to the extracted RNA and RNA was incubated at 95 °C for 7 min. 30 μl of Dynabeads MyOne Streptavidin C1 (Invitrogen) was added to purify the RNA. Purified RNA was washed and incubated with Polynucleotide kinase (Thermoscientific) at 37 °C for 30 min. After Phenol:Chloroform purification, 3' end adapter were ligated to the RNA by incubating the RNA with T4 RNA ligase 2 Truncated KQ (home-made) at 15 °C for 16 h. RNA was purified by SPRI beads and biotinylated RNA was enriched by using the Dynabeads MyOne Streptavidin C1 as described above. 5'-end adapter was ligated to the RNA by incubating the RNA with T4 RNA ligase 1 at 25 °C for 2 h. RNA was purified by SPRI beads and biotinylated RNA was enriched by using the Dynabeads MyOne Streptavidin C1 as described above. The purified RNA was reverse transcribed by using SuperScript IV Reverse Transcriptase (ThermoFisher Scientific) following by incubation at 50 °C for 1 h. cDNA was amplified with specific primers by using Q5 High fidelity 2× mastermix.

## GRO-seq data processing and analysis

The GRO-seq data was processed as in (Ibarra-Morales et al, 2021). In brief, 5' adaptamers were trimmed using Cutadapt (Martin, 2011) (version 2.5) (-a "NNNNTGGAATTCTCGGGTGCCAAGG" --overlap=3 --minimum-length=12 --max-n). Afterward, the SE library was mapped using Bowtie2 v2.3.3.1 with default parameters and bowtie2 –local alignment (Langmead and Salzberg, 2012). Only alignments with MAPQ3 or higher were kept. For sequence read quality control samtools v1.10.0 (Li et al, 2009), deeptools v3.3.1 (Ramirez et al, 2014), FastQC v0.11.5 (Andrews, 2010) and MultiQC v1.8 (Ewels et al, 2016) were used.

For differential gene expression analysis, featureCounts from the subread package v1.5.3 (-t gene -Q 3 -s 1) (Liao et al, 2014) followed by DESeq2 analysis 1.30.1 (Love et al, 2014) were used. For the DESeq2 run, genes with less than 10 reads were discarded and the condition was used for the design matrix. After the DESeq2 analysis, log-fold change shrinkage was applied using the DESeq2 built-in 'normal' mode.

For differential repeat expression, the trimmed fastq files were processed by the noncoding-RNA-seq module from snakePipes v2.5.1 (Bhardwaj et al, 2019). In brief, the noncoding-RNA-seq module used TEtranscript (Jin et al, 2015) to estimate the abundance in repeat families followed by the snakePipes implementation of the DESeq2 analysis.

## CUT&Tag processing

The CUT&Tag data was processed as in (Ibarra-Morales et al, 2021). Here, snakePipes version 2.4.0 (Bhardwaj et al, 2019) (parameters: --trim --fastqc --properPairs –dedup --mapq 1) with specific CUT&Tag Bowtie2 alignment option, as reported in (Kaya-Okur et al, 2019) (--local --very-sensitive-local --no-discordant --no-mixed -I 10 -X 700) was used. Since the CUT&Tag contains lambda phage spike-ins for reliable quantification of global effects, the libraries were mapped to a constructed hybrid genome of dm6 and lambda phage (NCBI GenBank ID: J02459.1). MACS2 (Zhang et al, 2008) was used to call CUT&Tag peaks using the following options: -g dm -q 0.05 --broad. CUT&Tag normalized signals were generated using bamCoverage from (Ramirez et al, 2016). More

specifically, the relative dm6/lambda total number of reads was used as normalization factor to rescale each sample through the bamCompare option --scaleFactor. For samples with H3 CUT&-Tag, the ratio between normalizations factors (H3 vs mark-specific CUT&Tag) was passed to bamCompare's option –scaleFactor to account for differences in starting material. When comparing CUT&Tag across different stages (specifically, before cycle 9, cycle 9–13 and cycle 14) we estimated the number of nuclei to account for differences in the starting material. The exact number of nuclei used for the normalization can be found at Dataset EV5. The coverage heatmaps were created using plotHeatmap from deep-Tools (v.2.5.7). To quantify the signal under the peaks, we integrated the normalized signal in a $+/-1$kb around each peak. The log2-ratio between the normalized signal under the peaks was used to quantify the changes between control and mutant embryo.

To quantify loss of HP1 at the pericentromeric region, we integrated the lambda and H3 normalized signal at the PCH in control and TM and calculated the ratio between the integrated signals. This resulted in ~35% decrease in HP1a when H3K9me3 is absent.

The code used for the H3 and lambda normalization have been placed on Github https://github.com/zhanyinx/atinbayeva_paper_2023.

## Repeats and PCH analysis

To quantify the statistics of repeats overlapping with H3K9me3 peaks at different developmental stage (before cycle 9, between cycles 9–13, and stage 5), we used UCSC repeatMasker table for repeats annotation. The following repeat families were excluded from the analysis: simple repeats, low complexity, artifact, satellite, other and unknown. The definition of pericentromeric regions was taken from (Zenk et al, 2021).

## Live-embryo imaging

As a control, flies that carried homozygous endogenously tagged HP1a-GFP and flies with P element insertion of H2Av-RFP were used for studying HP1a clustering and for counting the mitotic defects, respectively. Triple mutant flies were created as described above, but on top they were carrying HP1a-GFP or H2Av-RFP. F1 flies were collected within 15 h and aged for 3 days, and embryos were imaged for live imaging. Briefly, embryos were collected on an apple juice plate, dechorionated and mounted and imaged on Zeiss LSM 900 Airyscan 2 with a ×40/1.2 water immersion objective. For HP1a-GFP, the imaging has started from cycle 13 and time=0 s of cycle 14 was chosen when nuclei was circular shape after mitosis. The images were taken every 30 s with 30 z-stacks with a z resolution 0.5 μm. For HP1a-GFP, the imaging has started from cycle 13 and time=0 s of cycle 14 was chosen when nuclei was circular shape after mitosis. The images were taken every 30 s with 30 z-stacks with a z resolution 0.5 μm. For H2Av-RFP, the imaging hast started from cycle 13 interphase. The images were taken every 15 s with 10 z-stacks with a z resolution 1.2 μm.

## Live-embryo image analysis of HP1a-GFP

Before analyzing the images with Arivis vision4D software, all images were processed with airyscan processing software. Briefly, nucleus was segmented by using the total intensity threshold with

more than 99% and filtering the nuclei with a size threshold bigger than 50 μm³ to remove small non-nuclei objects. Then, HP1a foci were segmented by using the total intensity threshold with more than 95% and filtering the foci with a size threshold bigger than 0.0125 μm³ which corresponds to 5 voxels (0.05 μm × 0.05 μm × 0.5 μm). At the end, the compartmentalization was performed to find out to which nucleus a given cluster belongs to. The cluster number was determined by counting the numbers of clusters coming from each nucleus. Then, for each time point the mean and ± standard deviation of all normalized cluster number was plotted. For each nucleus, the cluster size was calculated by normalizing the cluster area to the area of the nucleus. Then, for each time point the mean and ± standard deviation of all normalized cluster size was plotted. For each nucleus, the cluster maximum intensity was calculated by normalizing the cluster maximum intensity to the average intensity of the nucleus. Then, for each time point the mean and ± standard deviation of all normalized cluster maximum intensity was plotted. For control, three embryos with approximately on average 70 nuclei per view per embryo were used. For triple mutant, four embryos with approximately on average 40 nuclei per view per embryo were used.

## Live-embryo image analysis of H2Av-RFP

Before analyzing the images, all images were processed with airyscan processing software. The number of total nuclei per embryo for mitosis was counted when chromosomes were at a metaphase plate. Then, the number of mitotic defects was counted during the division. After 3.75 min (15 time points) the number of daughter nuclei was counted per embryo. Then the number of defects at cytokinesis was counted. The number of apoptotic cells was counted within 15 min after nuclei were divided, as control embryos showed apoptosis within this time interval. Then, the fraction of mitotic and cytokinesis defects was counted per embryo by diving the number of occurring mitotic defect to the total number of nuclei at the metaphase plate or cytokinesis defects to the total number of daughter cells, respectively. Mean with ±standard deviation was plotted. For control 13 embryos with approximately on average 26 nuclei per view per embryo at metaphase and average 49 nuclei per view per embryo after division were used. For triple mutant 15 embryos with approximately on average 17 nuclei per view per embryo at metaphase and average 23 nuclei per view per embryo after division were used.

## Modeling

The model used to describe the interplay between HP1, H3K9 methylation and formation of aggregates of HP1 describes chromosome as a sequence of units of 10 kbp resolution. Each unit can be either methylated or not, it can be either bound by HP1 or not and can build spatial contacts with up to $n_{val} = 5$ other units.

We defined "HP1 binding sites" using H3 and spike-in normalized CUT&Tag data. In particular, we fit the distribution of normalized HP1 CUT&Tag signal at 10 kb resolution with a Gaussian function and consider as binding sites all sites whose signal is larger than the average plus 4 times the standard deviation.

The dynamics of the system is simulated with a Gillespie algorithm. At time zero all sites are free of HP1 and there are no contacts among them; HP1 binding sites are initially methylated in the control. In the TM, no methylation is present initially.

The random events that the system can undergo are:

1. Random methylation of a site with rate $k_m = 3.4 \times 10^{-6}$ min⁻¹.
2. Random demethylation of a site, if it is not bound to HP1, with rate $k_d = 0.34$ min⁻¹.
3. Binding of a site to HP1 with rate $k_b = 0.27$ min⁻¹.
4. Unbinding of HP1 from a HP1 binding site, if it is not involved in a contact, with rate $k_u = 0.034$ min⁻¹.
5. Unbinding of HP1 from a site that is not a HP1 binding site, if it is not involved in a contact, with rate $k_y = 3.4$ min⁻¹.
6. If a site is methylated and bound to HP1, it can spread its methylation state to those that are in contact with it with rate $k_n = 0.51$ min⁻¹.
7. Any pair of sites can form a contact with rate $k_c/(L_{ij})^\alpha$, where $L_{ij}$ is the minimum number of links (either along the chain or in space) that connect sites i and j, $\alpha = 1.5$ is an exponent that accounts for the contact probability in a polymer and $k_c = 0.1$ min⁻¹.
8. If two sites are in contact, both are methylated and bound to HP1, they can break the contact with rate $k_w = 0.034$ min⁻¹.
9. Otherwise, if two sites are in contact but some of them is not methylated or bound to HP1, they can break the contact with rate $k_z = 0.69$ min⁻¹.

These rates are obtained first by setting $k_d = 1$ in arbitrary time units and varying the other rates to optimize the agreement between the calculated and the experimental patterns of HP1 and methylation levels (cf. Fig. 5G). We then rescaled all rates (by a factor 0.34 min⁻¹) so that the simulated kinetics of HP1 aggregates matches the best the observed experimentally (cf. Fig. 5J). All simulations are performed in triplicates for ~100 h. All the calculations are performed on the last 5 Mbp of chromosome 2 L, corresponding to 500 sites. We assign the value 1 to methylated sites and 0 to unmethylated sites; the average value over time in the triplicate is defined the methylation probability. A similar procedure gives the HP1 binding probability. To visually compare the methylation and HP1 binding probabilities of sites with the experimental data, we rescaled the CUT&Tag signals by a factor which minimizes the square differences between the normalized experimental data and simulations probabilities. The normalization factors are 59 and 13 for methylation (or binding) probability is larger than 0.5 in the simulations. The accuracy of these simulated binding loci was compared with the CUT&Tag peaks called using MACS2. The *P* values are calculated by randomly reshuffling 10,000 times the simulated binding sites. In the algorithm to calculate $L_{ij}$ in point 7 of the algorithm, to make the calculation faster we approximate $L_{ij}$ assuming that only a single spatial contact can shortcut the actual genomic distance between two sites. The aggregates of HP1 are calculated from the contact map between sites, that is a matrix whose elements are 1 if two sites are in contact and zero otherwise. We define an aggregate the giant component of the contact map corresponding to sites bound to HP1. From the states of the system generated in the triplicate, we obtained the distribution of aggregate sizes in terms of number of HP1 molecules. Aggregates with size smaller than 2 are filtered out. Aggregate sizes are then rescaled by a factor 0.096, which optimizes the match with the experimental data (cf. Fig. 5J).

# Data availability

Mass spectrometry proteomics data have been deposited to the ProteomeXchange Consortium via the PRIDE partner repository with the dataset identifier PXD045480. CUT&Tag data can be found https://www.ncbi.nlm.nih.gov/geo/query/acc.cgi?acc=GSE240507 with an accession code mpqloucovzinzwb. Gro-seq data can be found https://www.ncbi.nlm.nih.gov/geo/query/acc.cgi?acc=GSE240508 with an accession code qjynoeiofbezrah.

The source data of this paper are collected in the following database record: biostudies:S-SCDT-10_1038-S44318-024-00127-z.

# Code availability

The code used to perform the simulations can be found at https://github.com/guidotiana/modellinoHP1. The code used for the H3 and lambda normalization have been placed on Github: https://github.com/zhanyinx/atinbayeva_paper_2023.

# Peer review information

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

## Acknowledgements

The authors are also very thankful for Chanseok Shin and Hideki Yoshida for kindly providing us with aliquots of dSetDB1 and G9a antibodies, respectively. The authors are grateful to our colleagues Julius Brennecke and Pierre Spierer for kindly sharing Su(var)3-9 and G9a knockout flies, respectively. We are thankful to Angela Andersen for carefully reading the manuscript; The Iovino lab members, particularly Melanie Schaechtle and Jule Friehs, for the help with CRISPR fly generations, Marina Mazina for her advice with protocols, Miao Yingjing for her feedback on the manuscript and the rest of the current and previous Iovino lab members for their critical thoughts about the work and constant support, the Imaging facility, the Fly facility, the Bioinformatics and Sequencing facilities at the Max Planck Institute of Immunobiology and Epigenetics (MPI-IE). We thank Thomas Jenuwein and Jenuwein lab members for the insightful discussion and for providing feedback on the work. The Bloomington *Drosophila* Stock Center (NIH P40OD018537) and the Transgenic RNAi Project at Harvard Medical School (NIH/NIGMS R01-GM084947) provided some fly stocks used in this study. NA is supported by the Max Planck Society and IMPRS program. NI is supported from the Max Planck Society; DFG:CRC992, Project B06; Behrens-Weise Stiftung; CIBSS - EXC 2189; Deutsche Forschungsgemeinschaft—Project ID 192904750—CRC992 Medical Epigenetics. Also, this project has received funding from the European Research Council (ERC) under the European Union's Horizon 2020 research and innovation programme (grant agreement No.819941) ERC CoG, EpiRIME.

## Author contributions

**Nazerke Atinbayeva**: Conceptualization; Data curation; Methodology; Writing—original draft; Writing—review and editing. **Iris Valent**: Data curation. **Fides Zenk**: Conceptualization. **Eva Loeser**: Methodology. **Michael Rauer**: Data curation. **Shwetha Herur**: Methodology. **Piergiuseppe Quarato**: Data curation; Methodology. **Giorgos Pyrowolakis**: Methodology. **Alejandro Gomez-Auli**: Data curation. **Gerhard Mittler**: Data curation. **Germano Cecere**: Data curation. **Sylvia Erhardt**: Data curation. **Guido Tiana**: Data curation; Methodology. **Yinxiu Zhan**: Conceptualization; Data curation; Software; Formal analysis; Supervision; Visualization; Methodology; Writing—original draft; Writing—review and editing. **Nicola Iovino**: Conceptualization; Supervision; Funding acquisition; Writing—original draft; Writing—review and editing.

Source data underlying figure panels in this paper may have individual authorship assigned. Where available, figure panel/source data authorship is listed in the following database record: biostudies:S-SCDT-10_1038-S44318-024-00127-z.

## Funding

## Disclosure and competing interests statement

The authors declare no competing interests.

# Expanded View Figures

**Figure EV1. H3K9me2/3 signals are specific.**

(**A**) Antibody for H3K9me2 and 4',6-diaminido-2 phenylindole (DAPI) staining of early embryos at different developmental cycles. For clarity, mitotic chromosomes for some cycles are shown. *n* is 3 embryos per genotype and per stage. Scale bars, 10 μm. (**B**) Heatmaps of H3K9me3 at different developmental stages of early embryogenesis: before cycle 9, cycle 9–13 and ZGA (cycle 14) clustered into pericentromeric regions (PCH) and chromosome arms (chr_arms) of ctr and TM. The last column represents the density of repeats at H3K9me3 ZGA peaks. The signal is ±10 kb centered on H3K9me3 peaks at cycle 14 embryos and ranked by signal intensity at cycle 14. Mean signal of two biological replicates is shown. (**C**) Heatmaps of H3K9me2 at different developmental stages of early embryogenesis: before cycle 9, cycle 9–13 and ZGA (cycle 14) clustered into PCH and chr_arms of ctr and TM. The last column represents the density of repeats at H3K9me3 cycle 14 peaks. The signal is ±10 kb centered on H3K9me2 peaks at cycle 14 embryos and ranked by signal intensity at cycle 14. Mean signal of two biological replicates is shown. (**D**) Quantification of H3K9me3 (left) and H3K9me2 (right) peak number at PCH and chr_arms of before cycle 9, cycle 9–13 and cycle 14 ctr embryos. (**E**) The fraction of repetitive elements enriched for H3K9me3 peaks at different developmental stages of early embryogenesis: before cycle 9, cycle 9–13 and cycle 14. The fraction was calculated by dividing the number of repeats with H3K9me3 signal at each stage to total number of repeats (~37,480). See also Dataset EV1. (**F**) Boxplots of H3K9me3 (left) and H3K9me2 (right) peak size at PCH and chr_arms at before cycle 9, cycle 9–13 and cycle 14. The boxplot indicates interquartile range from 1st (Q1) and 3rd (Q3) quartile, whiskers denote 1.5 times interquartile region (IQR) below Q1 and above Q3. Dots represent outliers. (**G**) Quantification of nucleotide coverage of H3K9me3 signal (left) and H3K9me2 signal (right) at PCH and chr_arms of before cycle 9, cycle 9–13 and cycle 14 ctr embryos. (**H**) IF as in Fig.1H with additional DAPI staining. Antibody for GFP for endogenously tagged dSetDB1 and Su(var)3-9 with GFP, antibody for G9a and DAPI staining of early embryos at different developmental cycles. Insets on the right show a magnification of the signals at cycle 14. n is 3 embryos per genotype and per stage. Scale bars, 10 μm.

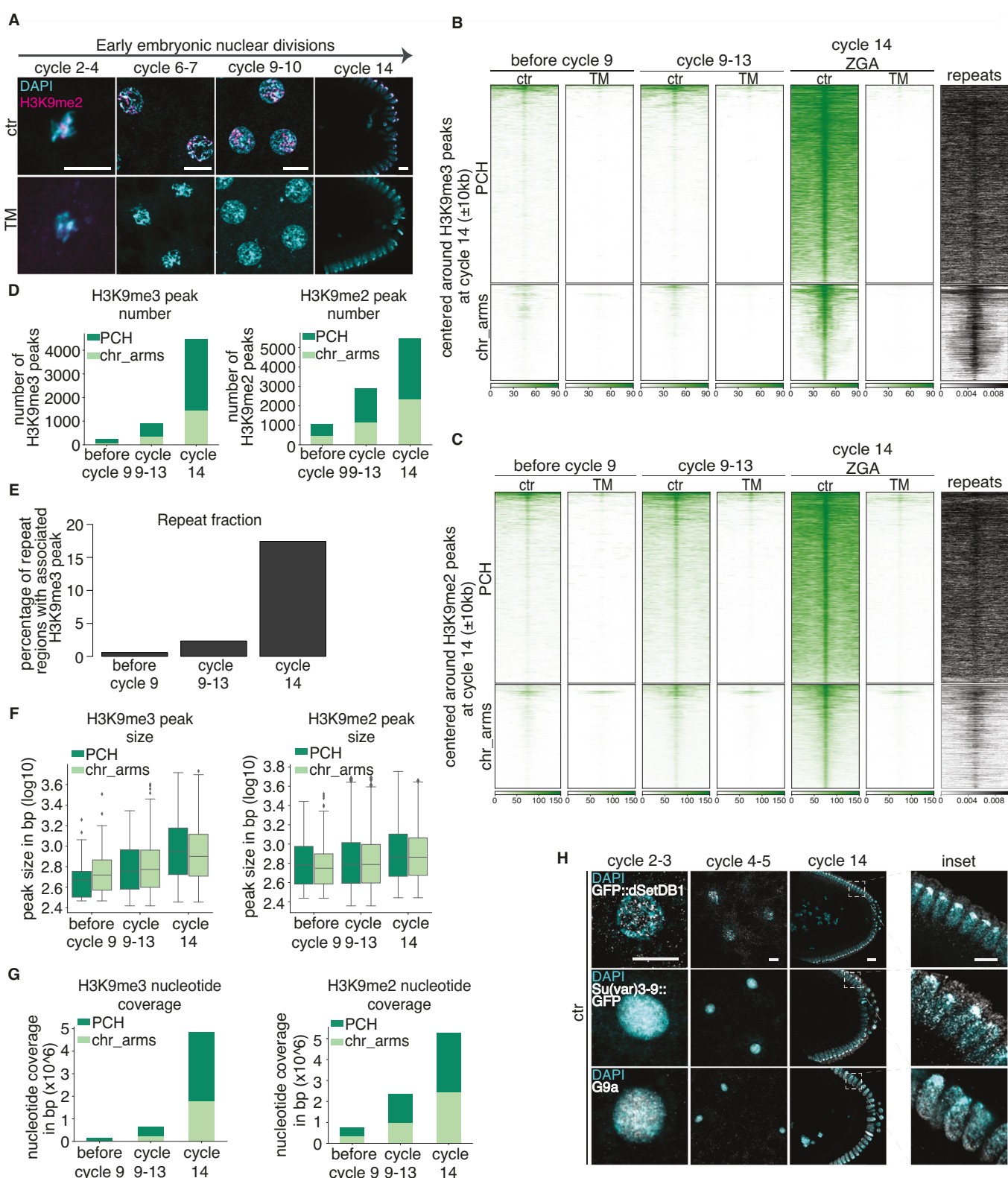

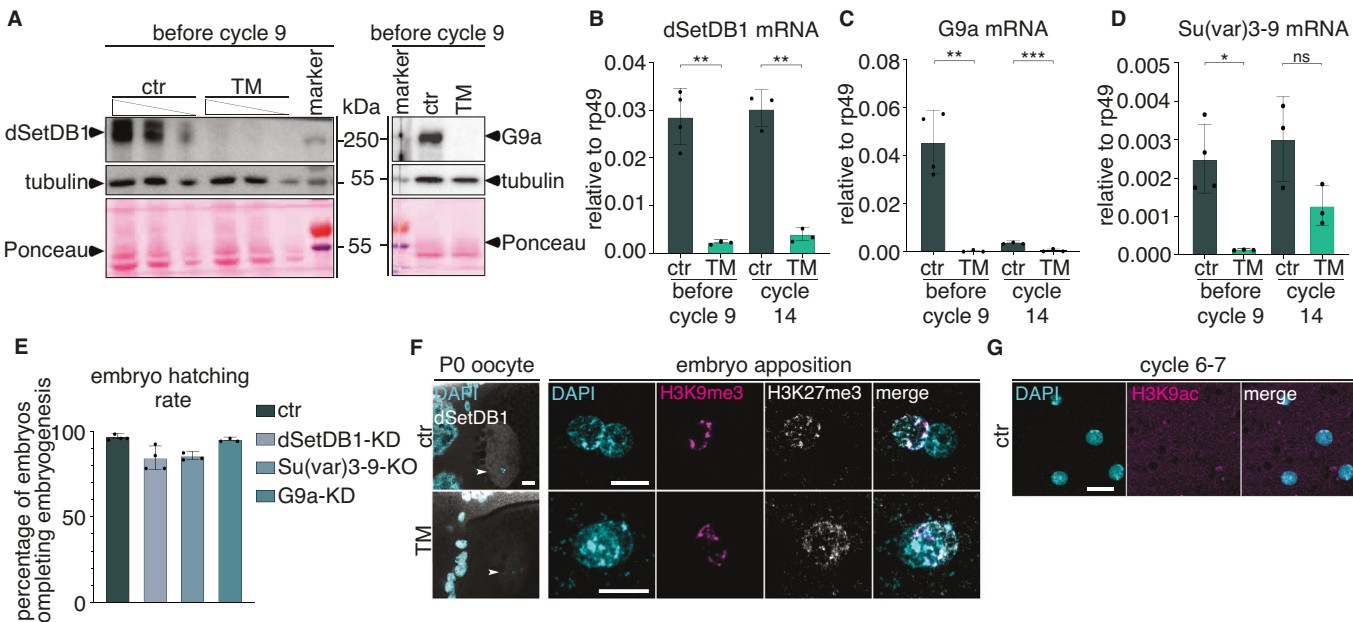

**Figure EV2. TM is devoid of all H3K9 HMTs.**

(A) Left panel: western blot for dSetDB1 on different amount of protein extract from ctr and TM before cycle 9 embryos. Before cycle 9 embryos represent the amount of maternally loaded proteins. Right panel: western blot for G9a on protein extract from ctr and TM before cycle 9 embryos. Tubulin and Ponceau staining are used as loading controls. See also Dataset EV2. (B) dSetDB1 mRNA levels relative to rp49 in ctr and TM before cycle 9 and cycle 14 embryos measured by RT-qPCR. Before cycle 9 embryos represent the amount of maternally loaded mRNAs. $n = 3$–$4$ biological replicates were used for each genotype at each time point (for before cycle 9 embryo, $P$ value = 0.0029, two-tailed Welch two-sample $t$ test, for cycle 14 embryo, $P$ value = 0.0033, two-tailed Welch two-sample $t$ test). Each dot represents one biological replicate. Shown mean $+/−$ s.d. (C) G9a mRNA levels relative to rp49 in ctr and TM before cycle 9 and cycle 14 embryos measured by RT-qPCR. $n = 3$–$4$ biological replicates were used for each genotype at each time point (for before cycle 9 embryo, $P$ value = 0.0061, two-tailed Welch two-sample t test, for cycle 14 embryo, $P$ value = 0.0003, two-tailed Welch two-sample $t$ test). Each dot represents one biological replicate. Shown mean $+/−$ s.d. (D) Su(var)3-9 mRNA levels relative to rp49 in ctr and TM before cycle 9 and cycle 14 embryos measured by RT-qPCR. $n = 3$–$4$ biological replicates are used for each genotype at each time point (for before cycle 9 embryo, $P$ value = 0.0136, two-tailed Welch two-sample $t$ test, for cycle 14 embryo, $P$ value = 0.0963, two-tailed Welch two-sample $t$ test). Each dot represents one biological replicate. Shown mean $+/−$ s.d. (E) Hatching rate of ctr and single mutant embryos: G9a-KO, Su(var)3-9-KO and dSetDB1-KD. $n = 4$ biological replicates with total 480 embryos were used for ctr and dSetDB1-KD and $n = 3$ biological replicates with total 360 embryos were used for Su(var)3-9-KO and G9a-KO. Each dot represents one biological replicate. Shown mean $+/-$ s.d. (F) Left panel: dSetDB1 and DAPI staining of ctr and TM stage 10 egg chamber oocyte. The arrowhead indicates the oocyte. Right panel: H3K9me3, H3K27me3 and DAPI staining of ctr and TM pronuclei at apposition in the zygote. To define the maternal pronucleus H3K27me3 was used as it stains the maternal pronucleus (Zenk et al, 2017). $n$ is at least three embryos per genotype and per stage. Scale bars, 10 µm. (G) H3K9ac and DAPI staining of ctr cycle 6–7 embryos. $n$ is at least 2 embryos. Scale bars, 10 µm.

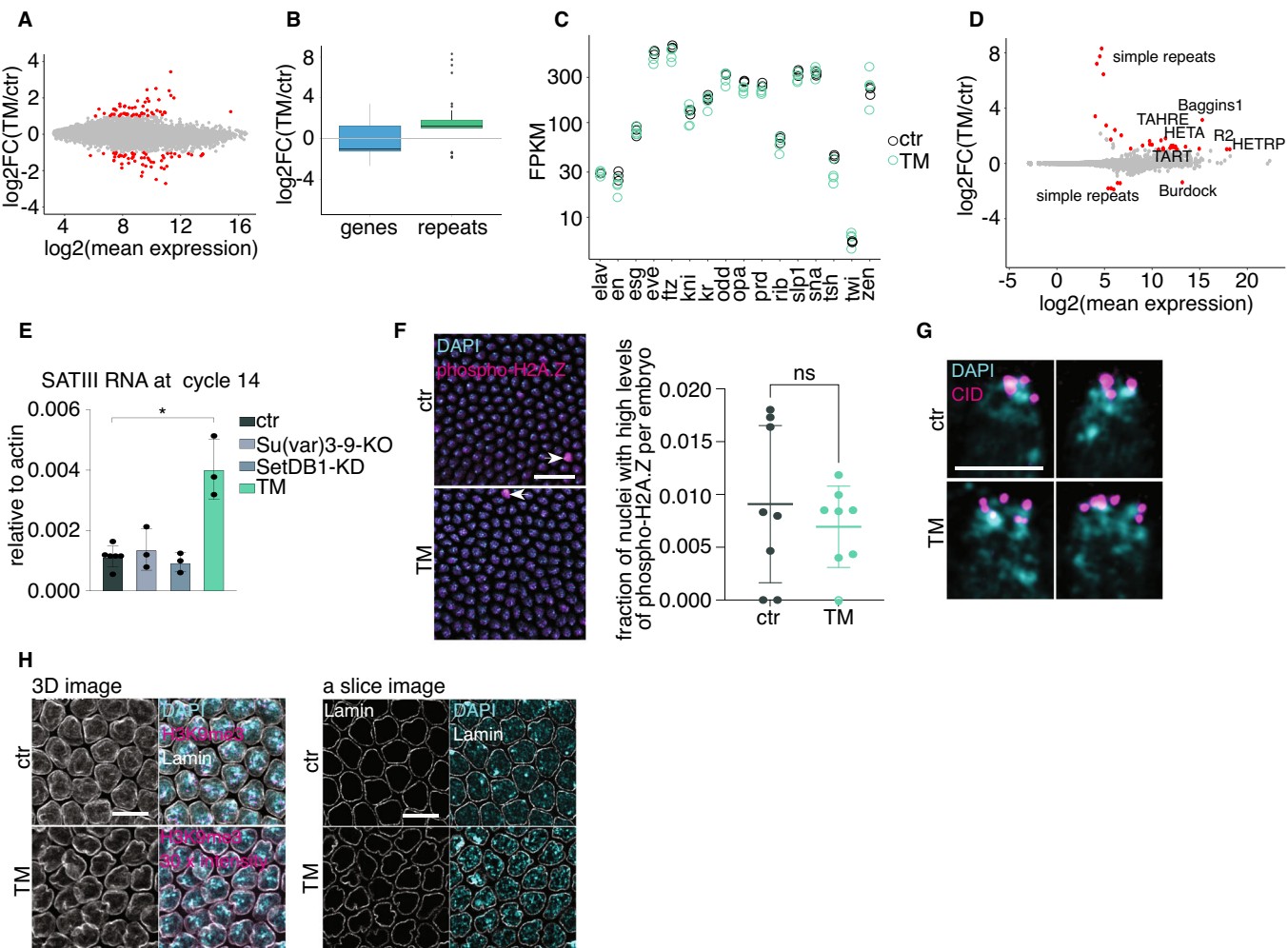

**Figure EV3.   Absence of H3K9me2/3 does not affect gene expression significantly, but some repeats.**

(A) MA plot of differential gene expression in TM versus ctr cycle 14 embryos determined by Gro-seq. $n = 3$ biological replicates for ctr and $n = 4$ biological replicates for TM were used for quantification. See also Dataset EV3. (B) Boxplots showing the distribution of expression changes for genes and repeats in TM versus ctr cycle 14 embryos. The boxplot indicates interquartile range from 1st (Q1) and 3rd (Q3) quartile, whiskers denote 1.5 times interquartile region (IQR) below Q1 and above Q3. Dots represent outliers. See also Dataset EV3. (C) Normalized counts of 16 pure zygotic genes in TM versus ctr cycle 14 embryos similar to (Zenk et al, 2021) obtained by Gro-seq. See also Dataset EV3. (D) MA plot of differential repeat expression in TM versus ctr cycle 14 embryos from Gro-seq data. See also Dataset EV3. (E) SATIII mRNA levels relative to actin in ctr, Su(var)3-9-KO, dSetDB1-KD and TM cycle 14 embryos measured by RT-qPCR ($P$ value $= 0.0314$, two-tailed Welch two-sample $t$ test). $n = 6$ biological replicates for ctr and n $= 3$ biological replicates were used for the rest. Each dot represents one biological replicate. Shown mean $+/-$ s.d. (F) Left panel: antibody for phosphorylated H2A.Z (phospho-H2A.Z) and DAPI staining of cycle 14 ctr and TM embryos. An arrow represents a nucleus with high levels of phospho-H2A.Z. Scale bars, 20 μm. Right panel: quantification of fraction of nuclei with high phospho-H2A.Z staining normalized by total number of nuclei per embryo in ctr and TM cycle 14 embryos ($P$ value $= 0.4856$, two-tailed Welch two-sample $t$ test). $n = 8$ embryos were used for quantification for each genotype. (G) Representative images of CID and DAPI staining of cycle 14 ctr and TM embryos. Scale bars, 5 μm. (H) Representative images of Lamin, H3K9me3 and DAPI staining of cycle 14 ctr and TM embryos. H3K9me3 signal intensity in TM embryos was shown 30 times more than H3K9me3 signal intensity of ctr embryos to see the leftover H3K9me3 signal. Left panel: 3D image of assembled Z-stacks. Right panel: a single slice image. $n = 2$ biological replicates with 9 embryos were used for quantification for each genotype. Scale bars, 10 μm.

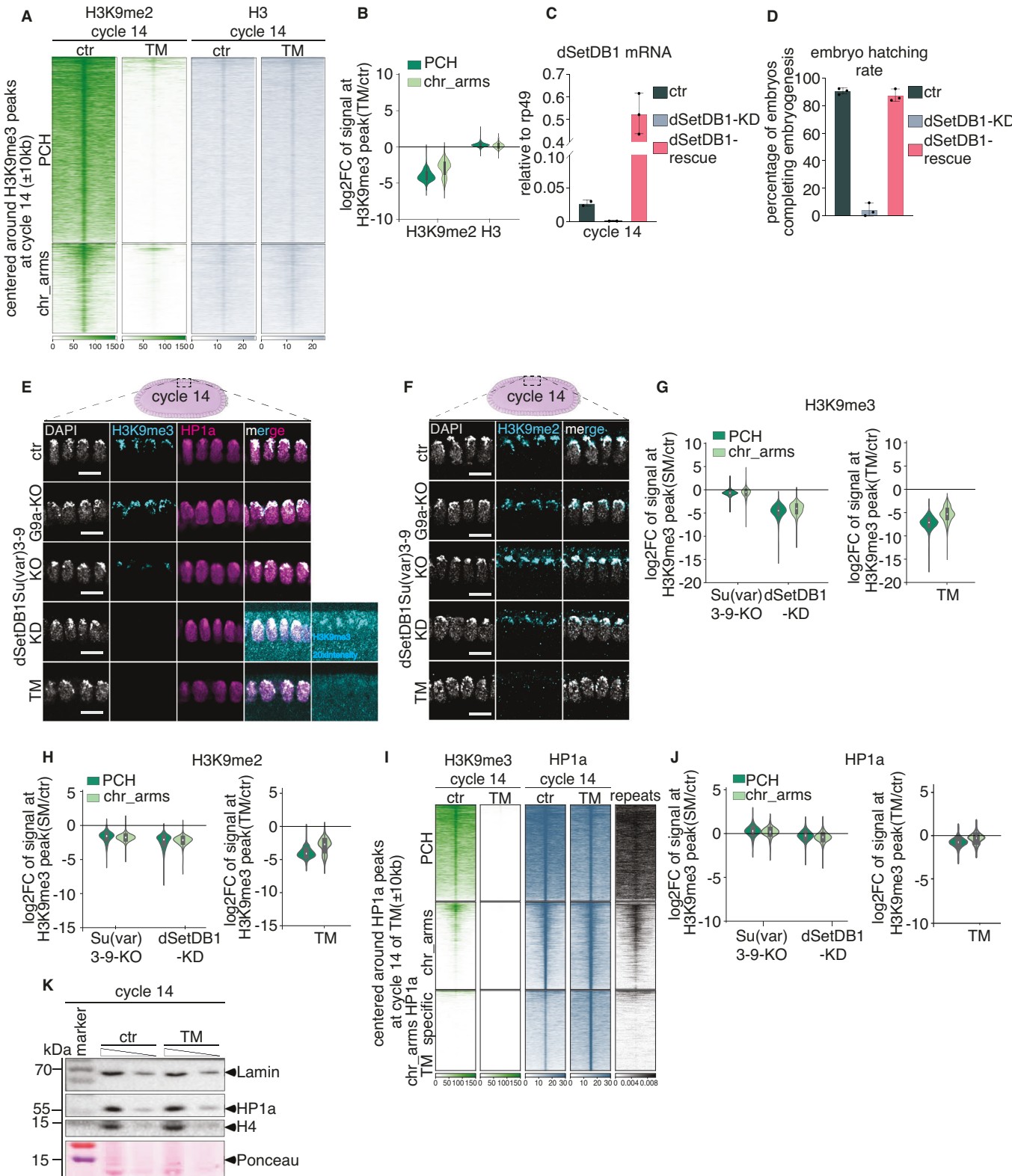

◀ **Figure EV4. H3K9me2/3 signals are absent only in triple mutant embryos.**

(A) Heatmaps of H3K9me2 and H3 at ZGA (cycle 14) clustered into PCH and chr_arms. The signal is ±10 kb centered on H3K9me3 peaks at cycle 14 ctr embryos and ranked by signal intensity at cycle 14 ctr. Mean signal of two biological replicates is shown. (B) Violin plot of log2 fold change of H3K9me2 and H3 signal on H3K9me3 peaks found in PCH and chr_arms regions in TM cycle 14 embryos compared to ctr embryos. The boxplot inside indicates interquartile range from 1st (Q1) and 3rd (Q3) quartile, whiskers denote 1.5 times interquartile region (IQR) below Q1 and above Q3. (C) dSetDB1 mRNA levels relative to rp49 in ctr, dSetDB1-KD and dSetDB1 rescue cycle 14 embryos measured by RT-qPCR. $n = 3$ biological replicates were used for each genotype. In (C, D) early oogenesis maternal tubulin driver (see "Methods") was used. Each dot represents one biological replicate. Shown mean $+/-$ s.d. (D) Hatching rate of ctr, dSetDB1-KD and dSetDB1 rescue embryos. $n = 3$ biological replicates with total 360 embryos were used for each genotype. Each dot represents one biological replicate. Shown mean $+/-$ s.d. (E) H3K9me3, HP1a and DAPI staining of ctr, G9a-KO, Su(var)3-9-KO, dSetDB1-KD and TM cycle 14 embryos. For dSetDB1-KD and TM cycle 14 embryos, H3K9me3 staining intensity was increased 20 times more than in ctr to see the leftover H3K9me3 signal. $n$ is at least 4 embryos per genotype. Scale bars, 10 μm. (F) H3K9me2 and DAPI staining of ctr, G9a-KO, Su(var)3-9-KO, dSetDB1-KD and TM cycle 14 embryos. $n$ is at least 4 embryos per genotype. Scale bars, 10 μm. (G) Violin plot of log2 fold change of H3K9me3 signal on H3K9me3 peaks found in PCH and chr_arms in single mutants: Su(var)3-9-KO and dSetDB1-KD and TM cycle 14 embryos compared to ctr embryos. The boxplot inside indicates interquartile range from 1st (Q1) and 3rd (Q3) quartile, whiskers denote 1.5 times interquartile region (IQR) below Q1 and above Q3. (H) Violin plot of log2 fold change of H3K9me2 signal on H3K9me3 peaks found in PCH and chr_arms in single mutants: Su(var)3-9 KO and dSetDB1-KD and TM cycle 14 embryos compared to ctr embryos. The boxplot inside indicates interquartile range from 1st (Q1) and 3rd (Q3) quartile, whiskers denote 1.5 times interquartile region (IQR) below Q1 and above Q3. (I) Heatmaps of H3K9me3 and HP1a at cycle 14 clustered into PCH, chr_arms and chr_arms with newly appearing HP1a peaks only in TM. The last column represents the density of repeats at H3K9me3 ZGA peaks. The signal is ±10 kb centered on HP1a peaks at cycle 14 TM embryos and ranked by H3K9me3 signal intensity at cycle 14 ctr. Mean signal of two biological replicates is shown. (J) Violin plot of log2 fold change of HP1a signal on H3K9me3 peaks found in PCH and chr_arms in single mutants: Su(var)3-9-KO and dSetDB1-KD and TM cycle 14 embryos compared to ctr embryos. The boxplot inside indicates interquartile range from 1st (Q1) and 3rd (Q3) quartile, whiskers denote 1.5 times interquartile region (IQR) below Q1 and above Q3. (K) Western blot for HP1a on different amount of protein extracts from ctr and TM cycle 14 embryos. Lamin, H4 and Ponceau staining are used as loading controls.

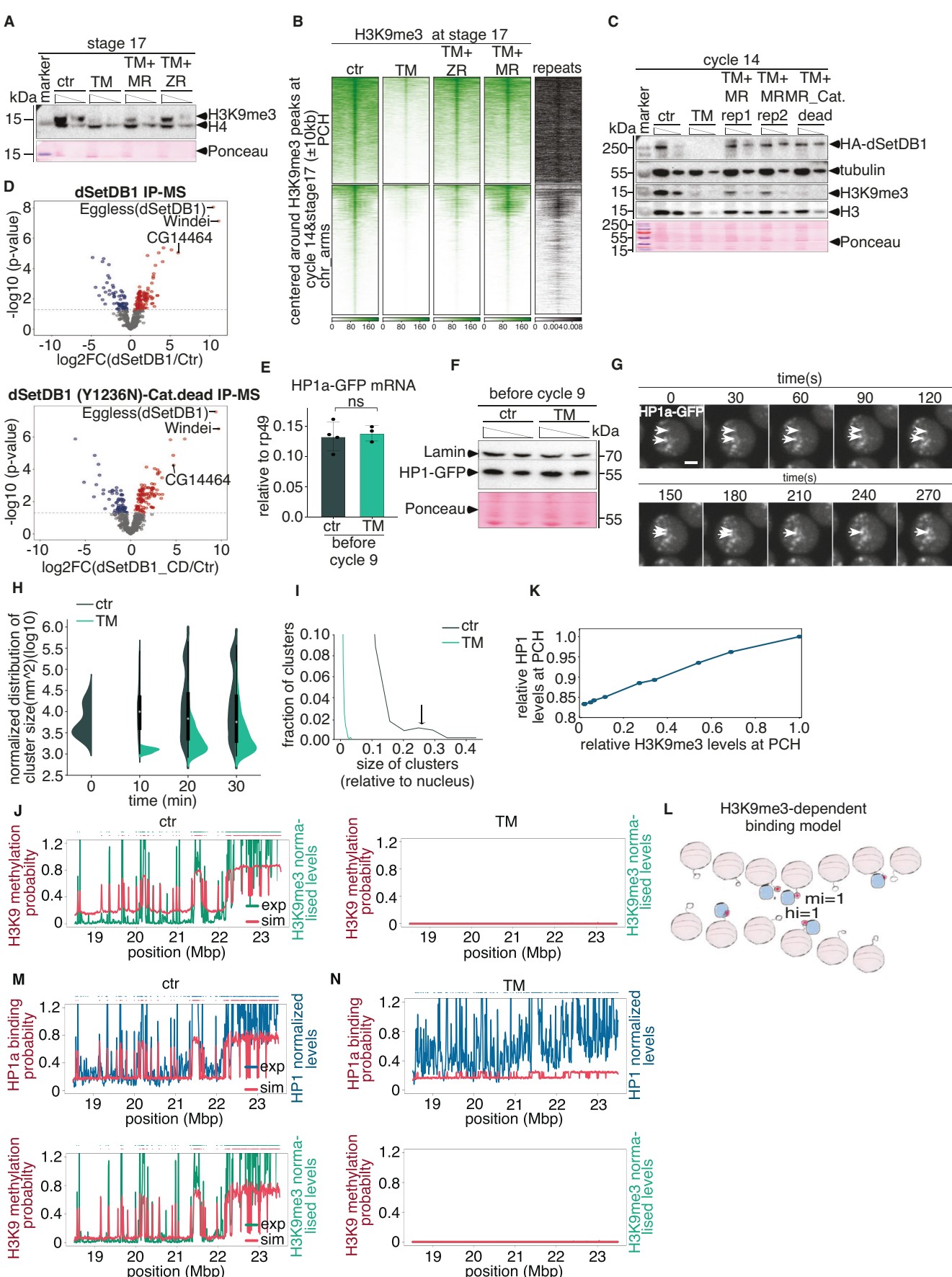

◀ **Figure EV5. Lack of H3K9me3 in prior and at ZGA cannot be rescued by zygotic expression of dSetDB1.**

(A) Western blot for H3K9me3 on different amount of protein extract from stage 17 ctr, TM, zygotic expression of dSetDB1 in the background of TM (TM + ZR) and maternal expression of dSetDB1 in the background of TM embryos (TM + MR). H4 and Ponceau staining are used as loading controls. (B) Heatmaps of H3K9me3 at stage 17 embryos clustered into PCH and chr_arms. The last column represents the density of repeats at H3K9me3 ZGA and stage 17 peaks. The signal is ±10 kb centered on H3K9me3 peaks at cycle 14 and stage 17 ctr embryos and ranked by signal intensity at stage 17 ctr embryos. Mean signal of two replicates is shown. (C) Western blot for flag-HA-dSetDB1 and H3K9me3 on different amount of protein extract from cycle 14 ctr, TM, maternal expression of wild-type dSetDB1 (TM + MR) and maternal expression of catalytic dead dSetDB1 in the background of TM embryos (TM + MR_cat.dead). H4 and Ponceau staining are used as loading controls. TM + MR is loaded as 2 biological replicates. (D) Volcano plot showing the enrichment of proteins:top: endogenously flag-HA-tagged wild-type dSetDB1 and bottom: endogenously flag-HA-tagged catalytic dead dSetDB1(Y1236N) over ctr embryos. Red and blue dots represent enriched and diminished proteins in the flag-HA-tagged IP compared to ctr embryos, respectively. $n = 3$ biological replicates were used for both ctr and wild-type and catalytic dead dSetDB1 IPs. (E) HP1a mRNA levels relative to rp49 in ctr HP1-GFP homozygous and TM HP1-GFP homozygous before cycle 9 embryos measured by RT-qPCR ($P$ value $= 0.7247$, two-tailed Welch two-sample $t$ test, for cycle 14 embryo). $n = 3$–4 biological replicates are used for each genotype. Each dot represents one biological replicate. Shown mean $+/-$ s.d. (F) Western blot on different amount of protein extract from before cycle 9 ctr and TM embryos that homogenously express endogenously tagged HP1a-GFP. Lamin and Ponceau staining are used as loading controls. (G) A single nucleus maximum projection of 30 z-stacks images of endogenously tagged HP1a with GFP for cycle 14 embryos for an interval of 30 s. 10 images correspond to a period of 4.5 min. Two arrows show the two clusters of HP1-GFP that over time fuse into one cluster. Scale bars, 2 μm. (H) The distribution of cluster size normalized to nucleus size for ctr ($n = 3$ embryos) and TM ($n = 4$ embryos) cycle 14 embryos. For clarity, only time points 0, 10, 20 and 30 min are shown. (I) Cluster size distribution in ctr and TM at late time points ( > 30 min). An arrow indicates the population with a bigger size. (J) Normalized H3K9me3 profile detected in CUT&Tag (green curve) and predicted by the model where H3K9me3 mediates only HP1-HP1 interactions (red curve) in ctr and TM. Sites with H3K9me3 are shown in the upper part of the panel for the simulation (red track) and in CUT&Tag (blue track, see Methods). See also Dataset EV4. (K) Fraction of HP1 bound to PCH relative to the fraction detected with the best model from Fig. 5G as a function of methylation rate. (L) A schematic representation of the model where H3K9me3 is important for HP1 binding to the chromatin. Each locus (pink bead) can undergo H3K9 methylation (red circle) and bind HP1 (blue shape). (M) Top panel: normalized HP1 binding profile detected in CUT&Tag (blue curve) and predicted by the model where H3K9me3 mediates only HP1 binding to chromatin (red curve) in ctr. Bottom panel: normalized H3K9me3 profile detected in CUT&Tag (green curve) and predicted by the model where H3K9me3 mediates only HP1 binding to chromatin (red curve) in ctr. Bound sites and sites with H3K9me3 are shown in the upper part of the panel for the simulation (red track) and in CUT&Tag (blue track, see "Methods"). (N) as in M for TM.

