## [Peer Review File · The EMBO Journal]

Inheritance of H3K9 methylation regulates genome architecture in *Drosophila* early embryos

Nicola Iovino, Nazerke Atinbayeva, Iris Valent, Fides Zenk, Eva Loeser, Michael Rauer, Shwetha Herur, Piergiuseppe Quarato, Giorgos Pyrowolakis, Alejandro Gomez-Auli, Gerhard Mittler, Germano Cecere, Sylvia Erhardt, Guido Tiana, and Yinxiu Zhan

Corresponding author(s): Nicola Iovino (iovino@ie-freiburg.mpg.de), Yinxiu Zhan (yinxiu.zhan@ieo.it)

Review Timeline:

Submission Date:	28th Feb 24
Editorial Decision:	13th Mar 24
Revision Received:	27th Mar 24
Accepted:	15th Apr 24

Editor: Cornelius Schneider

Transaction Report:

The initial review process for this manuscript took place with another journal. The initial reviewers' comments and authors' responses for this article have been made available.

Point by point response:

Inheritance of H3K9 methylation regulates genome architecture in early embryos (Nature Genetics manuscript NG-A61658)

We would like to thank all the Reviewers for their useful and constructive comments. In this revised version of the manuscript, we have responded to all the Reviewer's suggestions and comments. We believe that their review has led to a considerable improvement in the manuscript.

In the revised manuscript, we have addressed the following major requests:

1. Complete new data figure on dissecting cell shape, mitotic defects, and DNA damage in triple mutant (TM) embryos
2. Contextualize the significance of the work in the light of mammalian work
3. Test HP1 RNA binding and identify potential cofactor of dSetDB1 by mass spectrometry
4. Check HP1 cluster dynamics
5. Disentangle the role of transposon upregulation.
6. Improve the visualization of the figures

The manuscript now includes the following:

- 1) Quantification of mitotic defects in control and TM embryos by live imaging of H2Av-RFP shown in **new figure 3a**. Moreover, H2A γ staining and quantification of the staining in **new Extended Data figure 3f** show that there is no increased DNA damage at ZGA in TM embryos compared to the control embryos. However, there are increased mitotic defects in TM embryos. Mainly, a significant proportion of TM embryos fail to condense their chromatin during metaphase, arrest during mitosis, and fuse, leading to polyploidy.
- 2) New data on nuclear roundness and sphericity of ZGA nuclei quantified by using Lamin staining in **new figure 3g and Extended Data figure 3i** show that a significant portion of TM nuclei have lower sphericity and roundness, suggesting that H3K9me is important for promoting proper nuclear shape.
- 3) Time-lapse images of HP1a-GFP from the live imaging are shown in **new Extended Data figure 5g** to represent the fusion of HP1 clusters as well as **new Extended Data figure 5h** to show the distribution of each HP1 cluster sizes at interphase in control and TM ZGA embryos. Notably, the bimodal distribution in control is lost in TM embryos with the disappearance of the large-sizes HP1 clusters.
- 4) New Hi-C experiment in control and TM ZGA embryos in **new Extended Data figure 3h** as an orthogonal approach to check the pericentromeric compaction. Hi-C results show that contact frequencies between pericentromeric regions across chromosomes are reduced significantly in TM compared to control embryos.
- 5) New data for the time an embryo takes from fertilization until the end of ZGA in control and TM embryos in **new figure 3b** shows that TM embryos take longer to reach the end of ZGA.

Reviewer#1:

Remarks to the Author:

In this manuscript, the authors present a characterization of the role of H3K9 methylation, its writers, and the associated protein HP1. They use imaging and genomics approaches to show that H3K9me is present on the genome earlier than previously reported and carried over from the meiotic oocyte as it's not actively erased like other epigenetic marks. They confirm that dSetDB1 is the main H3K9 HMT

among the three present in *Drosophila*, but show that there are redundancy/compensatory effects as removing any one of the three HMT is not sufficient to abolish H3K9me and that a triple mutant is required for this scope. Abolishing H3K9me deposition severely impacts embryo viability, not by affecting 'normal' gene expression but rather because of transcriptional derepression of repetitive elements and heterochromatin disorganization. Using rescue experiments the authors show these molecular phenotypes strictly depend on the catalytic activity of dSetDB1 and impact embryo viability during early development before ZGA. The authors next show that HP1 recruitment is only minimally dependent on pre-established H3K9me and can be independently recruited to DNA. Finally, the authors describe a biophysical model of HP1-H3K9me interactions which they use to show that H3K9me is instead required for HP1-mediated heterochromatin compaction. We advise seeking the opinion of a computational biophysicist on this last point to evaluate the technical aspects of the approach.

The concept of the paper is interesting, although some aspects have been shown in other species, this paper nicely brings things together (and is the first time it's shown in flies). Overall the paper is logically constructed and the claims are largely supported by the data shown. However, a number of the figures contain visualization choices that make it difficult to visually compare the different conditions shown and, in some cases, it impacts the interpretation of the data.

We are very grateful for the Reviewer's appreciation of our work and helpful and excellent comments.

- Prompted by the Reviewer's request, we realised that nuclei number differences represent an important source of variability across nuclear cycles. Therefore, we re-quantified the Cut and Tag heatmaps in **figure 1c** by also taking into account the average nuclei number at different stages together with the lambda spike-in. To the best of our knowledge, this quantification strategy is above the current state of the art. The new quantification shows that the fold difference between before cycle 9 and ZGA is around 10-fold difference (**figure 1c and e**).
- We gave the example of **HP1 cluster fusion** by showing the time-lapse images of HP1a-GFP from the live imaging (**Extended Data figure 5g**)
- We plotted the distribution of every HP1 cluster at interphase in control and TM ZGA embryos (**Extended Data figure 5h**). It shows that there is a **bimodal distribution** of HP1 cluster size in control embryos but not in TM embryos, suggesting that the large-sized HP1 clusters are H3K9me3 dependent.
- We modified the text and figures according to the Reviewer's comments
- We added the data in Excel formats
- We added the links for the bioinformatic codes used in this study

We believe that the reviewer's suggestions for the data representation as well as contextualizing the current work in the light of the work done in other species, have improved the current manuscript considerably. We hope that the Reviewer finds the new version of the manuscript as improved and exciting as we do.

1. Fig 1b, 1h and associated supplementary figures:

While we are aware that nuclear size dramatically shrinks during early development and the authors duly report scale bars in the pictures, the chosen magnifications make it difficult (if not impossible) to evaluate the claims in the text. In lines 79-81 authors state: "By ZGA, both Su(var)3-9 and dSetDB1 were mainly enriched in DAPI-dense regions, whereas G9a was also distributed along the chromosome arms (Fig. 1h and Extended Data Fig. 1g)." This is difficult to see in the NC5-10 images and impossible to tell in NC14 images. Nuclei from the different stages should be reported at similar sizes and readers can use scale bars to understand the relative size of the nuclei, which is not the point discussed. Alternatively, the authors should add higher magnification insets for both NC5-10 and NC14 as they do for NC14 in supplementary Figure 1g

We are grateful for the Reviewer's useful comment. We have now added magnified insets with the same scale for the images of all stages (**figure 1h**).

2. Fig 1c, 1g and heatmaps throughout the manuscript

Heatmaps are presented with different colour scales or with missing colour bars, preventing the reader from doing meaningful comparisons. Particularly in Fig1c the difference between pre-ZGA stages and NC14 looks relevant, but not nearly close to the probably >100 folds that can be deduced from the colour scales. We do acknowledge that displaying the three stages with the same scale would result in a non-discernible signal in pre-ZGA, so if authors want to keep the heatmap visualization they should:

- Use the same scale for NC<9 and NC9-13, as they are in a comparable range
- Either make the color scale different more conspicuous or mention the difference in the text when referring to figure 1c and 1e. In either case mention the color scale difference in the legend.
- Add a more quantitative reference, such as a profile plot next to the heatmap

We thank the Reviewer for her/his useful suggestions. We re-quantified the Cut and Tag heatmaps in **figure 1c** by also now taking into account the average nuclei number at different stages together with the spike-in. The new quantification shows that the fold difference between before cycle 9 and ZGA is around 10-fold difference (**figure 1 c and e**). Therefore, we have replotted the heatmaps with the same scale and added missing color bars in all heatmaps in this manuscript. We have also added profile plots on top of the corresponding heatmaps (**figure 1c**)

3. Fig 1d: While Venn diagrams can be effectively used to visualize overlaps in this case it does not transmit the difference of magnitude between the sets (>10 folds in NC14 vs NC9). An Euler diagram (proportional Venn) or, better, an upset plot could better serve the purpose.

We thank the reviewer for the nice suggestion. We have now replaced the Venn diagram with an upset plot, as the reviewer suggested (**figure 1d**).

4. Fig 1f: pre-ZGA tracks should be on the same scale, the scale of the ZGA track should be increased so as not to clip the signal. Differences in scale should be mentioned in the legend.

As we mentioned in point 2, we have put the scale the same across the stages and changed **figure 1f** with the same scale.

5. Lines 11-14 "H3K9me3 main function at this developmental stage is to drive HP1a clustering and subsequent heterochromatin compaction. Thus, HP1a binding to constitutive heterochromatin in the absence of H3K9me3 is dispensable for development and heterochromatin formation; H3K9me3-dependent genome organization is essential"

But that means that heterochromatin compaction is dispensable for development – is that really what you want to claim?

We apologize for misleading the reviewer. We have changed the text with the new text in Line 12-15 "H3K9me3 main function at this developmental stage is to drive HP1a clustering and subsequent heterochromatin compaction. Our results show that HP1a binding to constitutive heterochromatin in the absence of H3K9me3 is not sufficient to drive the proper embryo development and heterochromatin formation; H3K9me3-dependent genome organization is essential."

6. Lines 57-58 "...(CUT&Tag) in hand-sorted embryos before cycle 9 (totipotent), at cycle 9-13 (pluripotent) and at cycle 14 (ZGA)." What is the evidence for these designations, totipotent (before cycle 9) and then pluripotent at nuclear cycle 9-13? Reference the paper or remove this. I don't know of any evidence for this, and it is misleading. There are very precise definitions for this in the mammalian stem cell field, and no such tests of these cells have been done in flies, to the best of my knowledge

Nuclei transfer experiments showed that the early stages of *Drosophila* embryos are totipotent (Haigh et al., 2005, Genetics). When the nuclei extracted from 70-100 minutes old embryos that correspond to approximately before cycle 9 embryos were transferred to the donor embryo, the adult clones arose from the transferred nuclei meaning the nuclei at these stages were capable of generating all the tissues needed for the development of an adult fly. We have also added a new reference for the cell-fate determination of early *Drosophila* embryos (Hartenstein et al., 1985, Roux's Archives of Developmental Biology).

7. Lines 99-100: Authors discuss H3K9me localization but refer to Fig2b, which is a hatching assay. The correct panel (Fig2 c?) should be referred to.

We have now corrected it, so the H3K9me localization refers now to **figure 2c**.

8. Lines 101-103: authors state: "These data suggest that genomic regions containing H3K9me3 in the germline retained this modification throughout meiosis and transmitted this epigenetic memory to the zygote." While the definition of epigenetic memory is debated and outside the scope of both the manuscript and this review, it is conceivable that it should be represented by a type of active mechanism. In this case, it can be argued that the persistence of H3K9me3 in the oocyte is simply the result of a lack of DNA replication, as confirmed by the loss of the mark within a few nuclear divisions in the triple mutant embryos. The statement should be clarified or removed.

We thank the reviewer for pointing this out. We have removed the sentence, and we point out that the oocyte is not cycling in Lines 116-118 as follows "Using IF, we found that H3K9me3 was still enriched in the oocyte of both control and TM ovaries (Fig. 2c PO oocyte, arrowhead), in line with the fact that the oocyte, which is arrested in meiosis I and therefore non-cycling [41], does not require H3K9 HMT activity to retain H3K9 methylation on chromatin."

Related to epigenetic memory, we have now included and discussed a new manuscript from the lab of D. Moazed Lines 337-339.

9. Figure 2c – how do you know that that is the maternal pronucleus? It has to be either the paternal or maternal – but is this just a guess?

Line 100 reference to Fig 2b, should be Fig 2c

Previously, our lab (Zenk et al., 2017, Science) has shown that the maternal pronucleus is stained with H3K27me3 at apposition but not the paternal pronucleus (**Extended Data figure 2f**). We have now clarified this in the text as well as in the figure legends and have corrected the reference to **figure 2c**.

10. Fig 2c: nuclei size, see point 1

As the Reviewer pointed out in point 1, we see that the nuclei size shrinks between cycles 9 to ZGA (**figure 2c**). The size and scale of the images between cycle 5 to ZGA are the same, so we hope that the Reviewer can see the shrinkage of the nuclei. Since the H3K9me3 is mostly visible from mitotic chromosomes at cycles 5-7, we used mitotic nuclei. Therefore, we apologize that, at this stage, we can't show nuclei shrinkage.

11. Fig 3a: colour scales for TM samples are missing. If the same as ctrl samples and removed for aesthetic reasons mention it in the legend for clarity.

We thank the Reviewer for pointing this out. We have now added the color scales for TM as well for all the heatmaps to make it clearer.

12. Fig 3d, Myo81F: adjust scales of H3K9me2/3 to not clip signal

We have now adjusted the scales for the signal tracks for this figure.

13. Supplementary figure 4c: dots are overlapping and hiding the data point below, which prevents from understanding how many data points are presented for each genotype. Add some

transparency or plot as empty circles

We have now replaced the filled dots with the empty circles, as the Reviewer suggested.

14. Lines 205-207 Fig 5d (and 5j):

Authors state “Along the same line, the foci size and maximum intensity increase over time, further supporting the fusion of HP1 foci at late cycle 14 (Fig. 5d-e)”. While the mean value clearly increases, what is evident is a large increase in the variability of the measured values also toward smaller and not bigger clusters. Such an averaging prevents from telling whether there is a bi-(or multi-) modal distribution of the values, which would indicate the presence of different types of HP1 clusters with different behaviour. As the authors have live-imaging data available to show cluster fusion they should:

- Present the time laps data as the quantification of the area of each cluster (or some of them) followed over time and not as pooled averages

- Report examples of fusion behaviour of HP1 clusters

We thank the Reviewer for this very useful suggestion. As the Reviewer suspected, there is indeed a bimodal distribution of HP1 cluster size. We have re-analyzed our HP1-GFP live images in a way that we can assign a cluster to a nucleus in which the cluster is found in order to normalize the size of an HP1 cluster by the corresponding nucleus size. Therefore, figure 5c-e have been re-plotted with re-quantified data, and we would like to point out that the large-sized HP1 cluster disappear in TM embryos. Having said that, we have now plotted the distribution of each cluster size (the normalized cluster size) followed over time (**Extended Data figure 5h**). We have shown only four different time points for the sake of the clarity of the plot. Also, we have added time laps HP1-GFP images as an example to report the fusion of HP1 clusters (**Extended Data figure 5g**)

Important for data accessibility and reproducibility

15. Supplementary tables are provided as text in the manuscript PDF, making them effectively non-usable. In the final version, these should be provided as independent, parsable files.

We have now uploaded the supplementary tables in an Excel format.

16. A substantial amount of the conclusions of the manuscript come from the analysis of genomics data or biophysical modelling. A brief explanation of the analysis process is provided in the methods section but there is no mention of code sharing, which makes it effectively impossible to reproduce the analysis if desired.

We have now added the links to the codes in the methods section.

Minor comments

Line 7 – abstract “Maternally inherited H3K9me3 is required for the de novo organization of constitutive heterochromatin:” It is not de novo if it is inherited

We thank the Reviewer for spotting this mistake. We have now corrected it in Line 7 as follows “Maternally inherited H3K9me3 is required for the organization of constitutive heterochromatin:”.

Reviewer #2:

Remarks to the Author:

The authors analyse H3K9me in Drosophila development and have tagged the H3K9 HMTs to follow their abundance and distribution during early stages of pre-larval growth. They distinguish HMT binding to genes and to repeats, and find that they have different kinetics of acquisition, as expected (see for example work from Zaret <https://doi.org/10.1038/s41556-021-00725-7> or Shinkai lab doi: 10.1101/gad.2027411). They then analyse a triple HMT knockout, and find, as shown in many other systems, that at least one HMT is essential for early development. The finding that H3K9me is present on germline chromatin is not new, nor is it specific to flies.

We thank the Reviewer for her/his feedback, and we agree with the Reviewer that there is a large amount of data related to the role of H3K9me in germline development.

What is new here is:

- The formal proof that H3K9me3 is inherited from the oocyte to the embryo.
- The mechanistic disentanglement of the SetDB1 (and more generally all the H3K9me3 HMTs) role in the germline versus the role in early embryogenesis. We are providing for the first time maternal mutants for those HMTs.
- The most precise and quantitative measure for the building of H3K9me3 during early embryogenesis. A resource that will be a gold mine for colleagues in the field, especially for biophysics and modeling approaches to study the formation of H3K9 domains.
- The characterization of cellular and nuclear shape phenotypes caused by the loss of H3K9me3; a finding that can have tremendous implications, especially for the field of cell biology and, more specifically, for cancer cell biology.

We would like to thank the reviewer for the extremely helpful comments related to the published literature. That has enormously helped us in re-writing the manuscript in a more broad and large-audience appealing format.

In that respect, we would like to point out that the reference suggested by the reviewer (<https://doi.org/10.1038/s41556-021-00725-7>) focuses on H3K9me importance in regulating gene expression. **However, this is not the main focus of our work.**

As a matter of fact, unexpectedly and surprisingly, our Triple mutant embryos show no obvious transcriptional defects. This finding further suggests that the role of H3K9me in early embryos is completely different from its function deeply described in somatic tissue and transcriptional regulation. Previously, many studies have focused on the H3K9me importance in somatic tissues or cells or embryos. Higher organisms like mice have at least 6 H3K9me3 HMTs, which makes it difficult to deplete all of them in early mouse embryos. All six H3K9 HMTs have been depleted by Montavon et al. 2021, but it is in somatic cells, mouse embryonic fibroblasts (MEFs). Therefore, to our knowledge, previous mice studies have done work on depleting either one, two, or three H3K9me HTMs in early embryos. Maternal and zygotic null animals for all the HMTs for H3K9 were generated in *C.elegans*, which do not require H3K9me for viability.

In contrast, in this manuscript, we deplete all H3K9me HMTs for the first time during *Drosophila*'s early embryogenesis. It allowed us to show that H3K9me3, which is found on the oocyte, is transmitted to the embryo. Moreover, we could address the question of the importance of H3K9me3 in early embryogenesis before ZGA.

They see the differences that are to be expected among the HMT homologues, Suvar3-9 and G9a vs SETDB1, in line with expectations. In somatic tissues Suv3-9 and G9a repress genes in a tissue-specific manner (targeted by TFs), which has been characterized in depth in other systems (reviewed recently in <https://doi.org/10.1038/s41580-022-00483-w>). The authors conclude that - as in multiple other systems - the dSETDB1/ESET homologue is the one that acts first in zygotic development, and it is essential in the earliest stages of growth (again this was already shown in the classic Li 2004 paper doi: 10.1128/MCB.24.6.2478-2486.2004; since then many other outstanding papers have studied the role of ESET/SETDB1 in germline and early zygotic growth - including Cell Death & Disease volume 5, e1196 (2014); <https://doi.org/10.1371/journal.pgen.1005970>; along with papers from the Torres-Padilla lab such as DOI: 10.1038/s41556-020-0536-6).

We would like to thank the Reviewer for their time and effort and apologize if it seemed that we were overlooking key literature. The Reviewer's comments have deeply helped us in re-writing the manuscript.

The 2004 paper Histone H3-K9 methyltransferase ESET is essential for early development. In the Li 2004 paper doi: 10.1128/MCB.24.6.2478-2486.2004, the authors generated Eset mouse knockouts

(zygotic null and **not maternal null**), which showed lethality between 3.5 and 5.5 dpc. **Of note, Eset is an Euchromatic H3K9 methyltransferase and not Heterochromatic as Setdb1.** Eset(-/-) blastocysts were recovered in less than Mendelian ratios and showed defective growth of the inner cell mass. Global H3-K9 trimethylation and DNA methylation at IAP repeats in Eset(-/-) blastocyst outgrowths were not dramatically altered.

The 2014 Cell Death & Disease paper shows that deleting SETDB1 during oogenesis caused oocyte arrest at MI of meiosis largely because of the misregulation of cell cycle genes. The few oocytes that passed the MI arrest were fertilized, but the embryos died before reaching the blastocyst stage. Unfortunately, due to the experimental setup, it is not possible to disentangle the germline defects from the embryonic defects.

The 2016 PLOS genetics paper Maternal SETDB1 Is Required for Meiotic Progression and Preimplantation Development in Mouse Oocyte (<https://doi.org/10.1371/journal.pgen.1005970>) showed that SETDB1 controls the global level of H3K9me2 in growing oocytes. Conditional deletion of Setdb1 in developing oocytes (maternal null) led to a meiotic arrest at the germinal vesicle and meiosis I stage, resulting in substantially fewer mature eggs. Embryos derived from mature Setdb1 null eggs exhibited severe defects in cell cycle progression, progressive delays in preimplantation development, and degeneration before reaching the blastocyst stage. Due to the experimental setup, it is not possible to disentangle the germline defects from the embryonic defects (the embryonic defects in the cell cycle could also be a consequence of the meiotic cell cycle progression defects. This is different from our experimental approach, where we can disentangle the germline function from the early embryo function). Rescue experiments suggested that the catalytic activity of Setdb1 is essential for meiotic progression and early embryogenesis. Setdb1 deficiency also led to the derepression of transposons and increased DNA damage in oocytes. This paper concluded that Setdb1 is a maternal-effect gene that controls meiotic progression and is essential for early embryogenesis.

The 2020 NCB paper Heterochromatin establishment during early mammalian development is regulated by pericentromeric RNA and characterized by non-repressive H3K9me3 from the Torres-Padilla lab (DOI: 10.1038/s41556-020-0536-6) focused on the role of Suv39h1 and h2 in the early embryo. They showed that Suv39h2 is important for de novo deposition of H3K9me3 at the paternal pronucleus and that major satellite RNA inhibits Suvar39h2 HMT activity.

As the Reviewer pointed out, dSetDB1 is essential for germline development, specifically at the very early stages of oogenesis in flies. Flies with mutant alleles do not make eggs (another name for the dSetDB1 fly mutant is Eggless). This germline defect precluded investigating the function of dSetDB1 in early embryos.

Only thanks to our new genetic and molecular tools were we able to uncover the role of dSetDB1 in the early embryo for the first time. Indeed, we show here for the first time that **embryos that are devoid of dSetDB1** (single mutant) at the early stage of **embryogenesis are viable**, which is the opposite finding compared to the studies mentioned by the Reviewer.

Overall, I would like to point out that in all of the manuscripts mentioned above and more generally in the whole literature **so far, the focus has been on the molecular and developmental function of single or double Histone Methyl Transferases in germline development and early embryos.**

On the contrary, in our work, we study and focus on the role of the posttranslational modification of the K9 of Histone H3. We molecularly dissect H3K9me role in germline inheritance and in the establishment of constitutive heterochromatin.

The authors of this current ms then look at HP1a, which in all species binds the H3K9me mark, and finds that it still binds weakly on pericentric heterochromatin in the absence of H3K9me3 (likely due to RNA interaction, which could be easily tested with a HP1 hinge-domain deletion, but was not). Once again a disconnect between HP1 and the HMT is not novel – it was already shown in fission yeast that *clr4* mutants are not identical to *swi6* mutant. This topic was extensively reviewed in 2017 Jun 11. doi: 10.1016/j.ceb.2017.05.004, covering studies in *S pombe*, mammals, as well as flies.

We would like to thank the Reviewer for the helpful suggestions. The comprehensive 2017 Current Opinion in Cell Biology review from Johnson & Straight highlights, in part, the role of RNA in heterochromatin formation. For instance, non-coding RNA limits heterochromatin spread by directly interacting with Swi6/HP1 (Keller et al., 2013, NSMB). RNA and H3K9me2 compete for binding to Swi6/HP1, even though RNA binds the Swi6/HP1 hinge domain and H3K9me2 binds the Swi6/HP1 chromodomain. Like Swi6/HP1, mammalian HP1 α can directly bind RNA through its hinge domain (Muchardt et al., 2002, EMBO reports), and HP1 α binds pericentric major satellite RNAs (Maison et al., 2011, Nature Genetics). Treatment of mammalian cells with RNase, or inhibition of HP1 α sumoylation, reduces HP1 α localization (Maison et al., 2002, Nature Genetics; Maison et al., 2011, Nature Genetics). In fission yeast, the chromodomain of Clr4/SUV39 was also found to bind RNA *in vitro* only in the presence of H3K9me3 histone tail peptide (Ishida et al., 2012. Mol.Cell).

Motivated by the Reviewer's comment, we performed new experiments to identify potential RNAs that could bind HP1 and help its recruitment to chromatin. Briefly, we performed Reverse Transcribe and Tagment (RT&Tag) (Khyzha et al., 2022. Nature Methods), which captures RNA–protein interactions, RNA–chromatin interactions, and RNA modifications. Antibody-mediated recruitment of oligo(dT) enables localized reverse transcription, which generates RNA/cDNA hybrids that are subsequently tagged by Tn5 transposases for downstream sequencing. We used an HA antibody to detect endogenously tagged HP1a-HA in ZGA embryos. As a positive control, we also performed RT&Tag using the HA antibody with endogenously tagged MSL2-HA protein, which binds rox RNAs. Our RT&Tag protocol with MSL2-HA systematically identified rox RNA, but our HP1-HA RT&Tag did not uncover any significantly enriched RNA targets (**figure A**). We have not included this negative result in the manuscript, but we can include it upon request.

Figure for reviewers removed

Finally, the authors show that heterochromatin is disorganized without H3K9 methylation (again, shown in many species, starting with the Peters et al Cell paper 2001 DOI: 10.1016/s0092-8674(01)00542-6). The current papers shows that reinduction of dSETDB1 in the zygote did not restore viability, while maternal expression did (shown, as well, in the 2004 En Li paper). This has been studied extensively in mice and there is an extensively characterized role for ESET/SetDB1 in germline and very early development (e.g. doi: 10.1242/dev.132746). The expression of a catalytically dead HMT does not restore early embryonic viability, reminiscent of what was done in mice 19 years ago (doi: 10.1128/MCB.24.6.2478-2486.2004). The authors speculate that H3K9me3 favors HP1

multimerization (also a hypothesis that has been presented by others - c.f. G. Narlikar's papers, e.g. <https://doi.org/10.1016/j.molcel.2010.12.016>). Again, however, the authors fail to discuss the interaction of HP-1 with the HMTs, although this has been extremely well established.

We appreciate these references provided by the Reviewer, and apologize for any misunderstanding – we agree that these papers deciphered the function of H3K9me and H3K9me HMTs. However, we do want to note that in Li's paper (doi: 10.1128/MCB.24.6.2478-2486.2004), due to the experimental setup, it is not possible to disentangle the germline defects from the embryonic defects (the embryonic defects in the cell cycle could also be a consequence of the meiotic cell cycle progression defects. This is different from our experimental approach, where we can disentangle the germline function from the early embryo function). In contrast, we depleted the maternal stock of dSetDB1 and restored it maternally or zygotically. Here, we clearly show that maternal dSetDB1 rescues embryonic development, whereas a maternally supplied catalytic mutant and zygotic dSetDB1 do not. These experiments show that the histone methyltransferase activity of maternally supplied dSetDB1 is required for embryogenesis. In addition, these data show that although zygotic dSetDB1 is required for development, it is not sufficient.

The 2011 Molecular Cell paper from Narlikar and Al-Sady's groups "Chromodomain-Mediated Oligomerization of HP1 Suggests a Nucleosome-Bridging Mechanism for Heterochromatin Assembly" shows that the interaction between Swi6/HP1 dimers and H3K9-methylated chromatin *in vitro* causes Swi6 to tetramerize on a nucleosome. An interface created by two chromodomains (one in each dimer) recognizes H3K9me. Their *in vitro* findings suggested that HP1 recognition of H3K9me is coupled to its spread on chromatin. In contrast, our work suggests that HP1 binding and even spreading are not compromised in TM embryos that lack H3K9me_{2/3}. Instead, our data suggest that HP1 remains diffusely localized and does not undergo phase separation into large clusters in the absence of H3K9me_{2/3}.

Nevertheless, we took the Reviewer's comment to heart, nevertheless, and to assess the HP1a binding partners in the early fly embryos, we performed Immunoprecipitation followed by Mass Spectrometry (IP-MS) on early wild-type embryos for flag-HA tagged HP1a (**figure B**). The results show that HP1a interacts with Su(var)3-9 H3K9 HMT as previously shown in other organisms (Eissenberg and Elgin., 2014, Trends in Genetics). However, we did not detect an interaction of HP1 with the main early fly H3K9me₃ transferase, dSetDB1.

Figure for reviewers removed

Despite the fact that some of these papers I mention are included in the reference list, the authors do not seem to have digested their contents, at least not sufficiently to realize that they are not asking

new questions. Rather they repeat experiments that yield results very similar to those published. This suggests a lack of familiarity with the existing literature, which of course makes it impossible for them to interpret their findings in a larger, trans-species field of H3K9me function, roles of the HMTs beyond K9me, and both K9me-dependent and -independent functions for the reader HP1. A few reviews and recent papers that appropriately cite early literature across multiple species include: TIG 2023 doi: 10.1016/j.tig.2022.12.005. NRMBC 2022 doi: 10.1038/s41580-022-00483-w; Life 2021 doi: 10.3390/life11080817; Curr Opin Genet Dev. 2021 doi: 10.1016/j.gde.2021.01.008. and Curr Opin Genet Dev. 2019 Apr;55:1-10. doi: 10.1016/j.gde.2019.04.013). There are also reviews that explore the H3K9me-independent roles of both HP1 and the H3K9 HMTs (e.g. doi: 10.3389/fcell.2022.1026406; or see references in doi: 10.1038/s41594-021-00712-4 and doi: 10.1016/j.celrep.2016.09.050). This is by far not an exhaustive list of missing information, and the authors are strongly encouraged to read the works of Paul J Lehner, Yoichi Shinkai, Ding-Sheng Jiang, which complement the papers indicated above from Peters, Gasser, Zaret, and Torres-Padilla. As for the originality and significance: except for using Cut&Run, which is a relatively new method, I fail to see any novel insight in this paper. I note that the H3K9me3 mark and its HMTs are extremely well conserved, and that even in species (such as *C. elegans*) that lack centromeric satellites (and thus suvar3-9), even there, the behavior of ESET/SETDB1 is highly analogous across species. The SETDB1 enzyme is essential for germline viability in mouse, worms and flies.

We are grateful for the Reviewers' extensive suggestions for previous literature and help. We have added most of the references in the revised version and included the works that have been done outside of the fly field.

We agree with the Reviewer that the SETDB1 function **in the germline** is very conserved across species. This is indeed what motivated us to create the **TM mutant that does not affect germline development**.

In the revision, we added new data on the nuclear shape at ZGA in TM embryos (lacking H3K9me2/3). Changes in H3K9 methylation patterns and mutations in the enzymes responsible for its regulation have been linked to various types of cancer. The correlation between nuclear shape and cancer is an intriguing area of research in the field of oncology and cell biology, and alterations in nuclear shape have been observed in many cancer types and are often considered potential diagnostic and prognostic markers. (Zink et al., 2004, Nature Review Cancer). We stained control and TM embryos with Lamin antibody and quantified the nuclear sphericity and roundedness as a proxy to quantify the abnormalities in the nuclear shape (**Figure C**). The results show that both sphericity and roundedness of TM ZGA embryo nuclei decreased significantly compared to the control ZGA nuclei.

Figure C. Immunofluorescence (IF) images control and TM ZGA embryos. 3D constructed images are shown on the left, and a slice of the same embryo is shown on the right. Quantification of ZGA nuclear sphericity and roundedness by using IF images by using Lamin staining as a proxy for the nuclear membrane. The intensity of H3K9me3 is 30 times increased in TM in order to make sure that the

embryo lacks H3K9me3.

Also, we now performed chromatin conformation capture (HiC) experiments in control and TM ZGA embryos as an orthogonal approach to check for changes in the chromatin contacts at pericentromeric regions. The results show that the contact frequencies between pericentromeric regions are significantly decreased in TM compared to the control (**Extended Data figure 3h**).

As for the text: besides the absence of novelty, there are many red flags in the text, i.e. statements that are simply wrong. In the summary ".. the importance of heterochromatin in development is unknown" is not true – c.f., the extensive literature on the topic. Similarly, in the summary, they imply that maternal inheritance of H3K9me is "de novo organization of constitutive heterochromatin" (by definition if it is inherited, it is not de novo established). On page 3 line 35-37 it is stated that "many studies have focused on the function of H3K9me2/3 in constitutive heterochromatin in somatic differentiated cells", yet just as much work has been done on the early embryonic role of these HMTs (see citations above). The final sentence of the introduction, "Whether H3K9me2/3 is essential in early embryonic development in more complex sexual organism and required to form constitutive heterochromatin remains unclear." is simply false: it is essential. I cite some of the relevant mouse studies above, which clearly show H3K9me-depositing HMTs are essential for both constitutive heterochromatin and for very early zygotic (preimplantation) development.

We thank the Reviewer for pointing out those mistakes. We have changed the sentences mentioned by the reviewer accordingly.

On the positive side: the methodology and quality of data are both acceptable. I find the data presented quite clear and the methods nicely executed. I do not dispute their conclusions, although often they do not go far enough to bring new insight (e.g. testing the linker deletion in HP1 to show a role for RNA binding). What is missing for Nature Genetics is novelty. Had the authors been more abreast of the field they might have explored questions that would increase our understanding of H3K9me and its HMTs. Do the cofactors of dSETDB1 play the same roles in early embryos as in later stages? Does the HUSH complex play key roles pre-ZGA? What might be the non-histone targets of dSETDB1 that are essential in early development? What role does transposon expression play at these stages (answered largely in mouse and worms, but could be checked in flies). The fly represents a second DNA methylation-free organism in which H3K9 methylation is crucial for fertility, thus the authors might have tried to compare their findings closely with *C. elegans* results, where DNA damage triggers a p53-dependent apoptosis in the germline. Yet they have not done this. Another important missing reference from *Drosophila* work is Smolko et al 2018 Nature Comms, 9(1) 4155 - where SETDB1 is shown to be required to maintain female identity in flies.

*We thank the Reviewer for her/his appreciation of the quality of the work and for the excellent questions she/he brought up that will broaden our understanding of H3K9me and its HMTs functions. To follow up on the Reviewer's request related to the cofactors of dSetDB1 and non-histone targets of dSetDB1, we now performed IP-MS on early embryos for endogenously flag-HA tagged dSetDB1, wild-type (left, **figure D**) and catalytic dead (right, **figure D**). Known interacting partners of dSetDB1 IP'd with both the wild-type and catalytic dead proteins, including Windei (Wde) and the uncharacterized protein that is encoded by *CG14464* gene. Other low-enriched binders of dSetDB1 could be non-histone targets, which can be explored further in the future.*

Figure D. IP-MS of endogenously *flag-HA-dSetDB1* (wild type) (left) and *flag-HA-dSetDB1* (catalytic dead) (right) on the nuclear soluble fraction of 0-4h (approx. ZGA) time collected embryos.

We note that the HUSH complex is absent in *Drosophila* but conserved from fish to humans (Tchasovnikarova et al., 2015, Science).

Regarding the expression of transposable elements (TEs) in early fly embryos, Fabry et al., 2021, Elife have shown that only a few TEs are expressed in very early embryos, and the expression of TEs, specifically the *roo* family, peaks only after embryo gastrulation. Moreover, our Global Run-On sequencing (Gro-seq) experiment on hand-staged ZGA control and TM embryos showed misregulation of only a subset of the transposable element (TE) family upon the loss of H3K9me2/3 (Supplementary Table 3_2). We speculate that most of the transposon regulation in early embryos occurs in the Primordial Germ cells (PGCs). It would be interesting in the future to study transposon regulation in PGCs in wild-type and triple-mutant embryos, therefore providing, with our triple mutant, a resource for the whole community studying transposons regulation.

As the reviewer pointed out, both flies and *C. elegans* lack DNA methylation, and in *C. elegans* double HMTs mutants cause DNA damage and triggers a p53-dependent apoptosis in the germline. We now performed staining for H2Avgamma (phosphorylated H2A.Z) in control and TM ZGA embryos (**Extended Data figure 3f**), and quantified nuclei with a high H2Avgamma signal. We did not detect significant DNA damage in the TM ZGA embryos compared to the control ZGA embryos. However, when we checked nuclei fall out (a process where defective nuclei are eliminated from the developing embryo, ensuring that only healthy cells contribute to the formation of the organism) by live imaging, we do see a significant increase in nuclear fall out nuclei in TM embryos (**figure 3a**). We also quantified mitotic defects in control and TM embryos (**figure 3a**). Live imaging data show that arrested and non-condensed nuclei are elevated in TM embryos, which lack H3K9me2/3, compared to control embryos. Moreover, there is a significantly increased proportion of fused nuclei in TM embryos than in control embryos.

Finally, we thank the Reviewer for the reference (Smolko et al., 2018, Nat comms), which shows that *dSetDB1* is required to maintain female identity in flies. We have included the reference in the manuscript. We want to emphasize that in our study, we deplete *dSetDB1* at later stages of oogenesis and that our single mutant (*dSetDB1*-KD) embryos do not show a significant decrease in hatching rate (**Extended Data figure 2e**), meaning that we did not affect *dSetDB1* function in oogenesis.

I cannot recommend this for Nature Genetics even if the paper were rewritten, because the findings are too repetitive of earlier publications. However, a revision might be appropriate for Nature Communications, if the results were discussed in the context of the above mentioned literature. The

writing as it currently stands, is substandard.

Reviewer #3:

Remarks to the Author:

In the article entitled, "Inheritance of H3K9 methylation drives the establishment of genome architecture in early embryos," Atinbayeva et al. reveal the role of epigenetically transmitted H3K9me in promoting embryonic nuclear structure in *Drosophila melanogaster*. Their data all raise the intriguing possibility that HP1a localizes to heterochromatin in a H3K9me-independent manner at this particular moment during development. The events that establish canonical chromatin organization immediately following fertilization remain poorly described. Moreover, the relative importance of transgenerational epigenetic transmission versus zygotic genome contributions to heterochromatin establishment are also remain incompletely described. The authors aim to fill this gap using an elegant and powerful integration of cell biology, chromatin profiling, and mutations in key histone methyltransferases on carefully sorted embryos of various stages.

The authors first show that H3K9me is present at the first mitotic division and accumulates as embryos cycle through to ZGA and appears, as expected, to be concentrated at DNA repeats. Furthermore, the three histone methyltransferases are all also present in the nucleus starting from the earliest embryonic cycles. The presence of H3K9me, however, disappears upon depletion of all three HMTs in the female germline, highlighting the importance of transgenerational transmission of these enzymes for sustaining/establishing H3K9me3 before ZGA. Remarkably, HP1a appears to localize to chromatin independent of H3K9me3 in the triple mutant-mothered embryos, suggesting an alternative mechanism recruiting HP1a to early embryonic heterochromatin. Despite HP1a localization to chromatin in these mutant embryos, indicators of heterochromatin integrity and silencing suggest disorganized nuclear structure. Using a biophysical model, the authors find that HP1a should be able to bind chromatin even when H3K9me is absent.

The manuscript offers a compelling window into the earliest events that establish heterochromatin in *Drosophila*. The recruitment of HP1a in the absence of H3K9me3 is certainly very exciting and raises many questions about why an alternative mechanism is employed at this developmental stage. The data should be of interest to both the chromatin and early development communities. The article lays out two major findings – the importance of H3K9me in establishing heterochromatin integrity and the H3K9me3-independent recruitment of HP1a. While these insights are certainly new for the *Drosophila* embryo, it was not clear from the context the significance of these insights in light of the mammalian literature, where there is a larger literature investigating this stage. Nevertheless, this is strong paper and I suspect that a more comprehensive and clear introduction could better set up the problem/gap in the literature.

Several questions remain that, if addressed, would significantly elevate the impact of the manuscript. We are very grateful for the Reviewer's enthusiastic comments and helpful insights to make the manuscript even more exciting. We believe the Reviewer's suggestions on the characterization of phenotypes and contextualizing the current work in light of the work done in the mammalian system have improved the current manuscript considerably. We took the Reviewer's comments to heart and performed new experiments to address the following points:

- Checked and quantified the mitotic defects, including micronuclei formation in control and TM embryos, by using H2Av-RFP for live imaging (**figure 3a**).
- Stained the control and TM embryos for H2Avgamma to check for DNA damage (**Extended Data figure 3f**)
- Contextualized the significance of the study in the light of mammalian work that has been done so far in early embryo development.
- Modified the text according to the Reviewer's comments

Major comments:

1. The authors show the consequences of disrupting the H3K9me mark during the earliest embryonic cell cycles – specifically, the disruption of heterochromatin clustering. The authors also show that these “triple mutant” embryos fail to hatch. Given that the authors are imaging DAPI-stained embryos at both early and later stages of development, I wondered why no data on chromosome segregation or genome integrity were presented. Do the chromosomes mis-segregate as a consequence of disrupted centromere clustering? Do activated transposable elements pepper the genome with new insertions? Is there rampant DNA damage/micronuclei formation? These functional consequences of the fascinating perturbations to nuclear structure are missing from this dataset and important for communicating the significance of de novo H3K9me3 establishment.

We thank the reviewer for these excellent points and suggested experiments; the result from these suggestions led to the completely new figure 3. Based on the Reviewer’s suggestion we have now performed live imaging by introducing H2Av-RFP reporter in both control and TM embryos (**figure 3a**). The results show that TM embryos have a defect in proper chromosome segregation: specifically, a significant portion of nuclei do not condense their chromosomes and arrest at metaphase—moreover, a significant portion of nuclei fuse, leading to polyploidy and nuclei death. However, there is no significant increase in micronuclei formation in TM compared to control embryos.

Based on the Reviewer’s suggestion, we also performed staining to identify DNA damage as a readout of transposons’ activity. We looked at phosphorylation of the histone variant H2Av (H2Avgamma) to identify DNA damage. We do not see a significant increase in the fraction of nuclei with high H2Avgamma (phosphorylated H2A.Z) signal compared to the control (**Extended Data figure 3f**). Since we did not see the increase in H2Avgamma signal in TM, we extrapolated that, at this developmental stage, there is not much DNA damage occurring from transposition events.

2. Related, the introduction states that “It remains unclear whether H3K9me3 is maternally inherited or deposited de novo.” H3K9me3 is a classic marker of the maternal pronucleus in mammals, contradicting this statement. Instead, if the authors might want to suggest that the rapid onset of ZGA in mammals makes it difficult to discern the relative importance of maternal deposition and de novo deposition after the dilution of maternally deposited H3K9me3.

We thank the Reviewer for pointing this out. Indeed, as the Reviewer wrote, H3K9me3 is a mark that is found in maternal pronucleus, but it was never formally proven to be inherited from the oocyte to the embryo. Here, by depleting all H3K9 HMTs in during oogenesis we formally show that H3K9me3 is maternally inherited to the embryo. Nevertheless, we agree with you and based on your suggestion we now write in the main text “Drosophila provide an opportunity to study the marks as it last so long until ZGA starts”

3. Line 48: The question of whether H3K9me3 is important for early embryonic development seems disingenuous – my understanding is that the field is currently focusing more on HOW H3K9me3 mediates early embryonic development (e.g., PMID: 32601371). The significance of this work should be better contextualized in the mammalian literature.

We thank the Reviewer for this suggestion, and we have changed the introduction and contextualized the significance of the work by taking into account the work was done in mammalian systems.

4. I am not in a position to rigorously evaluate the biophysical model; however, I must say that it’s not clear if the model was necessary to infer that H3K9me-independent recruitment of HP1a to heterochromatin can be attributed to “redundant mechanisms of HP1 recruitment via RNA/DNA or chromatin binding partners.”

We apologize to the referee for the lack of clarity. Our analysis of Cut&Tag data showed that HP1 exhibits the ability to bind to chromatin independently of H3K9me3. This assertion is supported by the nearly complete retention of HP1 even after the complete depletion of H3K9me (with only a 30%

reduction observed). However, live embryo data implied that interactions between HP1 molecules might depend on the presence of H3K9me3 marks, as HP1 foci were lost upon H3K9me3 depletion.

To rigorously explore the interplay between HP1, H3K9me3, and their interactions, we adopted a biophysical modeling approach. The overarching objective was to understand the potential essentiality of H3K9me3 in facilitating HP1-HP1 interactions while simultaneously investigating whether H3K9me3 might be dispensable for the H3K9me-dependent recruitment of HP1 to chromatin. Our thorough analysis revealed that only a model incorporating H3K9me3-mediated HP1-HP1 interactions could effectively reproduce and explain all the intricate patterns observed in our experimental data.

Minor comments

We thank the reviewer for all the suggestions. We considered and corrected them all.

- It would be helpful to see a histone modification that is erased as a control for the retention of H3K9me.

We have included H3K9ac as an example of a histone mark that is reprogrammed as the signal is not detected around cycle 6 embryos (**Extended Data figure 2g**), but it appears around cycle 8 as shown by Ciabrelli et al., 2023.

- Line 21: evolutionarily - [...] which are evolutionary conserved [...] replaced with evolutionarily Line 22

- Line 22: humans - [...] from fission yeast to humans' recognize and bind [...] replaced with humans Line 23

- Line 33: "It remains unclear whether H3K9me3 is maternally inherited or deposited de novo." H3K9me3 is a class marker of maternal chromatin in mammalian early embryos.

We agree with the Reviewer that the H3K9me3 is a classic mark of the maternal pronucleus. However, it was not functionally shown to be maternally inherited. We have clarified this message in the text.

- Lines 35-43: This paragraph could be revised heavily for clarity.

We are grateful to the Reviewer for the suggestion. We have revised the works conducted on the early embryogenesis of the mammalian system on H3K9me3, as the Reviewer suggested.

- How was the maternal pronucleus delineated from the paternal? Shouldn't there be a paternal nucleus marker to allow you to make this inference?

We thank the Reviewer as well as Reviewer 1 who also brought up this point. The maternal pronucleus is defined as the pronucleus that is stained with H3K27me3 shown in Zenk et al., 2017, Science. We have co-stained the pronuclei at apposition with H3K9me3 and H3K27me3, and the pronucleus with H3K27me3 signal is the maternal pronucleus (**Extended Data figure 2f**). We have defined the maternal pronucleus in the text (Line 120) as well as in the **figure 2** legends.

- Line 139: I am not sure I agree that a 32% reduction of HP1a on TM embryos corresponds to "slightly reduced." Similarly, the statement that loss of H3K9me3 "does not compromise the de novo recruitment of Hp1a..." is not supported by the data.

We have rewritten the part with HP1a quantification and reduction in TM embryos. [...] Unexpectedly, HP1a binding was only slightly reduced in TM embryos. Lines 139-140 [...] was replaced with [...] Unexpectedly, HP1a binding was reduced in TM embryos albeit not completely [...] Lines 218-219.

Also, [...] does not compromise the de novo recruitment of HP1a [...] Line 271 [...] was replaced with [...] does not completely compromise the *de novo* recruitment of HP1a [...] Line 368. We would like also to point out that we refined the quantification by using peaks without resizing them to 2kb. This led to a reduction of 35% instead of 32%.

- Line 167: At last – [...] At last, using DNA-FISH, we observed [...] removed Line 177

- The H3K9me staining of pre-ZGA embryos should be reconciled with previous work showing that this mark is not evident until ZGA (for example, PMID: 26915820, as cited in the manuscript

Discussion). This discrepancy could simply be an antibody issue, but regardless, it would be helpful to clarify this difference and to articulate if this work is the first to document H3K9me accumulation prior to ZGA, etc.

We have reconciled this work with Yuan et al., 2016, Genes and Development and added a text [...] Possibly, the absence of H3K9me2/3 detection in earlier studies could be attributed to antibody sensitivity issues for the low abundant H3K9me2/3 levels at the earliest stages. [...] Lines 335-337.

- Figure 1 legend: “till” – [...] Fig. 1: H3K9me3 is present from fertilization till ZGA in the early *Drosophila* embryos [...] replaced with until

- I do not understand the final repeats column of Figure 1C. Density of repeats under the H3K9me peaks?

We thank the Reviewer for pointing out this issue. The last column of **figure 1c** represents the density of repeats at H3K9me3 peaks at ZGA. We have clarified it at the figure legends.

- Figure 2c: What is the n for these images? What are the different stages of the cell cycle represent – each cycle should have an S and an M image.

The n number per stage and per genotype is at least 3 embryos. We have added the number of n under the figure 2 legends. We only show mitotic chromosome images as it is difficult to get the signals in S-phase early embryos.

Reviewers' Comments after Revision:

Reviewer #1:

Remarks to the Author:

The authors have improved the figures (i.e. the presentation of their previous data) substantially, which better supports their claims. The introduction is also better. As mentioned before, many of the points have been found in other studies, while this manuscript brings them together and for the first time in *Drosophila*. So the novelty is a bit borderline. The fact that HP1 can still bind to chromatin without H3K9me by cut-n-tag is interesting, and this does not seem to be due to RNA. Two things:

1. A small thing - To go back to my comment about using totipotent for embryos before cycle and multipotent to describe embryos after 9-13.

I was referring to the transition point specifically at NC 9 from totipotent to pluripotent. The paper by Haigh et al. – injected nuclei at 70-100 minutes and showed that they could clone *Drosophila*. But this was a broad time-window. They didn't test later collections 100-150 minutes, for example, and show that they are not totipotent.

Also given the efficiency (~0.5%) – perhaps it was only the nuclei at 70-80 minutes that did the job. Your statement is very temporally specific to when this transition occurs from totipotent to pluripotent – but there is no precise data at this temporal resolution to make this claim.

We thank the Reviewer for the point. We have removed those words from the text and from the figures.

2. The addition of the Hi-C data brings nothing as it is presented.

There are no quality control metrics provided. The authors state that “The results show that the contact frequencies between pericentromeric regions are significantly decreased in TM compared to the control (Extended Data figure 3h).”

How many informative reads (normalised counts) are in the TM data set compared to the control? If it is drastically lower – of course there will be lower normalised counts.

If the result is biological, and not technical, it should be specific for pericentromeric regions.

Show the same plots for euchromatic regions.

Show the reader the data – provide contact maps for pericentromeric regions and euchromatic regions in TM and WT embryos in a Supplementary figure.

We thank the Reviewer for these points. We have removed the HiC data from the manuscript.

Reviewer #2:

Remarks to the Author:

The revised article by Atinbayeva et al. has been improved by editing and the addition of some additional data, along with clarification and re-phrasing of many of their earlier conclusions. The authors have made some effort to embed their results into the rich landscape of data on H3K9 methylation and its role in both genome stability and gene expression in development. The added information on Windei and the ARLE-14 homologue are useful. A more complete analysis of TE expression is included. However, the authors continue to ignore some very important work that they may think “compromises” the originality of their own efforts... Notably work in *C. elegans*,

which similarly showed that H3K9me was inherited through meiosis in the mother / hermaphrodite (among others: transgenerational inheritance in Woodhouse et al. Cell Rep 2018 10.1016/j.celrep.2018.10.085 or doi.org/10.1016/j.cub.2017.03.008 or one specifically looking at repressive chromatin and parallel pathways of inherited repression Padeken et al., Genes Dev. 2021 doi: 10.1101/gad.344234.1202021). Actually, correctly positioning the *Drosophila* results in the field would allow the authors to highlight the originality of their contribution accurately.

We appreciate the reviewer's input; however, we respectfully disagree with their interpretation of the cited studies. Upon reviewing the three articles provided by the reviewer (Woodhouse et al., Cell Reports 2018, DOI: 10.1016/j.celrep.2018.10.085; Lev et al., Current Biology 2017, DOI: 10.1016/j.cub.2017.03.008; Padeken et al., Genes & Development 2021, DOI: 10.1101/gad.344234.1202021), we find that none of them conclusively demonstrate meiotic inheritance as claimed:

1. Lev et al. (2017) explores how MET-2, a homolog of dSetDB1, influences transgenerational inheritance via heritable small RNAs, rather than providing direct evidence of H3K9me meiotic inheritance. The study observes that in *met-2* mutant worms, the regulation of strong RNAi inheritance is not mediated by H3K9me2. It notes similar levels of H3K9me3 in the F15 and P0 generations at the transgene region exclusively under *met-2* mutation, correlating this with small RNA abundance. Although the term "inherited H3K9me" is used, the paper does not functionally demonstrate germline inheritance.

2. Woodhouse et al. (2018) hypothesizes that the germline H3K9 HMT SET-25 may initiate a heritable silencing signal, potentially through the anchoring of heterochromatic arrays to the nuclear periphery. However, this study also stops short of functionally demonstrating that H3K9me is germline inherited.

3. Padeken et al. (2021) investigates the establishment and maintenance of H3K9me3 using a heterochromatic array but does not provide functional evidence of H3K9me3 meiotic inheritance.

In summary, while these studies contribute valuable insights into the mechanisms of epigenetic regulation, they do not directly support the claim of H3K9me meiotic inheritance.

Although improved, the paper as currently written still shows a limited mastery of the field, and still contains a few false statements about past literature (particularly in other species; required changes listed below). Overall the paper is descriptive, providing limited insight into how the HMTs contribute to heterochromatin inheritance and the deposition of H3K9me in early development. On the other hand, the authors do a better job at defining what they actually show in the revision.

It would now be important to note whenever what they show is fly specific (and why), and to underscore when it supports previous studies. Being done in *Drosophila* does not make the work any less interesting. However, to merit publication in a mainstream journal, the paper must elaborate clearly which results are "fly specific" and which reinforce published results from either *C. elegans* or mammalian organisms. It would help, for instance, to insert the word "*Drosophila*" in the title (Inheritance of H3K9 methylation in *Drosophila* regulates genome architecture in early embryos" or Inheritance of H3K9 methylation regulates genome architecture in early *Drosophila*

embryos”)... especially as it is not true in all species.

We understand the concerns of the Reviewer, and we expressly stated that the results were generated in flies where needed. Also, we included previous works in *C.elegans* and mice where applicable.

Other sweeping generalizations that should be modified to reflect critical thinking as described below.

The three main three weaknesses in the paper are:

1. They repeatedly and erroneously equate the loss of the HMTs (triple mutant) with loss of H3K9 methylation, when they discuss downstream phenotypes. This is not accurate nor warranted. Of course, the loss of three HMTs leads to a loss of H3K9me, but it is well established that SETDB1 has other targets that could be responsible for some of the phenotypes, not to mention indirect effects. Thus – for example – at the end of the abstract where it is stated that changes in chromatin organization arise in the HMT mutants, the authors state that H3K9me is essential for compact chromatin organization. This is an overstatement and is not justified by their data. The correct conclusion is, “the loss of H3K9 HMTs triggers the loss of H3K9me and correlates with changes in chromatin organization.” In other words, the observed changes in chromatin organization correlate with loss of HMTs and whereas they may stem from loss of H3K9me, they could also stem from other events linked to HMT loss - such as DNA breaks, improper nuclear envelope formation, and/or changes in the timing of replication or cell division. Or, might stem from the lack of methylation on other targets of these HMTs. In other words, they cannot conclude that loss of H3K9me3 per se is the direct cause of chromatin misorganization. What they do show is that HP1 binding only partially correlates with compaction and does not always require H3K9me. The overstatement that H3K9me is essential for proper chromatin compaction is all the more unjustified in view of recent papers (e.g. *iScience* 2023) that show that cytoplasmic SETDB1 is important for maintenance of pluripotency in mouse ES cells and that SETDB1 regulates PRC2 independently of H3K9me (*Genome Res.* 2015 (9):1325-35. doi: 10.1101/gr.177576.114.) In brief - SETDB1 has functions beyond H3K9me. Loss of H3K9me correlates with loss of chromatin compaction; the HMTs are necessary (but probably not sufficient) for chromatin compaction. This is an important distinction, as to claim causality is misleading.

We acknowledge the reviewer's concerns regarding the potential for H3K9 histone methyltransferases (HMTs) and other HMTs to target non-histone substrates. This emerging area of research is indeed intriguing. Reflecting on the extensive body of work surrounding these enzymes, we present our findings as observed. In response to the reviewer's comments, we have incorporated a note of caution in our discussion, acknowledging the possibility that observed phenotypes may arise from the action of HMTs on non-histone targets.

The study published in *iScience* 2023, which investigates the cytoplasmic functions of SETDB1 in mouse embryonic stem cells, presents an interesting case. However, we advise careful interpretation of these findings due to potential limitations in experimental controls. For example, the study's approach of expressing SETDB1 with a nuclear localization signal (NLS) in a SETDB1 knockout (KO) background raises questions about whether the nuclear levels of SETDB1-NLS match those of wild-type SETDB1. Discrepancies in SETDB1 levels could explain the observed effects, as demonstrated in *Drosophila*, where maternal overexpression of Su(var)3-9 leads to developmental defects, whereas overexpression of dSetDB1 does not. This difference may be attributed to the fact that, despite dSetDB1 overexpression, nuclear levels remain unchanged,

with excess dSetDB1 localizing to the cytoplasm (based on our data). Such nuances underscore the need for more rigorous investigations in this field.

Furthermore, Genome Research 2015 (DOI: 10.1101/gr.177576.114) references the work of Dodge et al., 2004, highlighting that SETDB1 loss does not significantly reduce H3K9me3 levels, supported by ChIP-seq data showing minimal overlap between SETDB1 and H3K9me3 peaks in mouse embryonic stem cells. Contrarily, our study reveals a substantial decrease in H3K9me3 peaks following dSetDB1 loss in a single mutant, suggesting possible tissue-specific functions of the enzyme. Additionally, while the cited study reports interactions between SETDB1 and PRC2 components EZH2/SUZ12, our dSetDB1 immunoprecipitation-mass spectrometry (IP-MS) does not identify PRC2 complex members, nor do we observe embryonic developmental defects in single maternal dSetDB1 mutant embryos. Defects only emerge when all H3K9 HMTs are deleted, indicating that the complete removal of H3K9me2/3 in triple mutants is necessary for the observed developmental issues. While we cannot exclude the possibility of redundancy among H3K9 HMTs in methylating non-histone substrates, we believe this to be unlikely based on our findings.

2. The authors stress that they study the impact of loss of H3K9me on constitutive heterochromatin (in particular centromeric satellite repeats) which are particularly abundant in flies. They state that the impact of constitutive heterochromatin (satellite and TE marked with H3K9me) in development is unknown (line 2 of abstract). However, Zeller et al (Nature Genetics, 2016) showed that the loss of H3K9me (both maternal / hermaphrodite and in the offspring) does not affect early development up to gastrulation. And En Li (Mol Cell Biol. 2004 doi: 10.1128/MCB.24.6.2478-2486.2004) showed that very early mouse development is strongly affected (loss of SETDB1 is embryonic lethal). The accurate statement would be “It is unclear exactly HOW loss of H3K9me affects early development”. It is true that mechanistic differences among species remain to be clarified. Note that while early phenotypes triggered by loss of SETDB1 in mouse were in part attributed to changed expression of genes, in worms germline apoptosis (i.e. oocyte and early zygotic apoptosis) was due to lost repression of satellite repeats, which led to genome instability (Zeller et al., Nature Gen 2016; Padeken et al., Genes Dev 2019). What’s the difference then, between *C. elegans* and flies (or mouse) ? Worms are holocentric, lack centromeric satellite repeat arrays, and meiotic and mitotic chromosome segregation is 100% intact in H3K9me-deficient worms, which is not the case in flies or mammals. Centromeric satellites play a structural role in proper chromosome segregation (I note that this current ms claims there were no “lagging chromosomes” in the TM, but that there was high cellular lethality in early embryos which may well be due to chromosome missegregation – or even genome instability - that went undetected due to cell loss). Such phenomena are consistent with delayed development, which they do observe. In worms there is also misexpression of about 300 genes in mid-embryonic development upon loss of H3K9me, but none is essential for early development. This is nearly the same conclusion that the authors draw from this current study in *Drosophila*... (lines 157 – 158) although with time they do see nuclear fusion and altered chromatin structure which was not reported in worms. More information on the mechanism would have been helpful; for instance, what about other H3K9me readers, other histone marks, etc.? In any case, the authors should not overinterpret their results nor claim to be unique when they are not, but rather they should note where their findings support or are supported by work in other species, and where not. This is likely to shed light on mechanistic differences that are species-specific with respect to genomic repeats.

We are grateful for the reviewer's feedback. In response, we have indeed included additional text to highlight the distinctions between *C. elegans* and flies/mice regarding mitotic anomalies in lines 405-409. However, we must respectfully disagree with the reviewer's observation concerning the absence of "lagging chromosomes" in fly embryos lacking H3K9 methylation and its associated histone methyltransferases (HMTs). Those data are shown in Fig 3A

To investigate mitotic defects, we employed live imaging techniques and also conducted counts of nuclei undergoing apoptosis due to mitotic errors. Our analysis specifically focused on nuclei post-division, and we did not observe an increase in the incidence of lagging chromosomes in triple mutant (TM) embryos. Additionally, we have introduced a discussion on the potential link between satellite sequences and mitotic defects to further explore this phenomenon in lines 402-405 .

3. The authors attribute basically all phenotypes to changes in heterochromatin compaction... and note with surprise that not "all satellites and TEs" are derepressed upon loss of H3K9me HMTs. This is not surprising and was already shown. Obviously, as they mention in the discussion, if the relevant TF is not present in the early embryo, loss of H3K9me is not sufficient to allow de novo gene (or TE) expression, and it has already been well documented that (only) about 30% of the simple satellite repeat classes show derepression in *C. elegans* embryonic cells upon loss of H3K9 HMTs (op cit). Moreover, loss of H3K9me is not sufficient to induce expression of a locus or satellite repeat that loses the mark. The limited expression of repeats despite loss of H3K9me in early embryos is not only shown in *C. elegans*, yet in worms individual repeat transcripts could be mapped quantitatively so the data are highly convincing. Once again, the claim of novelty is not true.

We thank the Reviewer for pointing out this point. We have noted that the results we see regarding the mis-regulation of only some repeats in our mutant is similar to *C.elegans* and mouse embryo results and added references (Zeller et al., 2016; Padeken et al., 2019) accordingly in lines 185-190.

Despite these weaknesses, this highly descriptive paper makes some interesting observations, and as stated earlier, if more care were given to integrating this fly work into the complete landscape of H3K9me in development, this paper could be a valuable contribution. At present the following specific points (and those mentioned above) need to be remedied.

In the summary line 2: should read: "...how constitutive heterochromatin arises in the embryo and how it contributes to development are unknown". Actually – if inherited why should it "arise" ? They do not address the very key question of how de novo H3K9me is targeted to the right spot - which clearly happens, even in flies.

We appreciate the reviewer's observation. It's important to clarify that prior research has not definitively demonstrated the germline inheritance of H3K9me3. Consequently, we have chosen the term "arise" to describe the process. The mechanism of de novo targeting of H3K9me to specific genomic locations is indeed a fascinating topic. However, we believe that exploring this mechanism falls outside the scope of our current manuscript.

Line 4: H3K9me3 is transmitted from the maternal germline in *Drosophila*, unlike most histone modifications (!?). What is the justification for this statement ? The author defends it in his rebuttal saying H3K9ac is lost but acetylation is known to be a highly transient mark, and is hardly

“most” histone modifications. What evidence is there that most histone methylation or even mono ubiquitination is lost in the maternal germline? Paternal germline is obvious due to histone replacement by protamines, but references are needed for the statement that “most histone marks” oocytes are lost. If references supporting this do not exist, the statement should be deleted.

We agree with the Reviewer. We deleted the phrase in line 4.

Line 7: “Maternally inherited H3K9me3 is required for the organization of constitutive heterochromatin: early embryos lacking H3K9me3 display de-condensation of pericentromeric regions, centromere-centromere” - see comment 1 above. Actually, these are correlations, the genetically defined “cause” is deletion of the HMT, not histone H3K9me3: the exact mechanism is missing, and there may be other HMT targets that play a role. It is not sufficient to state that H3K9 to R mutation also shows defects, since arginine rarely can substitute for lysine.

As we wrote at the beginning of the response to Reviewer2, we agree that we can't rule out that the H3K9 HMTs might have non-histone targets that may play a role. We added “HMTs” in the sentences where necessary in line 7.

Line 12: The authors are correct in saying that “H3K9me3 is largely dispensable for HP1 recruitment.” However, they should take note that a drop of 35% could well be significant, in particular as HP1 binding to chromatin through other mechanisms is not sufficient to drive the proper embryo development. But more importantly, it is an overstatement to write that H3K9me3-dependent genome organization is essential for embryonic development. The accurate statement is, “Loss of the HMTs leads to a loss of H3K9me, which correlates with a change in genome organization and impaired embryonic development.”

Until a mechanism is defined that explains what H3K9me3 or me2 does, it remains a correlation. At present, what they can say is that the HMTs are responsible. My guess is that other polyvalent H3K9me readers are involved (not HP1)...

We changed the sentence according to the suggestion of the Reviewer in lines 15-16. It would be very interesting to find out the other polyvalent readers of H3K9me in the future in fly embryos.

Introduction:

Lines 28-29: The same criticism raised above, can be brought to the conclusion that long-distance contacts are the mode of transgenerational inheritance for the Pc mark: actually all data point to a correlation: loss of H3K27me3 leads to altered long-range interactions, and correlates with loss of heritable repression. There may be other unrecognized essential factors that trigger either the clustering or the repression.

We changed the sentence to following “Moreover, loss of H3K27me3 in *D.melonogaster* leads to altered long-distance chromatin contacts and it correlates with a loss of transgenerational heritable repression.” in lines 35-38

Lines 35-36: “However, whether H3K9me3 is maternally inherited or deposited de novo after fertilization is not functionally shown.” I believe this is known: Antoine Peters' laboratory has made maternally deficient SETDB1 and other HMT deletions in mouse and monitored H3K9me in the oocyte and zygote. In *C. elegans* H3K9me2 and me3 can be de novo deposited after fertilization (Padeken et al. GD 2019).

We respectfully differ in opinion from the reviewer. The laboratory of Antoine Peters has indeed conducted studies involving the depletion of SETDB1 and other histone methyltransferases (HMTs). However, to conclusively demonstrate the inheritance of H3K9 methylation, it is

necessary to eliminate all maternally expressed HMTs simultaneously and then examine the maternal pronucleus. Such an approach has not been undertaken in either mice or *C. elegans*. It's important to note that the *C. elegans* H3K9 HMT mutants referenced are complete knockouts, not conditional mutants specific to maternal expression.

Lines 45-48, it should be mentioned that the sextuple ko was in tissue culture cells, and that it was not necessary to knock out all 6 to see early embryonic defects in mouse. The sextuple ko is not essential for observing phenotypes in early embryos because the enzymes are not all expressed equally at all stages.

We have added that sextuple KO was performed on tissue culture cells as well as more studies involved in DAPI dense region formation in discussion in lines 445-456.

Minor note: simultaneously is misspelled.

We corrected it in the sentence.

In the rebuttal the authors state that ESET is a euchromatic HMT. That is not true, actually SETDB1 = ESET (names were used interchangeably in early mouse H3K9 HMT literature – they are the same gene). This enzyme modifies satellite repeats and TEs often depositing me1 that is then modified to me2-me3 by SUV39 or to me2 by G9a. G9a is the mammalian HMT that is most often correlated with gene promoters, and it is sometimes called the “euchromatic” H3K9 HMT. Although G9a is found in euchromatic zones to repress genes, it also regulates Telomeric repeats and transposable elements, especially intact (recent) viral insertions. In worms the homologue most similar to SUV39 and G9a is the enzyme that targets intact transposons. This is not what one called “euchromatin”. This confused statement made forcefully in the rebuttal illustrates a limited mastery of the field. In brief, the HMTs are neither “euchromatic” nor “heterochromatic”; some complexes may show bias. But above all, the authors need to understand that ESET/SETDB1 are one HMT.

We agree with the Reviewer and we thank the Reviewer for the point. We apologise for the non-correct sentence in the rebuttal.

Line 56 – stress that this is in flies: Here by depleting all H3K9 HMTs, we show that H3K9me3 in *Drosophila* is inherited from the oocyte to the zygote, and it is actively maintained throughout early embryonic divisions. (again it might be good to mention that this is not a unique or novel observation, cf worms, where maternal (hermaphrodite) deletions were made; citation is needed if there is no inheritance in mouse).

We have cited works in mouse that show H3K9me3 at maternal pronucleus in lines 43-46, but it was not formally proven in mouse if the maternal H3K9me3 is inherited from oocyte to the embryo.

Lines 63 -65: “In the absence of H3K9me3, embryos have defects in chromosomal segregation, pericentromere compaction, repression of repetitive elements and nuclear shape.” This statement is OK as it does not claim direct causality. It would be even better to state that this is a correlation or to state that it arises from HMT deletion. As stated above, there may be other targets or indirect effects leading to any or all of these phenotypes.

We added “ In the absence of H3K9me3 and its HMTs, embryos have defects in chromosomal segregation, pericentromere compaction, repression of repetitive elements and nuclear shape.”

In

lines

85-87

Lines 152-53 : they state: “Only a few genes were misregulated in TM embryos compared to control embryos, similar to results seen in mouse embryos [37], and these genes were not involved in the formation or maintenance of constitutive heterochromatin “ Also true in *C. elegans* embryos deficient for H3K9me [ref 23].

We added *C.elegans* and the reference for it in lines 179-181.

In the Discussion: lines 330 -333: mention *Drosophila* more prominently. “We show that H3K9me3 resists early epigenetic reprogramming during meiosis in *Drosophila*, and it is intergenerationally inherited from the maternal germline to the embryo. “

We changed as follows “We show that H3K9me3 resists early epigenetic reprogramming during meiosis in *D. melanogaster*, and it is inter-generationally inherited from the maternal germline to the embryo” in lines 369-370.

Lines 326 – 328 “The presence of H3K9me3 mediates correlates with clustering and compaction of heterochromatin; (3) Loss of H3K9me3 correlates with improper mitosis and altered nuclear shape (4) HP1a can bind heterochromatin in an H3K9me3-independent manner; (5) HP1a cluster formation depends on the presence of H3K9me3 (Fig. 6a).

We did not change this sentence since we believe that it is going to be repetitive to mention *Drosophila* again.

Line 381 (minor point) “HP1a promiscuous nucleic acid binding”. Should be promiscuous.

We corrected it.

Lines 388 – 389 they state: “Instead, in mouse cell lines, the complete removal of all six H3K9 HMTs leads to almost complete loss of DAPI dense regions [73]. This means that there might be additional mechanisms to form the DAPI dense regions in early fly embryogenesis.”

Importantly here (and above at Lines 330 – 335), it would be critical to test (or at least mention) other H3K9me readers – most notably the MBT-domain factors (discovered in flies) which are known to bind H3K9me2/3 (for early review see Reinberg summary from 2010 (Bonasio et al., *Semin Cell Dev Biol.* 2010 Apr; 21(2): 221–230 doi: 10.1016/j.semcdb.2009.09.010). MBT domain proteins have structural motifs that bind specifically methylated lysines (many have been co-crystallized), and for some of the major homologues, the MBT protein binds H3K9me1/2 and me3 (Koester-Eiserfunke and Fischle 2011). Importantly, because the MBT motifs are duplicated (sometimes 4 in a row) the MBT domain proteins can bind multiple histone tails and may be instrumental in bringing nucleosomes together (compaction). MBT family was in fact discovered in flies (Wismar J, Löffler T, Habtemichael N, et al. The *Drosophila melanogaster* tumor suppressor gene lethal(3)malignant brain tumor encodes a proline-rich protein with a novel zinc finger. *Mech Dev.* 1995;53:141–154). There are three *Drosophila* orthologues, dL(3)MBT, dSCM, and a protein containing four MBT repeats known as dSFMBT. Whereas dSCM is implicated in Pc mediated regulation, dL(3)MBT was identified as a regulator of the E2F/Rb pathway in a genome-wide RNAi screen (which acts through H3K9me), and this pathway has many parallels to the *C. elegans* homologue LIN-61 which binds H3K9me2, K9me3 (see paper: H3K9me2/3 Binding of the MBT Domain Protein LIN-61 Is Essential for *C. elegans* Vulva Development; Koester-Eiserfunke and Fischle 2011 PLOS Gen doi.org/10.1371/journal.pgen.1002017). Fischle showed that this key *C. elegans* MBT-repeat-containing protein LIN-61 acts with other class B synMuv proteins to regulate vulval development through gene repression. Furthermore, LIN-61 has a role in maintaining genome stability not evident for other SynMuv class factors, LIN-35 or LIN-15B. While *C. elegans*

LIN-61 binds H3K9me2-me3, it is not required for peripheral anchoring of heterochromatin, yet it is necessary for full repression of a heterochromatic array in worms (ref 22).

The paper in general is highly descriptive and it would elevate it greatly to dig more deeply into mechanism (e.g. Do other H3K9 readers contribute to compacted chromatin? What about other marks like H3K27me3 ? or histone ubiquitination ? Is HP1 acting redundantly with a MBT protein ? Which one ? Does RNA recruit HP1 ? Does genome stability and nuclear shape correlate with MBT domain protein binding to H3K9me3 or with Rb/E2F pathway which appears to overlap with H3K9me repression ?). For a Nature Genetics paper one would expect this level of mechanistic insight.

We extend our gratitude to the reviewer for their insightful comments. We are indeed examining the levels of other histone marks in the absence of H3K9 methylation.

In our investigation into the role of MBT domain-containing proteins in *Drosophila melanogaster*, we found no evidence of interaction between dSetDB1 and the three known *Drosophila* MBT domain-containing proteins: dScm, dL(3)mbt, and dSfmbt, based on our dSetDB1 immunoprecipitation-mass spectrometry (IP-MS) studies. However, our additional experiments involving HP1a-IP-MS from nuclear extracts of 0-4 hour embryos revealed a low, yet significant, enrichment of the dSfmbt protein, which closely resembles the *C. elegans* LIN-61 protein (Bonasio et al., 2010). This suggests the possibility that HP1a may facilitate the interaction between the MBT domain-containing protein dSfmbt and H3K9 methylation. Nonetheless, this hypothesis requires further detailed investigation. We wish to highlight to the reviewer that this aspect is not the central focus of our current manuscript, as addressing this question comprehensively represents a substantial and separate line of inquiry.

Lines 347 - 348: H3K9me2/me3 and K9-HMTs clearly play roles in differentiated tissue maintenance (shown in worms, flies and mammals) and there are abundance examples in mammalian tissues where H3K9 HMTs are essential for hematopoiesis, neuronal differentiation, muscle etc. I agree that this is not the focus of this current paper, but it should not be stated (line 348) that "H3K9me could be involved in differentiated tissue maintenance." It has been shown in many systems, by literally 100's of papers that H3K9me helps repress tissue-specific genes. Importantly, the majority of genes that gain H3K9me in early development are tissue specific genes – in worms many are germline genes which have to be repressed in differentiated tissues [ref 61]. This is also true in mouse.

We understand the point of the Reviewer and we have changed the sentences as following "H3K9me2/3 instead is important for tissue-specific gene regulation at later stages of embryogenesis [62, 63]. Indeed, we identified new H3K9me3 peaks that appear at euchromatic repeat-free regions in embryos at the end of embryogenesis. This newly established H3K9me3 could play a role in tissue-specific gene regulation in fly embryos [7, 62, 63]." in lines 390-394 We wrote the last sentence like "This newly established H3K9me3 could play a role in tissue-specific gene regulation in fly embryos [7, 62, 63]." as it has to be tested in fly embryos if it is the case.

Lines 398 – 410 – "The binding of HP1a in the absence of H3K9me2/3 suggests that there are other mechanisms for HP1 binding to chromatin. One mechanism could be via RNA." Should read: "could be through HP1 binding to chromatin associated RNA." otherwise it does not make sense. Actually the authors could be more explicit in this context as there is ample evidence: "HP1 can bind chromosomes through its association with chromatin-bound RNA (give refs) or chromatin motif-binding proteins, such as the AT-hook domain protein HP2, given that HP2 binds HP1" add refs.

We did not find any articles that claim HP1a binding to chromatin through chromatin associated RNAs. Evidences come from *in vitro* studies, or treatment of mammalian cells with RNAses and dispersal of DAPI dense regions. Since, there is no direct evidence, we left the sentences as they are.

Line 420: this last line is again an overinterpretation (or overstatement) as it implies that they are making an original insight when there is ample published evidence for this from other species. If there is no appropriate TF expressed at this stage, loss of K9me will not derepress a promoter (e.g. ref 61, but originally proposed by Amanda Fisher in early 2000's see her reviews). Moreover, there may be other mechanisms of H3K9me repression (LIN-61 , other MBT proteins) that act in parallel. It has been well established that loss of HP1 does not equal delta K9 HMT - amply shown by Julie Ahringer, among others.

We agree with the Reviewer that there could be mechanisms other than HP1a that are responsible for gene repression. We deleted the sentence.

Dear Dr. Iovino,

Thank you for submitting your manuscript for consideration by the EMBO Journal. I was unfortunately not able to get feedback from referee #3 and did not contact referee #1 given that this referee was already satisfied with the revisions during the previous assessment of your manuscript. I therefore decided to contact referee #2 who was initially concerned regarding the novelty of the results as well as their interpretation in context of the literature. As you can see from the comments referee #2 thinks that the manuscript in its current form is well suited for publication at The EMBO Journal.

Given this positive feedback there remain only editorial points that have to be addressed before I can extend formal acceptance of the manuscript.

Our data editors have noted the following points:

- MANUSCRIPT FORMAT: ms needs to be in .docx format;
- figures need to be removed from ms file, uploaded as individual files, and figure legends should be placed below the References
- Please add the ORCID ID for Zhan
- Please add up to 5 KEYWORDS below the Abstract
- Please reformat the REFERENCE into alphabetic order, with 10 authors + et al.
- Please rename the COI section into "DISCLOSURE AND COMPETING INTERESTS STATEMENT"
- Please remove the AC/CRedit section
- Synopsis:

Papers published in The EMBO Journal are accompanied online by a 'Synopsis' to enhance discoverability of the manuscript. It consists of A) a short (1-2 sentences) summary of the findings and their significance, B) 3-4 bullet points highlighting key results and C) a synopsis image that is 550x300-600 pixels large (width x height, jpeg or png format). You can either show a model or key data in the synopsis image. Please note that the image size is rather small and that text needs to be readable at the final size. Please send us this information together with the revised manuscript.

- Please rename the "Summary" section to "Abstract"
- Please correct the section order to: title page with complete author information, abstract, keywords, introduction, results, discussion, materials & methods, data availability section, acknowledgements, disclosure and competing interests statement, references, main figure legends, tables, expanded figure legends.
- My colleague Hannah Sonntag will contact you shortly with additional details regarding our source data policy and with a checklist which will help you to compile all the necessary source files.

Should you have any additional questions regarding the requested formatting changes or our editorial policies please consult the attached document or contact me directly. Thank you for the opportunity to consider your work for publication. I look forward to your revision.

Yours sincerely,

Cornelius Schneider

Cornelius Schneider, PhD
Editor
The EMBO Journal
c.schneider@embojournal.org

For more details on our Transparent Editorial Process, please visit our website:
<https://www.embopress.org/page/journal/14602075/authorguide#transparentprocess>

We generally allow three months as standard revision time. As a matter of policy, competing manuscripts published during this period will not negatively impact on our assessment of the conceptual advance presented by your study. However, we request that you contact the editor as soon as possible upon publication of any related work, to discuss how to proceed.

We realize that it is difficult to revise to a specific deadline. In the interest of protecting the conceptual advance provided by the work, we recommend a revision within 3 months (11th Jun 2024). Please discuss the revision progress ahead of this time with the editor if you require more time to complete the revisions. Use the link below to submit your revision:

Referee #2:

The paper by Atinbayeva et al summarizes the role of the Histone methyltransferase dSETDB1 in early fly development. They show that H3K9me3 is inherited from the maternal germline to the next generation, and is not erased. The mark is necessary for proper compaction of repetitive sequences. It appears that Hp1 is not recruited by this mark in the germline and that H3K9me3 itself drives genome organization. Compaction could also result from the binding of the HMT and/or of other ligands.

The paper relies on quantitative imaging and is somewhat descriptive, but the questions are important and well phrased. The data obtained throughout early embryogenesis are unique.

The disconnect between HP1 and H3K9me3 has been noted in other species, but it is nonetheless important, given the propensity of HP1 to cluster or form foci. I have no major revisions to suggest for the paper and recommend that it is accepted by EMBO Journal.

All editorial and formatting issues were resolved by the authors.

Dear Dr. Iovino,

I am pleased to inform you that your manuscript has been accepted for publication in the EMBO Journal.

Yours sincerely,

Cornelius Schneider, PhD
Editor
The EMBO Journal
c.schneider@embojournal.org
